# Calibrating Transformers via Sparse Gaussian Processes

**Wenlong Chen & Yingzhen Li**
Imperial College London
{`wenlong.chen21, yingzhen.li`}@imperial.ac.uk

## Abstract

Transformer models have achieved profound success in prediction tasks in a wide range of applications in natural language processing, speech recognition and computer vision. Extending Transformer's success to safety-critical domains requires calibrated uncertainty estimation which remains under-explored. To address this, we propose Sparse Gaussian Process attention (SGPA), which performs Bayesian inference directly in the output space of multi-head attention blocks (MHAs) in transformer to calibrate its uncertainty. It replaces the scaled dot-product operation with a valid symmetric kernel and uses sparse Gaussian processes (SGP) techniques to approximate the posterior processes of MHA outputs. Empirically, on a suite of prediction tasks on text, images and graphs, SGPA-based Transformers achieve competitive predictive accuracy, while noticeably improving both in-distribution calibration and out-of-distribution robustness and detection.

## 1 Introduction

Significant improvements have been made for accuracies in prediction tasks for computer vision, speech recognition and natural language processing using deep learning (He et al., 2015; Graves et al., 2013; Vaswani et al., 2017). In particular, Transformers (Vaswani et al., 2017) based on multi-head attention (MHA) have gained popularity in recent years. With Transformers being deployed in many downstream applications (Vaswani et al., 2017; Dosovitskiy et al., 2021; Brown et al., 2020), it is crucial to prevent poor robustness which often comes from erratic outputs with high confidence from these models (Guo et al., 2017b; Mukhoti et al., 2020). This requires calibrated uncertainty quantification for Transformers which is much less well-studied at the time of this work, and it raises concerns about using Transformers for safety-critical tasks which require rational and risk-averse decision making under uncertainty.

Regarding uncertainty quantification, Bayesian inference is a powerful and principled framework to build probabilistic models for rational prediction and decision-making under uncertainty (Gal, 2016). Significant progress is observed for applying (approximate) Bayesian inference methods to quantify uncertainty in fully-connected, convolutional and recurrent neural networks (Blundell et al., 2015; Gal & Ghahramani, 2016; Zhang et al., 2019; Ritter et al., 2021). Initial efforts have been made on extending these techniques to Transformers but with mixed results (Tran et al., 2019; Xue et al., 2021).On the other hand, Gaussian processes (GPs) are gold standard methods for tasks requiring reliable function-space uncertainty estimates (Rasmussen & Williams, 2006; Wilson et al., 2020). Researchers have proposed to integrate deep learning ideas to GP model design, including deep kernel learning (Wilson et al., 2016) and deep GPs (Damianou & Lawrence, 2013; Salimbeni & Deisenroth, 2017). Still these models have yet to be scaled to modern deep learning tasks such as large-scale image classification and language modelling.

In this work, we propose sparse Gaussian process attention (SGPA), a novel uncertainty quantification technique for attention-based models (e.g., Transformers), by leveraging techniques from sparse variational Gaussian processes (SVGP) (Snelson & Ghahramani, 2005; Hensman et al., 2013) for improved uncertainty estimates. Our work presents the following insights and contributions:

- Our key observation is that kernel-based attention (Tsai et al., 2019) is equivalent to the posterior mean of an SVGP. This inspires us to extend SVGP to Transformers for uncertainty estimation.

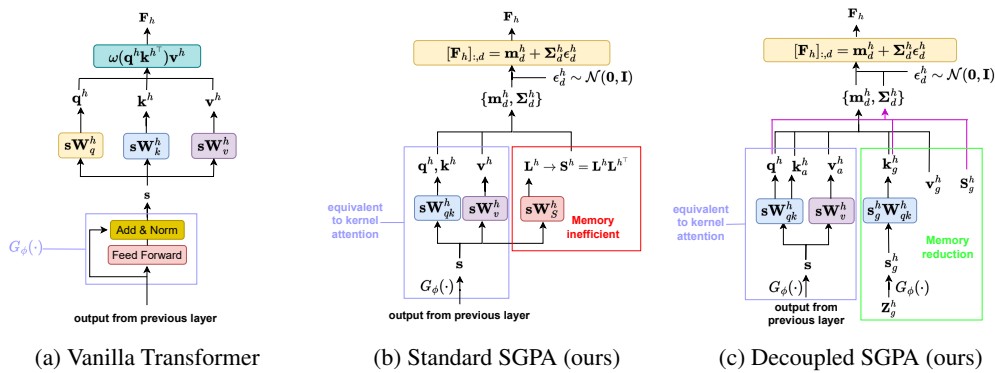

(a) Vanilla Transformer     (b) Standard SGPA (ours)     (c) Decoupled SGPA (ours)

Figure 1: Illustration of one head (h) of multi-head self attention in one layer of (a) vanilla Transformer, (b) Transformer based on standard SGPA and (c) Transformer based on decoupled SGPA.

The resulting Transformer based on our SGPA approach can be viewed as a sparse deep GP (Salimbeni & Deisenroth, 2017) with deep kernel in use for each GP layers.

- We address the computational inefficiency issues of a naive extension of SVGP to multi-head self-attention with decoupled inducing points techniques (Salimbeni et al., 2018), making SPGA scalable to deep learning tasks that Transformers are applied to.

- Empirically, on a variety of vision, NLP and graph prediction tasks and compared with baselines, SGPA-based Transformers improve considerably over in-distribution calibration, out-of-distribution (OOD) robustness, and OOD detection, while achieving competitive accuracy against Transformers with standard (Vaswani et al., 2017) or kernel attention (Tsai et al., 2019).

## 2 BACKGROUND

Attention mechanism, first introduced in Graves et al. (2013), has become the core bulding block for Transformer models. In this work, we consider Transformers using multi-head self-attention (MHSA) as in Vaswani et al. (2017); Dosovitskiy et al. (2021). Here, we briefly review MHSA and sparse variational Gaussian process, based on which our method is developed.

### 2.1 MULTI-HEAD SELF-ATTENTION (MHSA)

Given $T$ queries $\boldsymbol{q} \in \mathbb{R}^{T \times d_q}$, keys $\boldsymbol{k} \in \mathbb{R}^{T \times d_k}$ ($d_k = d_q$) and values $\mathbf{v} \in \mathbb{R}^{T \times d_v}$, dot-product attention (Vaswani et al., 2017) is computed as follows, using a nonlinear activation function $\omega$:

$$\boldsymbol{F} = \omega(\boldsymbol{q}\boldsymbol{k}^\top)\mathbf{v}. \tag{1}$$

The dot product ($\boldsymbol{q}\boldsymbol{k}^\top$) measures the similarities between the queries and keys. For self attention the keys are simply set to be equal to the queries, i.e., $\boldsymbol{k} = \boldsymbol{q}$. Transformers use multi-head self-attention (MHSA) which modifies dot-product self-attention as follows. Assume $H$ attention heads are in use. Then given $T$ inputs $\boldsymbol{s} \in \mathbb{R}^{T \times d_s}$ to the MHSA block, we first project them to the queries for each head $h$ with a projection matrix $\boldsymbol{W}_q^h \in \mathbb{R}^{d_s \times d_q}$: $\boldsymbol{q}^h = \boldsymbol{s}\boldsymbol{W}_q^h$. We obtain the keys $\boldsymbol{k}^h$ and values $\mathbf{v}^h$ accordingly by projections using matrices $\boldsymbol{W}_k^h \in \mathbb{R}^{d_s \times d_k}$ and $\boldsymbol{W}_v^h \in \mathbb{R}^{d_s \times d_v}$ respectively. Typically we use the same $d_q = d_k = d_v$ for all the heads. Then the head's output $\boldsymbol{F}_h$ is obtained by plugging $\boldsymbol{q}^h$, $\boldsymbol{k}^h$ and $\mathbf{v}^h$ in eq.(1). Lastly the attention outputs from each head is combined as follows with the output projection matrix $\boldsymbol{W}_F \in \mathbb{R}^{(Hd_v) \times (Hd_v)}$:

$$\boldsymbol{F} = \text{concat}(\boldsymbol{F}_1, \cdots, \boldsymbol{F}_H)\boldsymbol{W}_F. \tag{2}$$

In Transformers multiple layers of MHSA may be in use, where the output of the $(l-1)$th MHSA layer is further processed by a non-linear function $G_{\phi^l}$ – parameterised by an MLP – to obtain the input to the $l$th MHSA layer, i.e., $\boldsymbol{s}^l = G_{\phi^l}(\boldsymbol{F}^{l-1})$. See Figure 1a for an illustration of an MHSA block in a Transformer model (excluding the combination projection step of eq.(2)).

### 2.2 SPARSE VARIATIONAL GAUSSIAN PROCESS (SVGP) WITH DEEP KERNEL

A Gaussian process (GP) (Rasmussen & Williams, 2006) is a distribution over function $f$ with infinite-dimensional index set $\mathcal{X}$ (domain of $f$). In Bayesian inference framework, a GP prior over

$f$ is specified with a mean function (often set to zero) and a covariance function parameterized by a kernel function $K_\psi(\cdot, \cdot)$ (with hyperparameters $\psi$). Specifically, the marginal distribution of function values $\boldsymbol{f}$ evaluated on any finite number of inputs $\boldsymbol{X} = [\boldsymbol{x}_1, \cdots, \boldsymbol{x}_N]^\top, \boldsymbol{x}_n \in \mathcal{X}$ is Gaussian:

$$\text{Prior: } p(f) \sim \mathcal{GP}(0, k_\psi(\cdot, \cdot)) \quad \Rightarrow \quad p(\boldsymbol{f}|\boldsymbol{X}) = \mathcal{N}(\boldsymbol{0}, \boldsymbol{K_{XX}}), \quad [\boldsymbol{K_{XX}}]_{i,j} = K_\psi(\boldsymbol{x}_i, \boldsymbol{x}_j). \quad (3)$$

Given training data $(\boldsymbol{X}, \boldsymbol{y})$, with a Gaussian likelihood $p(\boldsymbol{y}|\boldsymbol{f}) = \mathcal{N}(\boldsymbol{f}, \sigma^2 \boldsymbol{I})$, the posterior process is also a GP, and the posterior predictive distribution of $\boldsymbol{f}^*$ evaluated at the test inputs $\boldsymbol{X}^*$ is:

$$p(\boldsymbol{f}^*|\boldsymbol{X}^*, \boldsymbol{X}, \boldsymbol{y}) = \mathcal{N}(\boldsymbol{K_{X^*X}}(\boldsymbol{K_{XX}} + \sigma^2 \boldsymbol{I})^{-1}\boldsymbol{y}, \boldsymbol{K_{X^*X^*}} - \boldsymbol{K_{X^*X}}(\boldsymbol{K_{XX}} + \sigma^2 \boldsymbol{I})^{-1}\boldsymbol{K_{XX^*}}). \quad (4)$$

Unfortunately, with non-Gaussian likelihoods (e.g., for classification) or when the number of training datapoints $N$ is large, the posterior process is intractable. Still we can approximate the posterior process with a GP, and a popular approach is sparse variational Gaussian process (SVGP) (Titsias, 2009; Hensman et al., 2013), which uses a small number of $M$ inducing points $(\boldsymbol{Z}, \boldsymbol{u}) = \{(\boldsymbol{z}_m, u_m)\}_{m=1}^M$ to summarise the training data and, to some degree, replaces the terms involving $\boldsymbol{X}, \boldsymbol{y}$ in eq.(4) with the inducing points.

A detailed introduction of SVGP is provided in Appendix B.1, in short it utilises the property of GP to augment the prior as $p(\boldsymbol{f}, \boldsymbol{u}|\boldsymbol{X}, \boldsymbol{Z})$, which is a Gaussian with zero mean and covariance matrix as a kernel matrix computed on $[\boldsymbol{X}, \boldsymbol{Z}]$, and define the approximate posterior process as:

$$\begin{aligned} p(\boldsymbol{f}^*, \boldsymbol{f}, \boldsymbol{u}|\boldsymbol{Z}, \boldsymbol{X}^*, \boldsymbol{X}, \boldsymbol{y}) &\propto p(\boldsymbol{y}|\boldsymbol{f})p(\boldsymbol{f}^*, \boldsymbol{f}|\boldsymbol{u}, \boldsymbol{Z}, \boldsymbol{X}^*, \boldsymbol{X})p(\boldsymbol{u}|\boldsymbol{Z}) \\ &\approx q(\boldsymbol{f}^*, \boldsymbol{f}, \boldsymbol{u}|\boldsymbol{Z}, \boldsymbol{X}^*, \boldsymbol{X}) := p(\boldsymbol{f}^*, \boldsymbol{f}|\boldsymbol{u}, \boldsymbol{Z}, \boldsymbol{X}^*, \boldsymbol{X})q(\boldsymbol{u}), \quad q(\boldsymbol{u}) := \mathcal{N}(\boldsymbol{m_u}, \boldsymbol{S_u}). \end{aligned} \quad (5)$$

Notice that the exact posterior and the approximate posterior share the conditional distribution $p(\boldsymbol{f}^*, \boldsymbol{f}|\boldsymbol{u}, \boldsymbol{Z}, \boldsymbol{X}^*, \boldsymbol{X})$. This simplifies the evidence lower-bound (ELBO) objective for optimising the variational parameters $\boldsymbol{m_u}, \boldsymbol{S_u}$ and the kernel hyperparameters $\psi$ to

$$\mathcal{L}_{ELBO} = E_{q(\boldsymbol{f}|\boldsymbol{X}, \boldsymbol{Z})}[\log p(\boldsymbol{y}|\boldsymbol{f})] - KL(q(\boldsymbol{u})||p(\boldsymbol{u}|\boldsymbol{Z})). \quad (6)$$

Since $q(\boldsymbol{u})$ and $p(\boldsymbol{u}|\boldsymbol{Z})$ are both Gaussian, the second term can be evaluated analytically. For non-Gaussian likelihoods, we resort to Monte-Carlo estimation for computing the first term. In prediction, the approximate posterior predictive distribution of $\boldsymbol{f}^*$ evaluated on test inputs $\boldsymbol{X}^*$ becomes:

$$\begin{aligned} q(\boldsymbol{f}^*|\boldsymbol{X}^*, \boldsymbol{Z}) &= \int p(\boldsymbol{f}^*, \boldsymbol{f}|\boldsymbol{u}, \boldsymbol{Z}, \boldsymbol{X}^*, \boldsymbol{X})q(\boldsymbol{u})d\boldsymbol{u}d\boldsymbol{f} \\ &= \mathcal{N}(\boldsymbol{K_{X^*Z}}\boldsymbol{K_{ZZ}^{-1}}m_u, \boldsymbol{K_{X^*X^*}} + \boldsymbol{K_{X^*Z}}\boldsymbol{K_{ZZ}^{-1}}(\boldsymbol{S_u} - \boldsymbol{K_{ZZ}})\boldsymbol{K_{ZZ}^{-1}}\boldsymbol{K_{ZX^*}}). \end{aligned} \quad (7)$$

Note that the computations of both the ELBO (eq.(6)) and the approximate posterior predictive distribution (eq.(7)) require matrix inversion of $\boldsymbol{K_{ZZ}}$ only. Since we usually use a small number of inducing points ($M \ll N$), the computational cost of SVGP ($O(NM^2 + M^3)$) is significantly lower than the $O(N^3)$ cost in full GP resulting from the inversion of $\boldsymbol{K_{XX}}$ (c.f. eq.(4)).

One way to take advantage of the expressiveness of DNN in GP is to parameterize the kernel function using DNN, so that the network weights become part of the hyperparameters of a deep kernel (Wilson et al., 2016). Given a regular base kernel, such as RBF kernel $K_{base}(\cdot, \cdot)$, we can first map the inputs $\boldsymbol{X}$ to a feature space using a DNN, $h_\theta(\boldsymbol{X})$, then apply the base kernel to the DNN features corresponding to the inputs: $K_{deep}(\cdot, \cdot) = K_{base}(h_\theta(\cdot), h_\theta(\cdot))$.

## 3 SPARSE GAUSSIAN PROCESS ATTENTION

We propose Sparse Gaussian Process Attention (SGPA) to perform approximate Bayesian inference for Transformer-based models. The key idea is to replace the softmax operation in scaled dot-product attention (Vaswani et al., 2017) with a kernel (Tsai et al., 2019), and connect the resulting attention to the mean of an SVGP. This insight allows us to apply SVGP equations for uncertainty estimation, and we further introduce decoupled inducing points to improve computational efficiency.

### 3.1 ATTENTION AS THE MEAN OF A SPARSE VARIATIONAL GAUSSIAN PROCESS

Standard Transformers use attention blocks based on scaled dot-product (Vaswani et al., 2017). Given queries $\boldsymbol{q}$, keys $\boldsymbol{k}$ and value $\mathbf{v}$, the scaled dot-product (SDP) attention is given as follows:

$$\text{SDP-Attention: } \boldsymbol{F} = \text{softmax}(\frac{\boldsymbol{q}\boldsymbol{k}^\top}{\sqrt{d_k}})\mathbf{v}, \quad (8)$$

where $d_k$ is the dimension of keys. Since attention involves measuring the similarity between $q$ and $k$, Tsai et al. (2019) generalised SDP-Attention by replacing $\text{softmax}(\frac{qk^\top}{\sqrt{d_k}})$ in eq.(8) with a kernel gram matrix $K_{qk}$ ($[K_{qk}]_{i,j} = K(q_i, k_j)$) computed using a valid symmetric kernel $K(\cdot, \cdot)$, for which we refer to it as underline{kernel attention} or $K$-Attention for short:

$$K\text{-Attention:} \quad F = K_{qk}\mathbf{v}. \tag{9}$$

Recall the posterior mean of SVGP in eq.(7) is $m = K_{XZ}K_{ZZ}^{-1}m_u$ when evaluated on training inputs ($X^* = X$). Now we reparameterise the variational mean parameter of SVGP as $[\mathbf{v}]_{:,d} := K_{ZZ}^{-1}m_u$ for each dimension ($d$) of $\mathbf{v}$, and define the queries and keys as the input locations and inducing point locations: $q := X, k := Z$. By doing so, underline{equivalence can be identified} between the posterior mean of an SVGP and each dimension of the output of a kernel attention block. This allows us to extend the toolbox of Gaussian processes and their scalable approximations for quantifying uncertainty in Transformers in the following sections.

## 3.2 STANDARD SGPA & ITS INEFFICIENCY FOR SELF-ATTENTION

Observing the equivalence between $K$-Attention and SVGP mean, a natural idea for uncertainty estimation is to apply SVGP techniques for approximate posterior variance computations. In detail, we introduce a set of variational covariance parameters $S \in \mathbb{R}^{T \times T \times d_v}$ (with $T$ as the number of keys/inducing inputs), and optimise them using the ELBO (eq.(6)). This procedure returns the mean and covariance for each dimension ($d$) of the posterior attention output as:

$$m_d = K_{qk}[\mathbf{v}]_{:,d}, \quad \Sigma_d = K_{qq} + K_{qk}(K_{kk}^{-1}[S]_{:,:,d}K_{kk}^{-1} - K_{kk}^{-1})K_{kq}. \tag{10}$$

In this way, we fit an SVGP for each dimension of attention outputs independently: for each dimension $d$, an SVGP given as eq.(10) is fitted using the same kernel, but with different variational mean ($[\mathbf{v}]_{:,d}$) and covariance ($[S]_{:,:,d}$) parameters. We name this approach as underline{standard SGPA} and provide a visualisation of the operations in Figure 1b.

Unfortunately, standard SGPA becomes computationally inefficient when applied to Transformers based on multi-head self-attention. In each attention layer the keys for each head $h$, $k^h$, are obtained by passing the output from previous layer through a neural network. Moreover, the projection matrices for queries and keys need to be tied (i.e., $W_q^h = W_k^h := W_{qk}^h$) to obtain a valid symmetric kernel (Tsai et al., 2019). As a result, the queries and keys in a self-attention layer are the same, more importantly they are input-dependent, i.e., $k^h = sW_k^h$ and they vary as the input sequence to the Transformer changes. Therefore, to extend standard SVGP framework (eq.(10)) to self-attention, the covariance parameters $S^h$ need to be input-dependent as well to accommodate the varying inducing inputs $k^h$. A naive idea would parameterise $S^h$ by linear projection, e.g., $vec(L^h) = sW_s^h$ for one head where $L^h$ is the Cholesky factor of $S^h$ (see Figure 1b). This will incur a memory cost of $O(T^2)$ per head even if we tie the variational covariances across output dimensions, and a run-time cost of $O(T^3)$ per head underline{per input sequence} for inverting $K_{k^h k^h}$ as $k^h$ is input-dependent. Therefore standard SGPA is both memory and run-time inefficient especially for long input sequences.

## 3.3 IMPROVING TIME & MEMORY EFFICIENCIES VIA DECOUPLED SGPA

We propose to address the aforementioned inefficiency issues by extending the orthogonally decoupled sparse Gaussian process approximation (Salimbeni et al., 2018) to self-attention. In addition to input-dependent (or "amortised") keys/inducing inputs $k^h$, which we will call $k_a^h$ from now on, for each head $h$, we also incorporate another $M_g$ number of "global" keys/inducing inputs $k_g^h$ that are shared across all input sequences. The main idea is to compute the variance of sparse GP using the global keys only, so that the variational parameters for the $S^h$ matrix become independent to the input sequences. Indeed, following the derivations presented in Appendix B.2, we can compute the mean and covariance for each output dimension ($d$) of each head as (we drop the superscript $h$ here for more concise notation):

$$
\begin{aligned}
m_d &= K_{qk_a}[\mathbf{v}_a]_{:,d} - K_{qk_g}K_{k_g k_a}[\mathbf{v}_a]_{:,d} + K_{qk_g}[\mathbf{v}_g]_{:,d}, \\
\Sigma_d &= K_{qq} + K_{qk_g}K_{k_g k_g}^{-1}([S_g]_{:,:,d} - K_{k_g k_g})K_{k_g k_g}^{-1}K_{k_g q},
\end{aligned} \tag{11}
$$

where $\mathbf{v}_g \in \mathbb{R}^{M_g \times d_v}, S_g \in \mathbb{R}^{M_g \times M_g \times d_v}$ are the variational parameters associated with the global keys $k_g$, and $\mathbf{v}_a \in \mathbb{R}^{T \times d_v}$ is computed via projection $\mathbf{v}_a = sW_v$. We name this approach as underline{decoupled SPGA} which is illustrated in Figure 1c.

Compared to standard SGPA (eq.(10), where $\boldsymbol{k}_a^h$ in decoupled SGPA is the same as $\boldsymbol{k}^h$ in standard SGPA), we see that the posterior mean of decoupled SGPA also involves two extra terms to take into account the effect of global inducing points. But more importantly, the posterior variance of the two SGPA methods differ only in the keys/inducing inputs in use (input-dependent keys $\boldsymbol{k}^h$ versus global keys $\boldsymbol{k}_g^h$), and this brings in the key advantage of decoupled SGPA. As the posterior covariance in eq.(11) only involves the global inducing points, the variational covariance no longer needs to be input-dependent, and (the Cholesky factor of) $\boldsymbol{S}_g^h$ can be parameterised freely. Now the number of parameters for the covariance part is of order of $O(M_g^2)$ (vs $O(T^2)$ in standard SPGA), and the computation of matrix inversion pays a one-off cost of $O(M_g^3)$ (vs $O(T^3)$ for every input sequence). Notice that we are free to choose the number of global inducing points $M_g$, and in practice we find $M_g = O(\frac{T_{avg}}{H})$ is usually sufficient, where $T_{avg}$ is the average length of training input sequences. In Table 1, we summarise time complexity (with batch size $B$) and the additional memory (number of parameters) required for SGPA in one head of a Transformer. We also include maximum likelihood estimation (MLE) for reference (note that memory complexity for MLE does not depend on input sequence length $T$).

As the time and memory savings are significant, we mainly evaluate decoupled SGPA in our experiments, and in the rest of the main text we will refer to decoupled SGPA as SGPA for short.

Table 1: Complexity comparison for standard and decoupled SGPA.

| Model | Time | Additional Memory |
|---|---|---|
| MLE | $O(BT^2)$ | - |
| Standard SGPA | $O(BT^3)$ | $O(T^2)$ |
| Decoupled SGPA | $O(BT^2 M_g + M_g^3)$ | $O(M_g^2)$ |

### 3.4 Transformer based on decoupled SGPA

So far we have presented SGPA methods for uncertainty quantification in a multi-head self-attention module. When applied to Transformer models, multiple layers of attention blocks are in use, and in the following we describe the construction of a Transformer model based on decoupled SGPA. Note that, as SGPA is equivalent to a sparse GP, the Transformer model presented below can be viewed as a sparse approximation to a deep GP (Damianou & Lawrence, 2013; Salimbeni & Deisenroth, 2017) with deep kernel in each layer.

Our Transformer architecture mostly follows the one in Vaswani et al. (2017). The input to the $l$th SGPA layer is the output from previous SGPA layer $\boldsymbol{F}^{l-1} \in \mathbb{R}^{T \times d^{l-1}}$. We first process the input with a non-linear mapping $G_{\phi^l} : \mathbb{R}^{d^{l-1}} \to \mathbb{R}^{d^l}$, and then perform projections to obtain the queries, amortised & global keys and values. Specifically, we have for each head $h$:

$$\boldsymbol{q}^{l,h} = \boldsymbol{k}_a^{l,h} = G_{\phi^l}(\boldsymbol{F}^{l-1})\boldsymbol{W}_{qk}^{l,h}, \quad \boldsymbol{k}_g^{l,h} = G_{\phi^l}(\boldsymbol{Z}_g^{l,h})\boldsymbol{W}_{qk}^{l,h}, \quad \boldsymbol{v}_a^{l,h} = G_{\phi^l}(\boldsymbol{F}^{l-1})\boldsymbol{W}_v^{l,h}, \quad (12)$$

where $\boldsymbol{Z}_g^{l,h} \in \mathbb{R}^{M_g \times d^{l-1}}$ are global inducing locations of the $l$th layer defined on the same space as $\boldsymbol{F}^{l-1}$. Then we apply a base kernel $K_{base}(\cdot, \cdot)$ to compute the kernel matrices. This is equivalent to using a deep kernel defined on the space of $\boldsymbol{F}^{l-1}$, and the parameters of $G_{\phi^l}$ are viewed as the hyperparameters of the deep kernel. Lastly, with variational parameters $(\mathbf{v}_g^{l,h}, \boldsymbol{S}_g^{l,h})$ associated with the global inducing locations $\boldsymbol{Z}_g^{l,h}$, we can obtain $\boldsymbol{m}_d^{l,h}$ and $\boldsymbol{\Sigma}_d^{l,h}$ using equation 11. We then propagate uncertainty to the next layer by generating samples of output for each head using the reparameterization trick as in Salimbeni & Deisenroth (2017):

$$[\boldsymbol{F}_h^l]_{:,d} = \boldsymbol{m}_d^{l,h} + \boldsymbol{\Sigma}_d^{l,h}\boldsymbol{\epsilon}_d^{l,h}, \quad \boldsymbol{\epsilon}_d^{l,h} \sim \mathcal{N}(\boldsymbol{0}, \boldsymbol{I}). \quad (13)$$

The final output $\boldsymbol{F}^l \in \mathbb{R}^{T \times d^l}$ of this SGPA layer is obtained by linear combination in the same way as in standard Transformers (see eq.(2)).

The ELBO objective for training the variational & kernel parameters is derived following deep GP and additive GP (Duvenaud et al., 2011) approaches. The key idea is that as each head in MHSA with SGPA is a (sparse) GP, the final output $\boldsymbol{F}^l$ can also be viewed as a weighted summation of (sparse) GPs, which is again a GP (Duvenaud et al., 2011). This allows us to perform variational approximations on each of the heads before the final combination instead of using a direct approximation on the $\boldsymbol{F}^l$ process (Sun et al., 2021). Assuming the approximate posterior $q$ for $\{\boldsymbol{F}_h^l\}_{h=1}^H$ factorises over $h$, the corresponding ELBO with input sequence $\boldsymbol{F}^0 := \boldsymbol{X}$ is (derivations in Appendix B.3):

$$\mathcal{L}_{ELBO} = E_{q(\boldsymbol{F}^L|\boldsymbol{F}^0,\{\boldsymbol{k}_g^{l,h}\}_{l=1,h=1}^{L,H})}[\log p(\boldsymbol{Y}|\boldsymbol{F}^L)]$$
$$- \sum_{l=1}^{L}\sum_{h=1}^{H} E_{q(\boldsymbol{F}^l|\boldsymbol{F}^0,\{\boldsymbol{k}_g^{j,h}\}_{j=1,h=1}^{l,H}))}[KL(q(\boldsymbol{u}_{a\cup g}^{l,h}|\boldsymbol{k}_g^{l,h},\boldsymbol{F}^{l-1})||p(\boldsymbol{u}_{a\cup g}^{l,h}|\boldsymbol{k}_g^{l,h},\boldsymbol{F}^{l-1}))]. \tag{14}$$

In practice, we resort to Monte-Carlo to estimate $\mathcal{L}_{ELBO}$ with samples of function values generated iteratively passing through each layer using the reparameterization trick (eq.(13)).

## 4 EXPERIMENTS

We evaluate SGPA on prediction tasks across modalities, with the following experimental set-up.

- Datasets: CIFAR10 & CIFAR100 (image classification (Krizhevsky et al., 2009), CV tasks); CoLA (linguistic acceptability prediction (Warstadt et al., 2019), NLP task) and IMDB (sentiment analysis, (Maas et al., 2011), NLP task).

- Network architectures: We use Vision Transformers (ViT (Dosovitskiy et al., 2021)) for CV tasks. For kernel attention we use the exponential kernel (Tsai et al., 2019) and the ARD-RBF kernel (Rasmussen & Williams, 2006) for NLP and CV tasks respectively. Scaled dot-product (SDP) attention based Transformers are also evaluated. As in Tsai et al. (2019), we find kernel attention tends to outperform SDP attention in most tasks considered, thus we do not include the results of SDP attention in the main text. These results can be found in the tables in Appendix G.

- Baselines: We compare our approach with the following "single-model" methods: maximum likelihood estimation (MLE), Bayesian inference methods including mean-field variational inference (MFVI, (Blundell et al., 2015)), Monte-Carlo Dropout (MCD, (Gal & Ghahramani, 2016)), Kronecker-factored last layer Laplace approximation (KFLLLA) (Kristiadi et al., 2020), and Spectral-normalized Neural Gaussian Process (SNGP) (Liu et al., 2020). For tasks where a validation set is used, we also consider temperature scaling (TS) (Guo et al., 2017a) and use the validation set as the calibration set. For CV tasks, we also consider ensemble methods: we compare SGPA ensemble (SGPAE) with deep ensemble (DE) (Lakshminarayanan et al., 2017). We don't consider ensemble models in NLP tasks since we use different train-(valid)-test splits in different runs for them.

- Evaluations & metrics: We consider three evaluation set-ups: in-distribution performance, out-of-distribution (OOD) robustness and OOD detection. The metrics on test set include predictive accuracy metrics for each task, uncertainty calibration metrics such as negative predictive log-likelihood (NLL), expected calibration error (ECE) and maximum calibration error (MCE) (Guo et al., 2017b). We report the mean±two standard errors for each metric obtained from 5 independent runs. For OOD detection tasks we consider the area under the ROC & precision-recall curves (AUROC & AUPR, respectively), and we report the average ranks in terms of AUROC and AUPR over all of the 6 OOD detection tasks for each method.

For fair comparisons, within each task, all the models are trained using the same architecture and optimisation setting. All the models are trained from scratch without pre-training. We include the experimental details in Appendix E. Results in tables are also presented in Appendix G.

### 4.1 IN-DISTRIBUTION CALIBRATION

We report the evaluation results for in-distribution test data on image classification (CIFAR10 & CIFAR100, without data augmentation), sentiment analysis (IMDB), and linguistic acceptability (CoLA) tasks in the first, second, third and fourth row of Figure 2 respectively. Here for the CoLA dataset, predictive accuracy is measured by Matthew correlation coefficient (MCC) (Matthews, 1975) instead of accuracy, as in Warstadt et al. (2019).

All "single-model" calibration methods considered tend to improve the calibration, except for sentiment analysis, where KFLLLA fails in the sense that it achieves worse calibration even than MLE (although KFLLLA achieves best calibration for linguistic acceptability (CoLA), its performance is unstable across tasks). Although MFVI tends to achieve the lowest calibration errors, it severely underfits the data in all the experiments. This is undesirable, as improvement in calibration should not come at a price of noticeable drop in predictive correctness. As a counter example, one can achieve

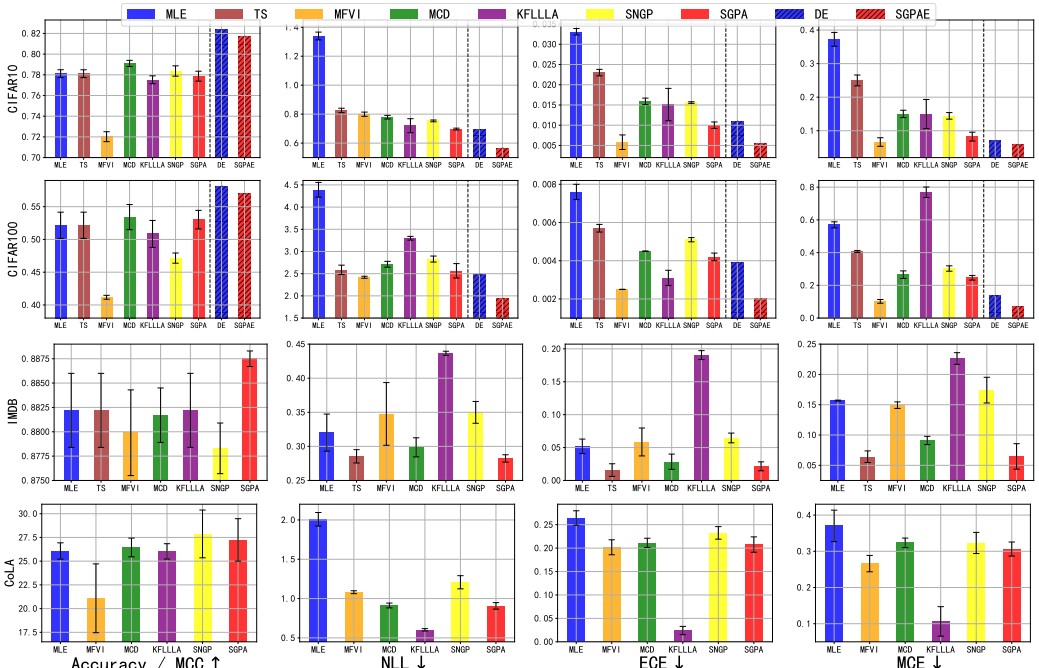

Figure 2: Test set accuracy (or MCC for CoLA) & calibration metrics of Transformers or ViTs trained on CIFAR10 (1st row), CIFAR100 (2nd row), IMDB (3rd row) and CoLA (4th row).

perfect calibration in terms of zero ECE by predicting marginal class probability, but this prediction is useless in practice. For image classification on CIFAR100, KFLLLA achieves lower ECE than SGPA, however, it achieves worse NLL and the worst MCE among all the methods. Overall, SGPA achieves the best performance when compared with the other "single-model" baselines: it consistently achieves better calibration across all tasks while maintaining competitive (or even better, on IMDB) predictive accuracy. Compared with "single-model" methods, both ensemble methods, DE and SGPAE, achieve much better predictive accuracy. SGPAE noticeably outperforms DE in terms of calibration while maintaining competitive predictive accuracy.

## 4.2 ROBUST PREDICTION ON OUT-OF-DISTRIBUTION DATA

Next we evaluate the performance of SGPA under distribution shift for both the linguistic acceptability task (CoLA) and the image classification tasks (CIFAR10 & CIFAR100). The OOD data for CoLA is introduced by the same authors of Warstadt et al. (2019), while for the CIFAR datasets, we use corrupted CIFAR datasets (CIFAR10-C and CIFAR100-C) (Hendrycks & Dietterich, 2019) as the OOD data, which contains noisy CIFAR images with different types of distortions introduced to their clean counterparts at different skew intensities. Note that we don't consider TS for OOD tasks in this and the next subsection since as a Frequentist method, it is proposed to calibrate the uncertainty on in-distribution data only.

We report in Figure 3 the MCC and calibration metrics for the OOD test on CoLA. The observations are similar with the in-distribution test: SGPA outperforms MLE, MCD, and SNGP in terms of NLL and calibration errors while achieving improved accuracy. MFVI and KFLLLA achieve lower

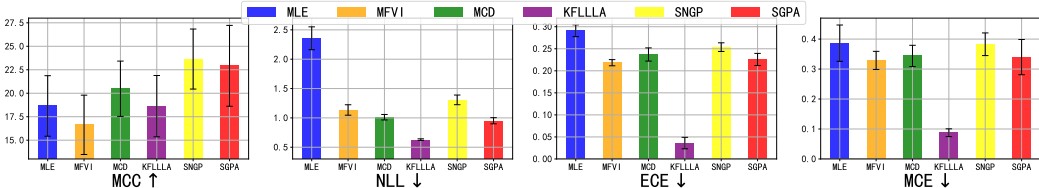

Figure 3: Test set MCC & calibration metrics for OOD test set of Transformers trained on COLA.

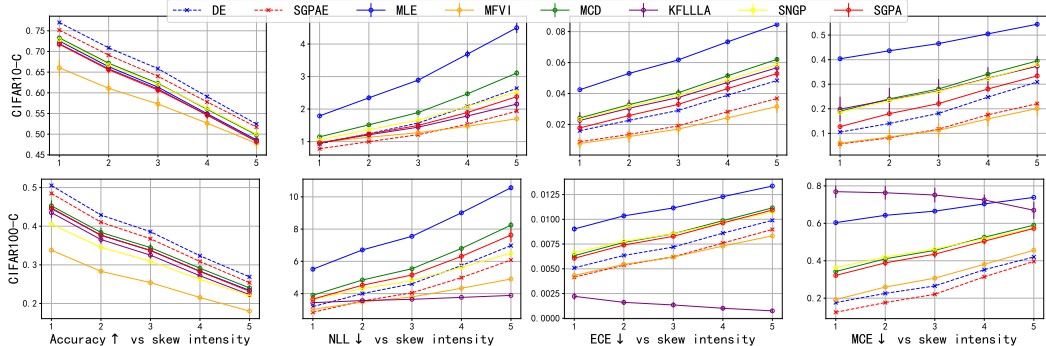

Figure 4: Test set accuracy & calibration metrics on CIFAR10-C (top row) and CIFAR100-C (bottom row) against skew intensity of corruption for ViTs trained on corresponding clean data.

calibration errors but they achieve worse predictive accuracy than SGPA. In particular, MFVI again underfits the data.

For OOD robustness test on image classification, we compute metrics against skew intensity on the corrupted CIFAR datasets and report them in Figure 4. Again the same story: among "single-model" methods, SGPA outperforms MLE, MCD and SNGP in terms of calibration without hurting accuracy. MFVI achieves lower calibration errors than SGPA but it pays the price of underfitting especially when the skew intensity is small. The performance of KFLLLA seems to be not as stable as SGPA. For CIFAR10-C, SGPA achieves better calibration than KFLLLA. For CIFAR100-C, KFLLA achieves the best NLL and ECE but the worst MCE. Ensemble methods again achieve better accuracy than "single-model" methods and SGPAE still outperforms DE in terms of calibration while achieving similar accuracy.

### 4.3 OUT-OF-DISTRIBUTION DETECTION

Lastly we consider OOD detection tasks on Transformer models trained for image classification, to further evaluate the quality of uncertainty estimates. Here we use predictive entropy to score each input (from either in-distribution or OOD data) and make decisions of "in/out" if the entropy is smaller/greater than a specified threshold. Using different thresholds for this detection task allows us to compute both the receiver operator characteristic (ROC) and the precision-recall (PR) curves, and we use the area under the curves, i.e., AUROC and AUPR, for performance evaluations. For each of the two CIFAR datasets, we consider the other CIFAR dataset, SVHN and mini-ImageNet as OOD datasets so that we construct 6 OOD detection tasks in total. For each method, here we report its average ranks in terms of AUROC and AUPR over 6 tasks in Table 2. Ensemble methods outperform "single-model" methods with SGPAE achieving the best performance. Among "single-model" methods, apart from KFLLLA which achieves the best performance, SGPA outperforms all the other baselines. For comparison within each task, see Figure 5 and 6 in Appendix A where the values of AUROC and AUPR for all methods are plotted for each task.

Table 2: Average ranks of different methods in terms of AUROC and AUPR over 6 OOD detection tasks

| Model | avg. rank (AUROC) ↓ | avg. rank (AUPR) ↓ |
|---|---|---|
| MLE | 6.1667 | 5.5000 |
| MFVI | 7.0000 | 7.1667 |
| MCD | 5.6667 | 5.6667 |
| KFLLLA | 4.3333 | 4.5000 |
| SNGP | 5.3333 | 5.3333 |
| SGPA | 4.5000 | 4.6667 |
| DE | 1.6667 | 2.0000 |
| SGPAE | **1.3333** | **1.1667** |

### 4.4 SUMMARY OF ADDITIONAL RESULTS

Additional experiments are reported in Appendix C where we summarise the main results as follows.

- In appendix C.1, we find that in addition to the parameter inefficiency problem, Transformers based on standard SGPA also suffer from underfitting. Compared with decoupled SGPA, standard SGPA achieves significantly worse accuracy on CIFAR10 classification task.

- In appendix C.2, we report results of image classification with data augmentation. While MFVI and SNGP underfit the data, both the accuracy and calibration improve for the other methods. The performance difference between SGPA and other "single-model" baselines (except MFVI and SNGP) becomes smaller as a result of strong regularisations from data augmentation. SG-PAE again achieves the best performance.

- In appendix C.3, we report results of graph property regression with ZINC dataset (Dwivedi et al., 2020). For this task, SGPAE achieves the best performance and SGPA outperforms the other "single-model" baselines.

## 5 RELATED WORK

**Bayesian Transformers.** Tran et al. (2019) and Xue et al. (2021) propose to perform approximate posterior inference using MFVI in weight space for a subset of layers in Transformers. However, in our experiments we find this type of approaches underfits the data. This pathology of underfitting is theoretically confirmed for weight space MFVI (Foong et al., 2020; Coker et al., 2022). Another line of research proposes to perform VI over the attention matrices directly (Fan et al., 2020; Cinquin et al., 2021). However, Fan et al. (2020) only considers finetuning with variational attention, and (Cinquin et al., 2021) only considers experiments on synthetic or simple datasets with shallow networks and the variational distribution fitted over the attention weights are shared across data, which might be too restrictive for complex problems. Moreover, they find that a data-dependent variational distribution over attention weights can even hurt the performance of their approaches. Liu et al. (2020) consider performing Bayesian inference directly over the Transformer output by fitting a GP over the last layer output (Bradshaw et al., 2017). This approach can be viewed as using a GP model with a deep kernel defined by the Transformer. Instead, SGPA fits a deep GP so that uncertainty is propagated through each attention layer of the Transformer. In addition, Liu et al. (2020) propose to preserve the distance awareness property for the deep kernel. Note that this distance-preserving trick is orthogonal to ours and can also be easily integrated into SGPA.

**Related GP methods.** The ELBO of SGPA is similar to that in Sun et al. (2021) which also propose to independently approximate the posterior for each additive component in an additive GP (Duvenaud et al., 2011). The difference is in the kernel design: Sun et al. (2021) aim to decompose a given kernel function into orthogonal "kernel basis", while in SGPA we consider the same type of kernel for each attention head but with different kernel hyperparameters. Our approach is also related to sparse within sparse Gaussian process (SWSGP) (Tran et al., 2021; Jafrasteh et al., 2022) which allows adaptive inducing points for each data point (similar to the input-dependent keys $k_a$ in SGPA). This connection between SGPA and SWSGP is further discussed in appendix D.

## 6 CONCLUSION AND FUTURE WORK

We have proposed SGPA to directly perform approximate Bayesian inference over the output of attention blocks in Transformers. Compared with other baselines, we showed Transformers based on SGPA achieve better balance between predictive accuracy and calibration. Furthermore, the improved quality of uncertainty estimation provided by SGPA has been proved useful in maintaining robustness under distribution shift and in out of distribution detection. Future work will investigate the following directions. First, masked pre-training (Delvin et al., 2019), which has been proved crucial for downstream tasks for standard Transformers, may also improve the performance of Transformers based on SGPA. In this work, we are not able to consider pretraining due to the high computational cost, and since SGPA replaces scaled dot-product with a valid kernel, there is no existing pre-trained backbone that can be directly used for the downstream fine-tuning tasks. Second, many tasks using Transformers, such as neural machine translation, require autoregressive prediction using an encoder-decoder based architecture (Vaswani et al., 2017; Brown et al., 2020), and we will adapt SGPA to the decoder as well. Lastly, we will investigate the introduction of hidden mapping distance preserving trick (Liu et al., 2020) to SGPA, which has been shown to be useful in regularizing the parameters in deep kernel learning.

## ACKNOWLEDGMENTS

We would like to thank Harrison B. Zhu and Zijing Ou at Imperial College London for their valuable comments on the manuscript. We also would like to thank the anonymous reviewers: their valuable feedback and helpful discussions during the rebuttal helped improve the paper.

## ETHICS STATEMENT

We believe that this work in its current status has minimal ethical implications: this research involves no human subjects and no data or domain where privacy/discrimination/fairness is a concern. However, as with most research in machine learning, new methods could be used on datasets and domains where privacy/discrimination/fairness is concerning to cause negative ethical impacts, but we do not think SGPA is more concerning than any other method in this regard.

## REPRODUCIBILITY STATEMENT

Example code can be found at: `https://github.com/chenw20/SGPA`. Details for the experimental set-up can be found in Appendix E

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

# A AUROC AND AUPR FOR OUT-OF-DISTRIBUTION DETECTION TASKS

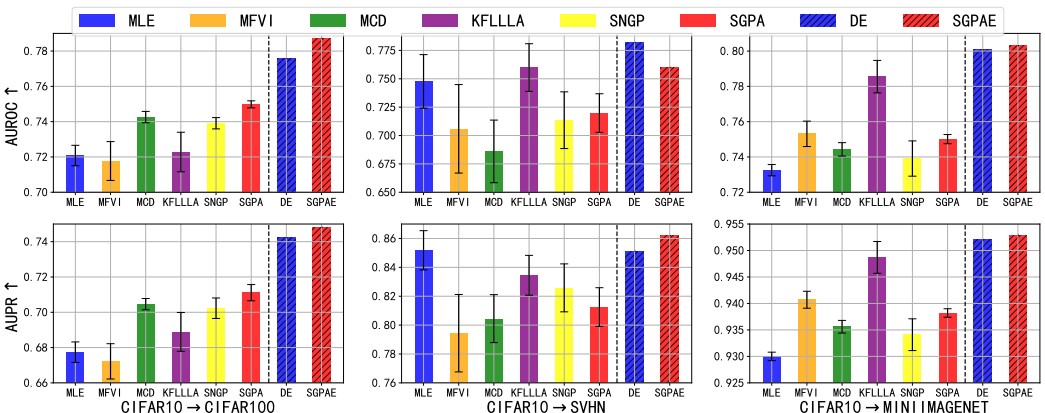

Figure 5: AUROC (top) and AUPR (bottom) for OOD detection using ViTs trained on CIFAR10.

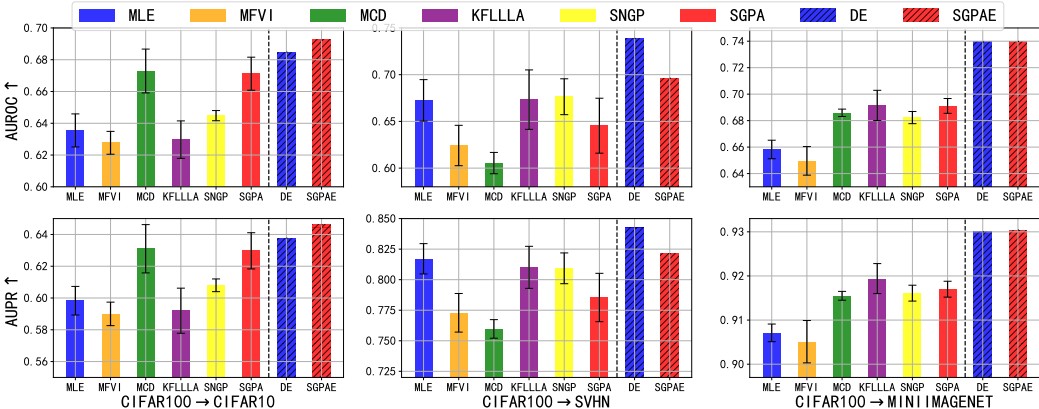

Figure 6: AUROC (top) and AUPR (bottom) for OOD detection using ViTs trained on CIFAR100.

# B DERIVATIONS

## B.1 ELBO DERIVATIONS FOR SVGP

Here we review the derivation of ELBO for standard SVGP (Titsias, 2009; Hensman et al., 2013). With $M$ inducing points pairs $\{(\boldsymbol{z}_m, u_m)\}_{m=1}^{M}$, the prior distribution of $[\boldsymbol{f}, \boldsymbol{u}]^{\top}$ is:

$$p(\boldsymbol{f}, \boldsymbol{u}|\boldsymbol{X}, \boldsymbol{Z}) = \mathcal{N}(\boldsymbol{0}, \begin{pmatrix} \boldsymbol{K}_{\mathbf{XX}} & \boldsymbol{K}_{\mathbf{XZ}} \\ \boldsymbol{K}_{\mathbf{ZX}} & \boldsymbol{K}_{\mathbf{ZZ}} \end{pmatrix}). \tag{15}$$

With prior conditional matching assumption (see eq.(5)), the approximate posterior conditional distribution of function values $\boldsymbol{f}$ for training inputs $\boldsymbol{X}$ given inducing points $\boldsymbol{u}$ is the same as the prior conditional distribution:

$$q(\boldsymbol{f}|\boldsymbol{u}, \boldsymbol{Z}, \boldsymbol{X}) = p(\boldsymbol{f}|\boldsymbol{u}, \boldsymbol{Z}, \boldsymbol{X}). \tag{16}$$

Under prior conditional matching assumption, $q(\boldsymbol{f}, \boldsymbol{u}|\boldsymbol{Z}, \boldsymbol{X}) = p(\boldsymbol{f}|\boldsymbol{u}, \boldsymbol{Z}, \boldsymbol{X})q(\boldsymbol{u})$, where $q(\boldsymbol{u}) = \mathcal{N}(\boldsymbol{m_u}, \boldsymbol{S_u})$. Suppose the observation likelihood is $p(\boldsymbol{y}|\boldsymbol{f})$, ELBO can be simplified as follows:

$$
\begin{aligned}
\mathcal{L}_{ELBO} &= E_{q(\boldsymbol{f}, \boldsymbol{u}|\boldsymbol{Z}, \boldsymbol{X})}[\log \frac{p(\boldsymbol{y}, \boldsymbol{f}, \boldsymbol{u}|\boldsymbol{Z}, \boldsymbol{X})}{q(\boldsymbol{f}, \boldsymbol{u}|\boldsymbol{Z}, \boldsymbol{X})}] \\
&= E_{q(\boldsymbol{f}, \boldsymbol{u}|\boldsymbol{Z}, \boldsymbol{X})}[\log \frac{p(\boldsymbol{y}|\boldsymbol{f})\cancel{p(\boldsymbol{f}|\boldsymbol{u}, \boldsymbol{Z}, \boldsymbol{X})}p(\boldsymbol{u}|\boldsymbol{Z})}{\cancel{p(\boldsymbol{f}|\boldsymbol{u}, \boldsymbol{Z}, \boldsymbol{X})}q(\boldsymbol{u})}] \\
&= \int (\int p(\boldsymbol{f}|\boldsymbol{u}, \boldsymbol{Z}, \boldsymbol{X})q(\boldsymbol{u})d\boldsymbol{u}) \log p(\boldsymbol{y}|\boldsymbol{f})d\boldsymbol{f} + \int q(\boldsymbol{u}) \log \frac{p(\boldsymbol{u}|\boldsymbol{Z})}{q(\boldsymbol{u})} \cancel{\int p(\boldsymbol{f}|\boldsymbol{u}, \boldsymbol{Z}, \boldsymbol{X})d\boldsymbol{f}} d\boldsymbol{u} \\
&= E_{q(\boldsymbol{f}|\boldsymbol{X}, \boldsymbol{Z})}[\log p(\boldsymbol{y}|\boldsymbol{f})] - KL(q(\boldsymbol{u})||p(\boldsymbol{u}|\boldsymbol{Z})).
\end{aligned}
\tag{17}
$$

Here $q(\boldsymbol{f}|\boldsymbol{X}, \boldsymbol{Z}) = \int p(\boldsymbol{f}|\boldsymbol{u}, \boldsymbol{Z}, \boldsymbol{X})q(\boldsymbol{u})d\boldsymbol{u}$ is a Gaussian and is given as:

$$
q(\boldsymbol{f}|\boldsymbol{X}, \boldsymbol{Z}) = \mathcal{N}(\boldsymbol{K_{XZ}}\boldsymbol{K_{ZZ}^{-1}}\boldsymbol{m_u}, \boldsymbol{K_{XX}} + \boldsymbol{K_{XZ}}\boldsymbol{K_{ZZ}^{-1}}(\boldsymbol{S_u} - \boldsymbol{K_{ZZ}})\boldsymbol{K_{ZZ}^{-1}}\boldsymbol{K_{ZX}}).
\tag{18}
$$

With Gaussian likelihood, the first term in ELBO can be evaluated analytically. Otherwise we estimate the first term using Monte-Carlo samples $\boldsymbol{f} \sim q(\boldsymbol{f}|\boldsymbol{X}, \boldsymbol{Z})$. The second term is a KL-divergence between two Gaussian distributions. Thus, it admits a closed form:

$$
KL(q(\boldsymbol{u})||p(\boldsymbol{u}|\boldsymbol{Z})) = \frac{1}{2}[Tr(\boldsymbol{K_{ZZ}^{-1}}\boldsymbol{S_u}) + \boldsymbol{m_u^\top}\boldsymbol{K_{ZZ}^{-1}}\boldsymbol{m_u} + \log \frac{|\boldsymbol{K_{ZZ}}|}{|\boldsymbol{S_u}|} - M].
\tag{19}
$$

In standard SGPA the ELBO objective remains almost the same, except that as the variational mean is reparameterised to $\mathbf{v} := \boldsymbol{K_{ZZ}^{-1}}\boldsymbol{m_u}$, the mean of $q(\boldsymbol{f}|\boldsymbol{X}, \boldsymbol{Z})$ becomes $\boldsymbol{K_{XZ}}\mathbf{v}$, and the quadratic term in $KL(q(\boldsymbol{u})||p(\boldsymbol{u}|\boldsymbol{Z}))$ becomes $\mathbf{v}^\top \boldsymbol{K_{ZZ}}\mathbf{v}$.

## B.2 ORTHOGONALLY DECOUPLED SVGP

The orthogonally decoupled SVGP (Salimbeni et al., 2018) can be interpreted as an SVGP (Titsias, 2009) with a structured variational distribution for the inducing points. Two sets of inducing points are in use: $\{(\boldsymbol{z}_a^{(m)}, u_a^{(m)})\}_{m=1}^{M_a}$ and $\{(\boldsymbol{z}_g^{(m)}, u_g^{(m)})\}_{m=1}^{M_g}$. Consider a structured Gaussian variational distribution over $\boldsymbol{u} := \boldsymbol{u}_{a\cup g} = \boldsymbol{u}_a \cup \boldsymbol{u}_g$, with variational mean and covariance given as follows:

$$
\begin{aligned}
\boldsymbol{m_u} &= \begin{pmatrix} \boldsymbol{K_{Z_a Z_g}}\boldsymbol{K_{Z_g Z_g}^{-1}}\boldsymbol{m_g} + \boldsymbol{m_a} \\ \boldsymbol{m_g} \end{pmatrix}, \\
\boldsymbol{S_u} &= \begin{pmatrix} \boldsymbol{K_{Z_a Z_a}}\boldsymbol{K_{Z_a Z_g}}\boldsymbol{K_{Z_g Z_g}^{-1}}(\boldsymbol{S_g} - \boldsymbol{K_{Z_g Z_g}})\boldsymbol{K_{Z_g Z_g}^{-1}}\boldsymbol{K_{Z_g Z_a}} & \boldsymbol{K_{Z_a Z_g}}\boldsymbol{K_{Z_g Z_g}^{-1}}\boldsymbol{S_g} \\ \boldsymbol{S_g}\boldsymbol{K_{Z_g Z_g}^{-1}}\boldsymbol{K_{Z_g Z_a}} & \boldsymbol{S_g} \end{pmatrix}.
\end{aligned}
\tag{20}
$$

Plugging the above $\boldsymbol{m_u}$ and $\boldsymbol{S_u}$ in eq.(18), we can obtain the posterior distribution of orthogonally decoupled SVGP for $\boldsymbol{f}$ after canceling some terms, which is a Gaussian with mean $\boldsymbol{m_f}$ and covariance $\boldsymbol{\Sigma_{ff}}$ given as:

$$
\begin{aligned}
\boldsymbol{m_f} &= (\boldsymbol{K_{XZ_a}} - \boldsymbol{K_{XZ_g}}\boldsymbol{K_{Z_g Z_a}})(\boldsymbol{K_{Z_a Z_a}} - \boldsymbol{K_{Z_a Z_g}}\boldsymbol{K_{Z_g Z_g}^{-1}}\boldsymbol{K_{Z_g Z_a}})^{-1}\mathbf{m}_a + \boldsymbol{K_{XZ_g}}\boldsymbol{K_{Z_g Z_g}^{-1}}\mathbf{m}_g, \\
\boldsymbol{\Sigma_{ff}} &= \boldsymbol{K_{XX}} + \boldsymbol{K_{XZ_g}}\boldsymbol{K_{Z_g Z_g}^{-1}}(\boldsymbol{S_g} - \boldsymbol{K_{Z_g Z_g}})\boldsymbol{K_{Z_g Z_g}^{-1}}\boldsymbol{K_{Z_g X}}.
\end{aligned}
\tag{21}
$$

If we further reparameterize $(\boldsymbol{K_{Z_a Z_a}} - \boldsymbol{K_{Z_a Z_g}}\boldsymbol{K_{Z_g Z_g}^{-1}}\boldsymbol{K_{Z_g Z_a}})^{-1}\mathbf{m}_a$ as $\mathbf{v}_a$ and $\boldsymbol{K_{Z_g Z_g}^{-1}}\mathbf{m}_g$ as $\mathbf{v}_g$, then we arrive at the final expressions used in decoupled SGPA (see eq.(11)):

$$
\begin{aligned}
\boldsymbol{m_f} &= \boldsymbol{K_{XZ_a}}\mathbf{v}_a - \boldsymbol{K_{XZ_g}}\boldsymbol{K_{Z_g Z_a}}\mathbf{v}_a + \boldsymbol{K_{XZ_g}}\mathbf{v}_g, \\
\boldsymbol{\Sigma_{ff}} &= \boldsymbol{K_{XX}} + \boldsymbol{K_{XZ_g}}\boldsymbol{K_{Z_g Z_g}^{-1}}(\boldsymbol{S_g} - \boldsymbol{K_{Z_g Z_g}})\boldsymbol{K_{Z_g Z_g}^{-1}}\boldsymbol{K_{Z_g X}}.
\end{aligned}
\tag{22}
$$

## B.3 ELBO FOR TRANSFORMERS BASED ON SGPA

An $L$-layer Transformer based on SGPA is a deep GP (Damianou & Lawrence, 2013), and we train it using the doubly stochastic variational inference framework (Salimbeni & Deisenroth, 2017). For

each input sequence $\boldsymbol{F}^0 := \boldsymbol{X}$, the joint distribution for $\boldsymbol{Y}, \{\boldsymbol{F}^l\}_{l=1}^L, \{\boldsymbol{u}_{a\cup g}^{l,h}\}_{l=1,h=1}^{L,H}$ is:

$$
p(\boldsymbol{Y}, \{\boldsymbol{F}^l\}_{l=1}^L, \{\boldsymbol{u}_{a\cup g}^{l,h}\}_{l=1,h=1}^{L,H}|\boldsymbol{F}^0) =
$$
$$
p(\boldsymbol{Y}|\mathbf{F}^L)[\prod_{l=1}^L p(\mathbf{F}^l|\{\boldsymbol{u}_{a\cup g}^{l,h}\}_{h=1}^H, \boldsymbol{F}^{l-1})p(\{\boldsymbol{u}_{a\cup g}^{l,h}\}_{h=1}^H|\{\boldsymbol{k}_g^{l,h}\}_{h=1}^H, \boldsymbol{F}^{l-1})], \tag{23}
$$

where $p(\{\boldsymbol{u}_{a\cup g}^{l,h}\}_{h=1}^H|\{\boldsymbol{k}_g^{l,h}\}_{h=1}^H, \boldsymbol{F}^{l-1}) = \prod_{h=1}^H p(\boldsymbol{u}_{a\cup g}^{l,h}|\boldsymbol{k}_g^{l,h}, \boldsymbol{F}^{l-1})$ since we assume the prior for inducing points factorizes across heads in each layer. Note here the amortised keys $\boldsymbol{k}_a^{l,h}$ depend on $\boldsymbol{F}^{l-1}$ in a deterministic manner, therefore we drop the amortised key terms in the conditioning.

Assuming prior conditional matching (i.e, $q(\mathbf{F}^l|\{\boldsymbol{u}_{a\cup g}^{l,h}\}_{h=1}^H, \boldsymbol{F}^{l-1}) = p(\mathbf{F}^l|\{\boldsymbol{u}_{a\cup g}^{l,h}\}_{h=1}^H, \boldsymbol{F}^{l-1}))$, the joint approximate posterior for $\{\boldsymbol{F}^l\}_{l=1}^L, \{\boldsymbol{u}_{a\cup g}^{l,h}\}_{l=1,h=1}^{L,H}$ is:

$$
q(\{\boldsymbol{F}^l\}_{l=1}^L, \{\boldsymbol{u}_{a\cup g}^{l,h}\}_{l=1,h=1}^{L,H}|\mathbf{F}^0) = \prod_{l=1}^L p(\mathbf{F}^l|\{\boldsymbol{u}_{a\cup g}^{l,h}\}_{h=1}^H, \boldsymbol{F}^{l-1})q(\{\boldsymbol{u}_{a\cup g}^{l,h}\}_{h=1}^H|\{\boldsymbol{k}_g^{l,h}\}_{h=1}^H, \boldsymbol{F}^{l-1}),
$$
$$\tag{24}$$

where $q(\{\boldsymbol{u}_{a\cup g}^{l,h}\}_{h=1}^H|\{\boldsymbol{k}_g^{l,h}\}_{h=1}^H, \boldsymbol{F}^{l-1}) = \prod_{h=1}^H q(\boldsymbol{u}_{a\cup g}^{l,h}|\boldsymbol{k}_g^{l,h}, \boldsymbol{F}^{l-1})$ since we let the approximate distribution for $\{\boldsymbol{u}_{a\cup g}^{l,h}\}_{h=1}^H$ also factorises across heads.

The ELBO is derived in a similar manner as the single-layer GP case (eq.(17)) and again the conditional distribution terms in $q$ and $p$ cancel with each other. This simplifies the ELBO to:

$$
\mathcal{L}_{ELBO} = E_{q(\{\boldsymbol{F}^l\}_{l=1}^L, \{\boldsymbol{u}_{a\cup g}^{l,h}\}_{l=1,h=1}^{L,H}|\boldsymbol{F}^0)}[\log \frac{p(\boldsymbol{Y}, \{\boldsymbol{F}^l\}_{l=1}^L, \{\boldsymbol{u}_{a\cup g}^{l,h}\}_{l=1,h=1}^{L,H}|\boldsymbol{F}^0)}{q(\{\boldsymbol{F}^l\}_{l=1}^L, \{\boldsymbol{u}_{a\cup g}^{l,h}\}_{l=1,h=1}^{L,H}|\boldsymbol{F}^0)}]
$$
$$
= E_{q(\boldsymbol{F}^L|\boldsymbol{F}^0, \{\boldsymbol{k}_g^{l,h}\}_{l=1,h=1}^{L,H})}[\log p(\boldsymbol{Y}|\boldsymbol{F}^L)]
$$
$$
- \sum_{l=1}^L \sum_{h=1}^H E_{q(\boldsymbol{F}^l|\boldsymbol{F}^0, \{\boldsymbol{k}_g^{j,h}\}_{j=1,h=1}^{l,H})}[KL(q(\boldsymbol{u}_{a\cup g}^{l,h}|\boldsymbol{k}_g^{l,h}, \boldsymbol{F}^{l-1})||p(\boldsymbol{u}_{a\cup g}^{l,h}|\boldsymbol{k}_g^{l,h}, \boldsymbol{F}^{l-1}))]
$$
$$\tag{25}$$

where

$$
q(\boldsymbol{F}^l|\boldsymbol{F}^0, \{\boldsymbol{k}_g^{j,h}\}_{j=1,h=1}^{l,H}) =
$$
$$
\int \prod_{j=1}^l p(\mathbf{F}^j|\{\boldsymbol{u}_{a\cup g}^{j,h}\}_{h=1}^H, \boldsymbol{F}^{j-1})q(\{\boldsymbol{u}_{a\cup g}^{j,h}\}_{h=1}^H|\{\boldsymbol{k}_g^{j,h}\}_{h=1}^H, \boldsymbol{F}^{j-1})d\boldsymbol{u}_{a\cup g}^{1:l,1:H}d\boldsymbol{F}^{1:l-1}.
$$
$$\tag{26}$$

Both terms in the ELBO can be estimated using samples generated iteratively through each layer using the reparameterization trick. For the second "regularisation" term, the KL-divergence within the expectation admits a simplified form as we assume for each attention output dimension $(d)$, an independent decoupled SVGP is fitted:

$$
KL(q(\boldsymbol{u}_{a\cup g}^{l,h}|\boldsymbol{k}_g^{l,h}, \boldsymbol{F}^{l-1})||p(\boldsymbol{u}_{a\cup g}^{l,h}|\boldsymbol{k}_g^{l,h}, \boldsymbol{F}^{l-1}))
$$
$$
= \frac{1}{2} \sum_{d=1}^D \{[\mathbf{v}_a^{l,h}]_{:,d}^\top (\boldsymbol{K}_{\boldsymbol{k}_a^{l,h}\boldsymbol{k}_a^{l,h}} - \boldsymbol{K}_{\boldsymbol{k}_a^{l,h}\boldsymbol{k}_g^{l,h}}\boldsymbol{K}_{\boldsymbol{k}_g^{l,h}\boldsymbol{k}_g^{l,h}}^{-1}\boldsymbol{K}_{\boldsymbol{k}_g^{l,h}\boldsymbol{k}_a^{l,h}})[\mathbf{v}_a^{l,h}]_{:,d} + [\mathbf{v}_g^{l,h}]_{:,d}^\top \boldsymbol{K}_{\boldsymbol{k}_g^{l,h}\boldsymbol{k}_g^{l,h}}[\mathbf{v}_g^{l,h}]_{:,d}
$$
$$
+ \text{Tr}([\boldsymbol{S}_g^{l,h}]_{:,:,d}\boldsymbol{K}_{\boldsymbol{k}_g^{l,h}\boldsymbol{k}_g^{l,h}}^{-1}) - \log |[\boldsymbol{S}_g^{l,h}]_{:,:,d}| + \log |\boldsymbol{K}_{\boldsymbol{k}_g^{l,h}\boldsymbol{k}_g^{l,h}}| - M_g\},
$$
$$\tag{27}$$

where $D$ is the total number of attention output dimensions and $M_g$ is the number of global inducing points for each head.

## C  ADDITIONAL EXPERIMENTS

### C.1  EMPIRICAL COMPARISON BETWEEN STANDARD SGPA AND DECOUPLED SGPA

In our preliminary experiments, we compare performance of ViT based on standard SGPA versus decoupled SGPA for image classification on CIFAR10 without data augmentation. We also consider decoupled SGPA based on Cheng & Boots (2017). There is no difference in the expressiveness of the basis functions between decoupled SGPA based on Cheng & Boots (2017) and decoupled SGPA based on Salimbeni et al. (2018) (the version shown in the main text). However, Salimbeni et al. (2018) tends to demonstrate faster convergence due to the orthogonal decomposition of the basis in the mean function. The posterior mean and covariance formula for decoupled SGPA based on Cheng & Boots (2017) is given as follows:

$$\begin{aligned}
\boldsymbol{m}_d &= \boldsymbol{K}_{\boldsymbol{q}\boldsymbol{k}_a}[\mathbf{v}_a]_{:,d}, \\
\boldsymbol{\Sigma}_d &= \boldsymbol{K}_{\boldsymbol{q}\boldsymbol{q}} + \boldsymbol{K}_{\boldsymbol{q}\boldsymbol{k}_g}\boldsymbol{K}_{\boldsymbol{k}_g\boldsymbol{k}_g}^{-1}([\boldsymbol{S}_g]_{:,:,d} - \boldsymbol{K}_{\boldsymbol{k}_g\boldsymbol{k}_g})\boldsymbol{K}_{\boldsymbol{k}_g\boldsymbol{k}_g}^{-1}\boldsymbol{K}_{\boldsymbol{k}_g\boldsymbol{q}},
\end{aligned} \tag{28}$$

Table 3 shows ViT based on standard SGPA and decoupled SGPA based on Cheng & Boots (2017) achieve worse performance than decouple SGPA based on Salimbeni et al. (2018). In particular, standard SGPA considerably underfits the data. Therefore we only consider decoupled SGPA based on Salimbeni et al. (2018) for the rest of the experiments.

Table 3: Test set accuracy and NLL of ViTs based on standard and two variants of decoupled SGPA, trained on CIFAR10 without data augmentation.

| Model | Accuracy | NLL |
|---|---|---|
| Standard SGPA | 0.6435±0.0039 | 1.0159±0.0065 |
| Decoupled SGPA (Cheng & Boots, 2017) | 0.7513±0.0014 | 0.7877±0.0045 |
| Decoupled SGPA (Salimbeni et al., 2018) | **0.7787±0.0024** | **0.6968±0.0032** |

### C.2  IMAGE CLASSIFICATION WITH DATA AUGMENTATION

For ViTs trained on CIFAR10 and CIFAR100 with data augmentation, we report results of in-distribution calibration and out-of-distribution robustness in Figure 7 and 8 respectively. Although some "single-model" methods can achieve lower ECE or MCE than DE and SGPAE in some cases, DE and SGPAE consistently outperform them in terms of accuracy and NLL. SGPAE again achieves the best overall performance. Among "single-model" methods, MFVI still underfits the data, but for the other methods, data augmentation improves the model performance with SNGP achieving relatively low accuracy. The difference between SGPA and other "single-model" baselines becomes smaller, perhaps due to the strong regularisation from data augmentation. Still, SGPA performs more robustly as it generally returns smaller error bars when compared to TS, MCD, and KFLLLA.

In table 4 we report for each method its average ranks in terms of AUROC and AUPR over 6 OOD detection tasks. Ensemble methods again outperform "single-model" methods with SGPAE achieving the best performance. Among "single-model" methods, SGPA achieves the best performance in terms of AUROC while KFLLLA achieves the best in terms of AUPR. In Figure 9 and 10 in Appendix A we further plot the values of AUROC and AUPR for all methods within each task.

### C.3  GRAPH PROPERTY REGRESSION WITH ZINC DATASET

For graph property regression, we assume a Laplace likelihood with a trainable scale parameter $b$ (i.e., the density of the obaservation likelihood is $g(y|f) = \frac{1}{2b}\exp-\frac{|y-f|}{b}$, where $f$ is the scalar function value output by the Transformer). We compute mean-absolute-error (MAE), root-mean-square error (RMSE) and negative-log-likelihood (NLL) to evaluate the models, with results presented in Figure 11. Moreover, we use predictive variances as scores and evaluate OOD detection performance in Figure 12. Note that MLE is useless for OOD detection in this case since it produces homogeneous predictive variances for all instances. We use a synthetic OOD dataset generated from test set: for each test instance, we remove the existing edges from the adjacency matrix and add edges between nodes that are not originally connected. Within "single-model" methods, SGPA and

Table 4: Average ranks of different methods (trained with data augmentation) in terms of AUROC and AUPR over 6 OOD detection tasks

| Model | avg. rank (AUROC)↓ | avg. rank (AUPR)↓ |
|---|---|---|
| MLE | 6.1667 | 6.0000 |
| MFVI | 8.0000 | 8.0000 |
| MCD | 4.1667 | 4.5000 |
| KFLLLA | 4.3333 | 4.0000 |
| SNGP | 6.3333 | 6.3333 |
| SGPA | 4.0000 | 4.1667 |
| DE | 1.8333 | 1.8333 |
| SGPAE | **1.1667** | **1.1667** |

MCD achieve much better results than MLE and MFVI. For this task, the difference in performance between SGPA and MCD is negligible. However, when compared to MCD, SGPA performs more robustly as it returns smaller error bars. Ensemble methods outperform "single-model" methods in OOD detection with SGPAE achieving the best result. Interestingly, for in-distribution calibration, they achieve worse performance than SGPA and MCE.

## D  LIMITATION AND CONNECTION WITH SPARSE WITHIN SPARSE GP

Unlike standard sparse GPs where the inducing points are shared across all inputs, SGPA consists of a set of input-dependent (or "amortised") inducing locations $\{k_a^{l,h}\}$ and the corresponding variational parameters, which means we are using a different mean function for each input sequence.

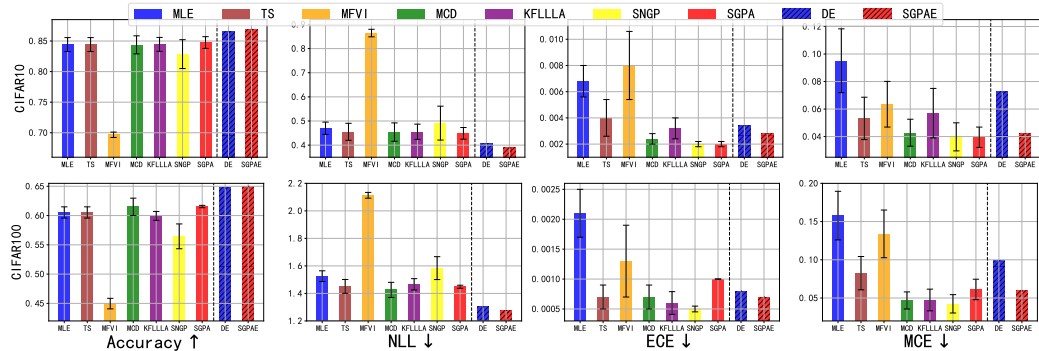

Figure 7: Test set accuracy and calibration metrics of ViTs trained on CIFAR10 (top row) and CIFAR100 (bottom row) with data augmentation.

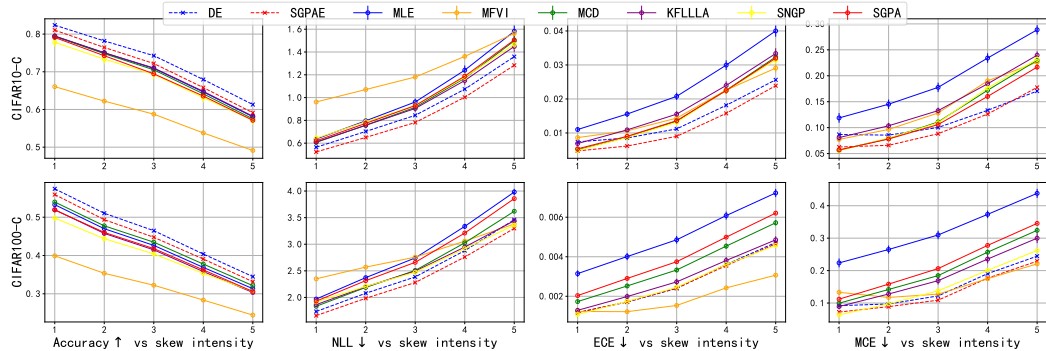

Figure 8: Accuracy and calibration metrics on CIFAR10-C (top row) and CIFAR100-C (bottom row) for ViTs trained on corresponding clean data with data augmentation.

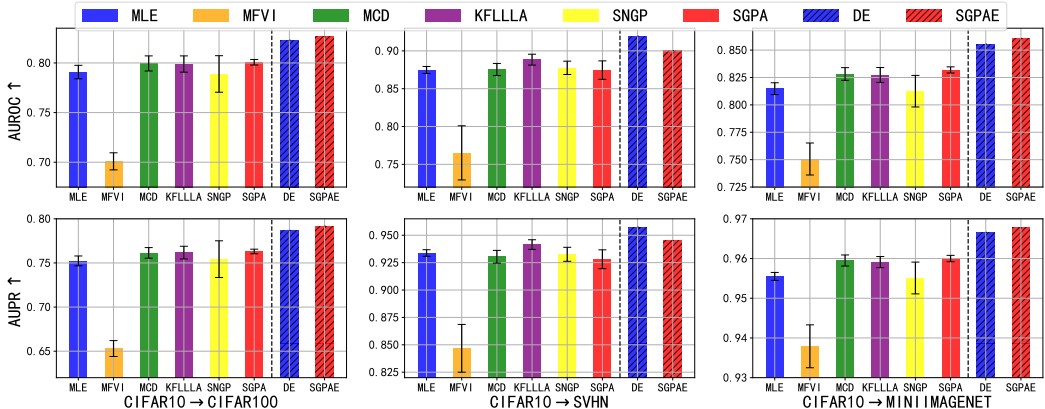

Figure 9: AUROC (top) and AUPR (bottom) metrics for OOD detection using ViTs trained on CIFAR10 with data augmentation.

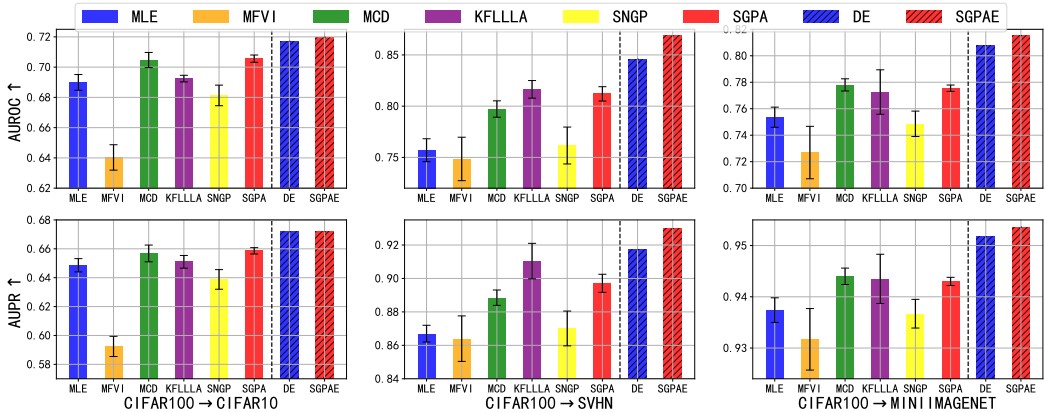

Figure 10: AUROC (top) and AUPR (bottom) metrics for OOD detection using ViTs trained on CIFAR100 with data augmentation.

Consequently, SGPA based Transformers can not perfectly model the correlation between input sequences. Instead, they can only provide marginal uncertainty for each input sequence. Nevertheless, empirically we found that correlation might not be critical in applications such as text or image classification.

One way to explain SGPA is to consider it as a sparse-within-sparse Gaussian process (SWSGP) (Tran et al., 2021; Jafrasteh et al., 2022), which allows adaptive inducing points for each input. Suppose the index set (in our case the embedding space) is $\boldsymbol{\chi}$, and $p(\boldsymbol{Z})$ is a distribution over $M$-element subset of $\boldsymbol{\chi}$ (ie. each random draw will give us $M$ inducing locations from $\boldsymbol{\chi}$). The joint prior and approximate posterior become $p(\boldsymbol{f}, \boldsymbol{u}, \boldsymbol{Z}) = p(\boldsymbol{f}|\boldsymbol{u})p(\boldsymbol{u}|\boldsymbol{Z})p(\boldsymbol{Z})$, and $q(\boldsymbol{u}, \boldsymbol{Z}) = q(\boldsymbol{u}|\boldsymbol{Z})p(\boldsymbol{Z})$, respectively. In Tran et al. (2021), for each input $\boldsymbol{x}$, they propose to select its $M$-nearest neighbours taken from the training inputs as inducing locations, so that $q(\boldsymbol{Z})$ is a delta distribution conditioned on $\boldsymbol{x}$, and $q(\boldsymbol{u}|\boldsymbol{Z})$ is the marginal variational distribution over function values evaluated at the selected inducing locations. In contrast, for each input sequence $\boldsymbol{x}$, in layer $l$, the inducing locations used by Transformer based on SGPA consist both input-dependent ones ($\{\boldsymbol{k}_a^{l,h}\}_{h=1}^H$), which are obtained from $\boldsymbol{x}$ using neural network as in Jafrasteh et al. (2022), and global inducing locations $\{\boldsymbol{k}_g^{l,h}\}_{h=1}^H$, which are shared across all input sequences.

Note that the input-dependent inducing points used during test time may not be encountered in training. Instead, we rely on the learned neural network to amortise them (Jafrasteh et al., 2022). Therefore the fitted mean function may not be consistent when test sequence includes tokens far away from the tokens in training sequences (as $\mathbf{v}_a$ may not be consistent). Empirically, we found

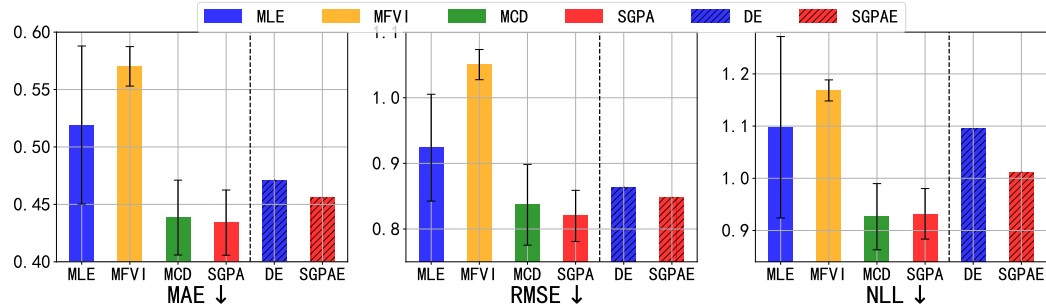

Figure 11: Test set regression error and NLL metrics for Transformers trained on ZINC.

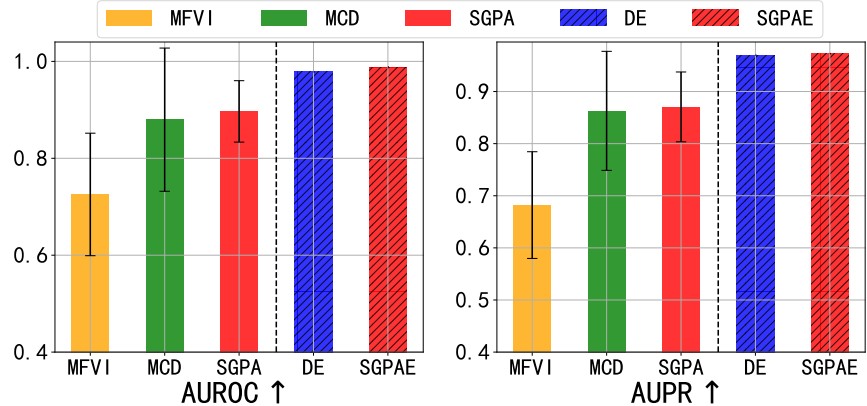

Figure 12: AUROC and AUPR metrics for OOD detection using Transformers trained on ZINC.

this inconsistency issue to be minor for in-distribution test sequences. Furthermore, we argue that for OOD inputs, although the fitted posterior mean might be unreliable, the uncertainty still increases since the posterior covariance is fully determined by the global inducing points which exhibits no inconsistency issue. Intuitively, the global keys $\{\boldsymbol{k}_g^{l,h}\}_{l=1,h=1}^{L,H}$, shared across all input sequences, play a similar role as the inducing locations in standard SVGP: they summarise the training set but focus on the uncertainty behaviour only. As a result, the posterior variance in eq.(11) still increases for queries that are less similar to the global keys as measured by the kernel. By propagating the uncertainty through each layer, we can still obtain increased uncertainty for input sequences that are very different from the training data, so that users can still be notified "when the model does not know" (Gal, 2016). However, unlike standard kernels such as RBF, it's less obvious how deep kernel measures similarity and some advantages of Bayesian inference still lose as we have no idea what prior (or inductive bias) the deep kernel represents. Therefore, in our future work, we will investigating how to inject meaningful inductive bias into deep kernels. For datasets where Euclidean distance is meaningful, we can adopt the distance-preserving techniques in SNGP (Liu et al., 2020) to enforce inductive bias into deep kernels. However, for other types of datasets, it still remains an interesting open question that we are keen to explore.

# E   EXPERIMENTAL DETAILS

**Training settings shared across experiments.** For MLE and MCD, we initially considered both dropout rates 0.1 and 0.2, but we found models trained with dropout rate 0.1 consistently outperformed models trained with dropout rate 0.2 in terms of accuracy in our preliminary experiments for image classification. Therefore, we decided to use dropout rate 0.1 for all methods except MFVI. For each layer, we use mean-pooling strategy. The non-linear mapping $G_{\phi^l}$ at each attention layer is parameterised by a 2-layer MLP as in Vaswani et al. (2017). For models with kernel-based attentions, we use exponential kernel for sentiment analysis and linguistic acceptability, (Tsai et al., 2019):

$k(\boldsymbol{x}, \boldsymbol{x}') = \sigma_f^2 \exp(\sum_{j=1}^{D} \frac{x_j x_j'}{\sigma_j^2})$, and we use ARD-RBF kernel (Rasmussen & Williams, 2006) for image classification and graph property regression: $k(\boldsymbol{x}, \boldsymbol{x}') = \sigma_f^2 \exp(-\frac{1}{2} \sum_{j=1}^{D} \frac{(x_j - x_j')^2}{\sigma_j^2})$, where $D$ is the dimension of $\boldsymbol{x}$ and $\boldsymbol{x}'$, $\sigma_f^2$ is the output variance, and $\sigma_j$ is the length-scale for the $j$-th dimension. For MFVI, MCD, SNGP and SGPA, predictive uncertainty is estimated using 10 Monte Carlo samples.

- **Initialization:** we train all our models from scratch (i.e. all parameters are randomly initialized without any pretraining). Apart from the global inducing points parameters, the rest of the parameters are the same as in standard transformers, and are initialized via the default method of the deep learning platform (we use Pytorch (Paszke et al., 2019)). Each dimension of global inducing locations and the global variational mean are randomly initialized with standard Gaussian. For the Cholesky factor used to parameterize the global variational covariance, we randomly initialize each element in the lower triangular part and each element in the log-diagonal with a standard Gaussian.

- **Optimization:** all the models are trained using ADAM optimiser (Kingma & Ba, 2015), and for each input sequence in a batch, we only draw one sample to estimate the ELBO (eq. 14). In our experiments, we observe no optimization issue: the ELBO is consistently minimized during the training without significant spikes.

**Sentiment analysis with IMDB (Maas et al., 2011).** We consider 5 different splits, each includes 35,000 training, 5,000 validation, and 10,000 test instances. The maximum number of tokens in each input sequence is 512. Architecture-wise, we use a Transformer with 1 MHSA layer and 8 attention heads, same embedding dimension and hidden dimension of 128. For SGPA we use 50 global inducing points for each head. We train all the models (except post-hoc methods, TS and KFLLLA) for 20 epochs with batch-size 32 and with a initial learning rate 0.001 which decays linearly to 0.0001. The best model is selected based on the validation accuracy computed in every training epoch.

**Linguistic acceptability with CoLA (Warstadt et al., 2019).** The 516 OOD samples provided by the original dataset are used to assess the models' OOD robustness. Within each of the 5 independent runs, the remaining 9,078 in-distribution samples are randomly split into 7,262 training and 1,816 in-distribution test instances. We use a Transformer with 2 MHSA layers, each with 4 attention heads, an embedding dimension of 128 and a hidden dimension of 256. For the input embeddings, we use ELMO-style representation (Peters et al., 2018). For SGPA we use 5 global inducing points for each head. We train all models (except post-hoc method KFLLLA) for 50 epochs with batch-size 32 and with a initial learning rate 0.0005 which decays linearly to 0.00001 and we use the model from the final epoch for evaluation.

**Image classification with CIFAR10 and CIFAR100 (Krizhevsky et al., 2009).** For both CIFAR10 and CIFAR100 datasets, we randomly split the original training set into 4,5000 training and 5,000 validation instances, and test on the original 10,000 test instances. The input images are tokenised with a patch size of $4 \times 4$. For CIFAR10 without data augmentation, we use a ViT (Dosovitskiy et al., 2021) with 5 MHSA layers, each with 4 attention heads and a hidden dimension of 128. For all the other experiments, we use a ViT with 6 layers 4 attention heads, a hidden dimension of 256. We train all models (except post-hoc methods, TS and KFLLLA) except SGPA for 600 epochs with batch-size 100 and with initial learning rate of 0.0005 which decays linearly to 0.00001. For SGPA, we use 32 global inducing points for each head, and we use the parameters from the 100th epoch of MLE to initialize the deep kernel hyperparameters, and continue training for 500 epochs. The best model is selected based on the validation accuracy computed every 10 epochs. For experiments with data augmentation, we consider the same data augmentation strategy (3-Augment) as in Touvron et al. (2022).

**Graph property regression with ZINC (Dwivedi et al., 2020).** The results are presented in C.3 which are averaged from 3 independent runs. We use the same split as in (Dwivedi et al., 2020), resulting in 10,000 training, 1,000 validation, and 1,000 test instances. Instead of applying graph-specific modifications to the network architecture, we use the feature engineering technique proposed in Kim et al. (2022) to transform each graph into a sequence of input embeddings. We consider Transformers with 8 layer and 8 attention heads, same embedding dimension and hidden dimension of 80. For SGPA we use 10 global inducing points for each head. We train all models for 500 epochs

with batch-size 64 and with an initial learning rate 0.0004 which decays linearly to 0.000002. The best model is selected based on the validation accuracy computed at the end of every 10 epochs.

## F   RUNNING TIME

We analyse the wall-clock training and inference time of SGPA here. In Table 5, we present the computational time for a single batch at the inference stage with 10 Monte Carlo samples for CoLA (batch size = 227) and CIFAR10 (batch size = 200) (results obtained using a single Nvidia GTX 2080 Ti GPU card). For SGPA, we first pay an one-off cost of inverting the kernel matrices related to global inducing points ($K_{k_g k_g}^{-1}$). Once this is done, we treat them as constant matrices and plug them in eq.(11). When generating samples that are passed to the next layer, we diagonalize the covariance ($\Sigma_d$) in eq.(11) to avoid the costly computation of the Cholesky factor of $\Sigma_d$.

The computational cost depends on the number of global inducing points used. For CoLA, we only used 5 global inducing points for each head, and the relative difference between inference times for MCD and SGPA is less than that for CIFAR10, where we use 32 global inducing points for each head. It is noteworthy that we haven't done extensive hyperparameter tuning for the number of global inducing points. It is likely that SGPA can still work well with a smaller number of global inducing points. For example, for CIFAR10, we recently trained an SGPA model with 16 global inducing points for each head, and we did not see a considerable performance drop (Accuracy: 0.7790, NLL: 0.7259, ECE: 0.0119, MCE: 0.0819). In this case, the inference time can be further reduced from 0.986 to 0.807.

Table 5: The computational time (in $s$) for a single batch at the inference stage with 10 Monte Carlo samples for CoLA (batch size = 227) and CIFAR10 (batch size = 200) (results obtained using a single Nvidia GTX 2080 Ti GPU card).

|  | MCD | MFVI | SGPA |
|---|---|---|---|
| CoLA | 1.839 | 1.882 | 2.358 |
| CIFAR10 (32 $k_g$) | 0.234 | 0.395 | 0.986 |
| CIFAR10 (16 $k_g$) | 0.234 | 0.395 | 0.807 |

In Table 6 we present the training time (in $s$) of one epoch for SGPA and MLE on CoLA (batch size = 32) and CIFAR10 (batch size = 100) (results obtained using a single Nvidia GTX 2080 Ti GPU card):

Table 6: The training time (in $s$) of one epoch for SGPA and MLE on CoLA (batch size = 32) and CIFAR10 (batch size = 100) (results obtained using a single Nvidia GTX 2080 Ti GPU card).

|  | MLE | SGPA |
|---|---|---|
| CoLA | 17.834 | 20.089 |
| CIFAR10 (32 $k_g$) | 30.593 | 138.609 |
| CIFAR10 (16 $k_g$) | 30.593 | 109.298 |

## G   RESULTS IN TABLES

We present numerical results (mean±standard error) for all experiments in tables.

We show in-distribution results in Tables 7 to 13.

We show OOD robustness results in Tables 14 to 30.

We show OOD detection results in Tables 31 to 35.

Table 7: In-distribution performance: sentiment analysis with IMDB

|        | Model  | Accuracy | NLL | ECE | MCE |
|--------|--------|----------|-----|-----|-----|
| SDP    | MLE    | 0.8806±0.0016 | 0.3267±0.0068 | 0.0548±0.0029 | 0.1432±0.0093 |
| Kernel | MLE    | 0.8822±0.0019 | 0.3202±0.0136 | 0.0519±0.0055 | 0.1568±0.0273 |
|        | TS     | 0.8822±0.0019 | 0.3202±0.0136 | 0.0519±0.0055 | 0.1568±0.0004 |
|        | MFVI   | 0.8799±0.0022 | 0.3476±0.0230 | 0.0585±0.0106 | 0.1493±0.0271 |
|        | MCD    | 0.8817±0.0014 | 0.2986±0.0070 | 0.0285±0.0058 | 0.0910±0.0218 |
|        | KFLLLA | 0.8822±0.0019 | 0.4366±0.0015 | 0.1905±0.0034 | 0.2263±0.0048 |
|        | SNGP   | 0.8783±0.0013 | 0.3499±0.0080 | 0.0646±0.0037 | 0.1741±0.0106 |
|        | SGPA   | 0.8875±0.0004 | 0.2823±0.0027 | 0.0215±0.0034 | 0.0647±0.0105 |

Table 8: In-distribution performance: linguistic acceptability with CoLA

|        | Model  | MCC | NLL | ECE | MCE |
|--------|--------|-----|-----|-----|-----|
| SDP    | MLE    | 25.0408±0.8223 | 1.8976±0.0223 | 0.2654±0.0039 | 0.3887±0.0230 |
| Kernel | MLE    | 26.0722±0.4300 | 2.0083±0.0432 | 0.2643±0.0077 | 0.3702±0.0217 |
|        | MFVI   | 21.0864±1.8124 | 1.0809±0.0092 | 0.2018±0.0080 | 0.266±0.0113 |
|        | MCD    | 26.4399±0.4939 | 0.9121±0.0154 | 0.2113±0.0049 | 0.3233±0.0066 |
|        | KFLLLA | 26.0295±0.4050 | 0.6023±0.0077 | 0.0241±0.0044 | 0.1063±0.0204 |
|        | SNGP   | 27.8715±1.2584 | 1.2075±0.0418 | 0.2325±0.0067 | 0.3230±0.0146 |
|        | SGPA   | 27.2380±1.1188 | 0.9070±0.0207 | 0.2076±0.0082 | 0.3062±0.0097 |

Table 9: In-distribution performance: image classification for CIFAR10 without data augmentation

|        | Model  | Accuracy | NLL | ECE | MCE |
|--------|--------|----------|-----|-----|-----|
| SDP    | MLE    | 0.7439±0.0015 | 2.2940±0.0277 | 0.0437±0.0004 | 0.4509±0.0079 |
| Kernel | MLE    | 0.7811± 0.0019 | 1.3395±0.0135 | 0.0331±0.0004 | 0.3724±0.0103 |
|        | TS     | 0.7811±0.0019 | 0.8254±0.0077 | 0.0230±0.0004 | 0.2499±0.0081 |
|        | MFVI   | 0.7202±0.0024 | 0.7995±0.0073 | 0.0058±0.0009 | 0.0664±0.0064 |
|        | MCD    | 0.7910±0.0015 | 0.7789±0.0063 | 0.0159±0.0004 | 0.1501±0.0055 |
|        | KFLLLA | 0.7752±0.0019 | 0.7201±0.0242 | 0.0151±0.0020 | 0.1497±0.0217 |
|        | SNGP   | 0.7837±0.0025 | 0.7534±0.0031 | 0.0156±0.0001 | 0.1443±0.0049 |
|        | SGPA   | 0.7787±0.0024 | 0.6968±0.0032 | 0.0100±0.0004 | 0.0825±0.0066 |
|        | DE     | 0.8235 | 0.6934 | 0.0110 | 0.0699 |
|        | SGPAE  | 0.8172 | 0.5657 | 0.0054 | 0.0579 |

Table 10: In-distribution performance: image classification for CIFAR10 with data augmentation

|        | Model  | Accuracy | NLL | ECE | MCE |
|--------|--------|----------|-----|-----|-----|
| SDP    | MLE    | 0.8898±0.0009 | 0.6073±0.0330 | 0.0163±0.0005 | 0.3250±0.0118 |
| Kernel | MLE    | 0.8442±0.0057 | 0.4700±0.0127 | 0.0068±0.0006 | 0.0951±0.0116 |
|        | TS     | 0.8442±0.0057 | 0.4552±0.0176 | 0.0040 ± 0.0007 | 0.0532 ± 0.0077 |
|        | MFVI   | 0.6967±0.0022 | 0.8643±0.0078 | 0.0080±0.0013 | 0.0635±0.0083 |
|        | MCD    | 0.8437±0.0074 | 0.4541±0.0190 | 0.0024±0.0002 | 0.0428±0.0049 |
|        | KFLLLA | 0.8445±0.0057 | 0.4554±0.0158 | 0.0032±0.0004 | 0.0570±0.0090 |
|        | SNGP   | 0.8286±0.0118 | 0.4912±0.0351 | 0.0020±0.0001 | 0.0398±0.0051 |
|        | SGPA   | 0.8475±0.0048 | 0.4489±0.0121 | 0.0020±0.0001 | 0.0395±0.0037 |
|        | DE     | 0.8654 | 0.4062 | 0.0034 | 0.0724 |
|        | SGPAE  | 0.8691 | 0.3885 | 0.0028 | 0.0421 |

Table 11: In-distribution performance: image classification for CIFAR100 without data augmentation

| | Model | Accuracy | NLL | ECE | MCE |
|---|---|---|---|---|---|
| SDP | MLE | 0.4893±0.0022 | 5.7677±0.0492 | 0.0086±0.0001 | 0.6216±0.0076 |
| Kernel | MLE | 0.5216±0.0100 | 4.3898±0.0829 | 0.0076±0.0002 | 0.5696±0.0093 |
| | TS | 0.5216±0.0100 | 2.5863±0.0527 | 0.0057±0.0001 | 0.4067±0.0032 |
| | MFVI | 0.4117±0.0015 | 2.4195±0.0113 | 0.0025±0.0000 | 0.1021±0.0057 |
| | MCD | 0.5341±0.0096 | 2.7122±0.0329 | 0.0045±0.0000 | 0.2651±0.0116 |
| | KFLLLA | 0.5084±0.0103 | 3.2984±0.0210 | 0.0031±0.0002 | 0.7690±0.0159 |
| | SNGP | 0.4715±0.0039 | 2.8296±0.0338 | 0.0051±0.0001 | 0.3028±0.0082 |
| | SGPA | 0.5302±0.0071 | 2.5643±0.0813 | 0.0042±0.0001 | 0.2463±0.0071 |
| | DE | 0.5815 | 2.4887 | 0.0039 | 0.1381 |
| | SGPAE | 0.5700 | 1.9434 | 0.0020 | 0.0704 |

Table 12: In-distribution performance: image classification for CIFAR100 with data augmentation

| | Model | Accuracy | NLL | ECE | MCE |
|---|---|---|---|---|---|
| SDP | MLE | 0.5984±0.0024 | 1.8743±0.2032 | 0.0036±0.0006 | 0.2792±0.0512 |
| Kernel | MLE | 0.6053±0.0048 | 1.5254±0.0195 | 0.0021±0.0002 | 0.1578±0.0159 |
| | TS | 0.6053±0.0048 | 1.4512±0.0251 | 0.0007±0.0001 | 0.0825±0.0109 |
| | MFVI | 0.4496±0.0045 | 2.1132±0.0105 | 0.0013±0.0003 | 0.1339±0.0156 |
| | MCD | 0.6149±0.0074 | 1.4255±0.0275 | 0.0007±0.0001 | 0.0468±0.0056 |
| | KFLLLA | 0.5994±0.0038 | 1.4661±0.0206 | 0.0005±0.0001 | 0.0474±0.0071 |
| | SNGP | 0.5645±0.0106 | 1.5839±0.0416 | 0.0005±0.0000 | 0.0424±0.0060 |
| | SGPA | 0.6154±0.0010 | 1.4486±0.0055 | 0.0010±0.0000 | 0.0612±0.0067 |
| | DE | 0.6471 | 1.3023 | 0.0008 | 0.0986 |
| | SGPEA | 0.65 | 1.2761 | 0.0007 | 0.0593 |

Table 13: In-distribution performance: graph property regression with ZINC dataset

| | Model | MAE | RMSE | NLL |
|---|---|---|---|---|
| SDP | MLE | 0.5733±0.0084 | 1.1459±0.0139 | 1.1459±0.0139 |
| Kernel | MLE | 0.5191±0.0344 | 1.0977±0.0869 | 1.0977±0.0869 |
| | MCD | 0.4385±0.0163 | 0.8369±0.0308 | 0.9264±0.0317 |
| | MFVI | 0.5702±0.0086 | 1.1050±0.0115 | 1.1683±0.0101 |
| | SGPA | 0.4341±0.0142 | 0.8199±0.0195 | 0.9319±0.0242 |
| | DE | 0.4711 | 0.8635 | 1.0949 |
| | SGPAE | 0.4557 | 0.8479 | 1.0118 |

Table 14: OOD robustness: linguistic acceptability with CoLA dataset

| | Model | MCC | NLL | ECE | MCE |
|---|---|---|---|---|---|
| SDP | MLE | 19.4835±2.0234 | 2.0574±0.0516 | 0.2914±0.0098 | 0.4913±0.0418 |
| Kernel | MLE | 18.6467±1.6116 | 2.3565±0.0974 | 0.2909±0.0067 | 0.3864±0.0303 |
| | MFVI | 16.6363±1.5793 | 1.1349±0.0445 | 0.2184±0.0035 | 0.3289±0.0151 |
| | MCD | 20.4774±1.4727 | 1.0114±0.0234 | 0.2370±0.0075 | 0.3439±0.0177 |
| | KFLLLA | 18.6250±1.6321 | 0.6299±0.0067 | 0.0361±0.0065 | 0.0876±0.0065 |
| | SNGP | 23.6410±1.5986 | 1.3070±0.0412 | 0.2537±0.0049 | 0.3828±0.0189 |
| | SGPA | 22.9190±1.5986 | 0.9514±0.0261 | 0.2257±0.0069 | 0.3399±0.0294 |

Table 15: Accuracy of CIFAR10-C for ViTs trained on clean data without data augmentation.

| | Model | Skew Intensity | | | | |
| | | 1 | 2 | 3 | 4 | 5 |
|---|---|---|---|---|---|---|
| SDP | MLE | 0.7120±0.0011 | 0.6704±0.0011 | 0.6359±0.0014 | 0.5883±0.0017 | 0.5354±0.0015 |
| Kernel | MLE | 0.7254±0.0019 | 0.6632±0.0028 | 0.6139±0.0023 | 0.5494±0.0028 | 0.4863±0.0027 |
| | MFVI | 0.6606±0.0093 | 0.6109±0.0083 | 0.5730±0.0077 | 0.5266±0.0073 | 0.4778±0.0065 |
| | MCD | 0.7327±0.0013 | 0.6717±0.0015 | 0.6229±0.0016 | 0.5603±0.0019 | 0.4982±0.0020 |
| | KFLLLA | 0.7186±0.0014 | 0.6580±0.0035 | 0.6092±0.0038 | 0.5462±0.0042 | 0.4825±0.0038 |
| | SNGP | 0.7281±0.0024 | 0.6676±0.0025 | 0.6208±0.0020 | 0.5622±0.0020 | 0.5006±0.0019 |
| | SGPA | 0.7175±0.0038 | 0.6559±0.0042 | 0.6067±0.0040 | 0.5466±0.0033 | 0.4832±0.0046 |
| | DE | 0.7700 | 0.7089 | 0.6585 | 0.5908 | 0.5246 |
| | SGPAE | 0.7516 | 0.6911 | 0.6401 | 0.5779 | 0.5173 |

Table 16: Accuracy of CIFAR10-C for ViTs trained on clean data with data augmentation.

| | Model | Skew Intensity | | | | |
| | | 1 | 2 | 3 | 4 | 5 |
|---|---|---|---|---|---|---|
| SDP | MLE | 0.8516±0.0010 | 0.8139±0.0011 | 0.7752±0.0010 | 0.7165±0.0013 | 0.6500±0.0012 |
| Kernel | MLE | 0.7944±0.0061 | 0.7499±0.0074 | 0.7090±0.0082 | 0.6465±0.0085 | 0.5807±0.0094 |
| | MFVI | 0.6605±0.0016 | 0.6222±0.0024 | 0.5875±0.0028 | 0.5377±0.0032 | 0.4913±0.0040 |
| | MCD | 0.7939±0.0068 | 0.7480±0.0082 | 0.7052±0.0089 | 0.6415±0.0092 | 0.5751±0.0097 |
| | KFLLLA | 0.7949±0.0030 | 0.7508±0.0037 | 0.7100±0.0041 | 0.6480±0.0043 | 0.5821±0.0047 |
| | SNGP | 0.7783±0.0062 | 0.7336±0.0069 | 0.6932±0.0069 | 0.6314±0.0062 | 0.5710±0.0057 |
| | SGPA | 0.7910±0.0027 | 0.7426±0.0021 | 0.6946±0.0075 | 0.6356±0.0033 | 0.5712±0.0020 |
| | DE | 0.8237 | 0.7819 | 0.7428 | 0.6795 | 0.6130 |
| | SGPAE | 0.8105 | 0.7646 | 0.7220 | 0.6581 | 0.5914 |

Table 17: Accuracy of CIFAR100-C for ViTs trained on clean data without data augmentation.

| | Model | Skew Intensity | | | | |
| | | 1 | 2 | 3 | 4 | 5 |
|---|---|---|---|---|---|---|
| SDP | MLE | 0.3947±0.0022 | 0.3484±0.0018 | 0.3223±0.0014 | 0.2805±0.0013 | 0.2402±0.0009 |
| Kernel | MLE | 0.4456±0.0075 | 0.3769±0.0058 | 0.3373±0.0049 | 0.2835±0.0034 | 0.2339±0.0029 |
| | MFVI | 0.3380±0.0020 | 0.2832±0.0022 | 0.2540±0.0018 | 0.2155±0.0014 | 0.1803±0.0015 |
| | MCD | 0.4526±0.0074 | 0.3840±0.0061 | 0.3446±0.0052 | 0.2913±0.0037 | 0.2403±0.0031 |
| | KFLLLA | 0.4350±0.0070 | 0.3656±0.0053 | 0.3251±0.0045 | 0.2717±0.0030 | 0.2229±0.0025 |
| | SNGP | 0.4053±0.0020 | 0.3455±0.0014 | 0.3093±0.0015 | 0.2624±0.0015 | 0.2203±0.0011 |
| | SGPA | 0.4472±0.0050 | 0.3764±0.0035 | 0.3371±0.0027 | 0.2828±0.0021 | 0.2317±0.0022 |
| | DE | 0.5054 | 0.4289 | 0.3858 | 0.3234 | 0.2690 |
| | SGPAE | 0.4848 | 0.4104 | 0.3675 | 0.3087 | 0.2539 |

Table 18: Accuracy of CIFAR100-C for ViTs trained on clean data with data augmentation.

| | Model | Skew Intensity | | | | |
| | | 1 | 2 | 3 | 4 | 5 |
|---|---|---|---|---|---|---|
| SDP | MLE | 0.5384±0.0030 | 0.4854±0.0038 | 0.4464±0.0041 | 0.3930±0.0037 | 0.3410±0.0029 |
| Kernel | MLE | 0.5327±0.0018 | 0.4700±0.0018 | 0.4264±0.0017 | 0.3690±0.0017 | 0.3134±0.0013 |
| | MFVI | 0.3991±0.0019 | 0.3535±0.0017 | 0.3224±0.0019 | 0.2836±0.0017 | 0.2444±0.0014 |
| | MCD | 0.5397±0.0020 | 0.4775±0.0018 | 0.4349±0.0015 | 0.3777±0.0011 | 0.3213±0.0010 |
| | KFLLLA | 0.5196±0.0019 | 0.4577±0.0018 | 0.4148±0.0017 | 0.3581±0.0014 | 0.3035±0.0014 |
| | SNGP | 0.4979±0.0038 | 0.4440±0.0035 | 0.4042±0.0034 | 0.3534±0.0029 | 0.3066±0.0023 |
| | SGPA | 0.5189±0.0054 | 0.4603±0.0010 | 0.4181±0.0013 | 0.3625±0.0014 | 0.3061±0.0014 |
| | DE | 0.5742 | 0.5104 | 0.4649 | 0.4039 | 0.3449 |
| | SGPAE | 0.5592 | 0.4936 | 0.4476 | 0.3908 | 0.3316 |

Table 19: NLL of CIFAR10-C for ViTs trained on clean data without data augmentation.

| | | | | Skew Intensity | | |
|---|---|---|---|---|---|---|
| | Model | 1 | 2 | 3 | 4 | 5 |
| SDP | MLE | 2.6411±0.0202 | 3.1620±0.0202 | 3.6449±0.0231 | 4.3843±0.0283 | 5.2626±0.0287 |
| Kernel | MLE | 1.7906±0.0179 | 2.3460±0.0321 | 2.8899±0.0385 | 3.6899±0.0544 | 4.5003±0.0811 |
| | MFVI | 0.9761±0.0274 | 1.1345±0.0267 | 1.2737±0.0259 | 1.4705±0.0237 | 1.7038±0.0222 |
| | MCD | 1.1466±0.0099 | 1.5153±0.0168 | 1.8926±0.0234 | 2.4692±0.0315 | 3.1086±0.0431 |
| | KFLLLA | 0.9573±0.0498 | 1.2092±0.0577 | 1.4435±0.0683 | 1.7889±0.0937 | 2.1537±0.1198 |
| | SNGP | 1.0579±0.0066 | 1.3681±0.0115 | 1.6570±0.0129 | 2.0807±0.0136 | 2.5271±0.0146 |
| | SGPA | 0.9603±0.0128 | 1.2340±0.0213 | 1.4972±0.0283 | 1.8951±0.0383 | 2.3812±0.0586 |
| | DE | 0.9317 | 1.2424 | 1.5716 | 2.0911 | 2.6373 |
| | SGPAE | 0.7816 | 0.9985 | 1.2125 | 1.5365 | 1.9349 |

Table 20: NLL of CIFAR10-C for ViTs trained on clean data with data augmentation.

| | | | | Skew Intensity | | |
|---|---|---|---|---|---|---|
| | Model | 1 | 2 | 3 | 4 | 5 |
| SDP | MLE | 0.8306±0.0422 | 1.0712±0.0559 | 1.3665±0.0753 | 1.8795±0.1085 | 2.5810±0.1571 |
| Kernel | MLE | 0.6364±0.0140 | 0.7960±0.0207 | 0.9614±0.0296 | 1.2404±0.0416 | 1.5818±0.0538 |
| | MFVI | 0.9616±0.0044 | 1.0706±0.0061 | 1.1802±0.0098 | 1.3612±0.0160 | 1.5629±0.0248 |
| | MCD | 0.6067±0.0180 | 0.7579±0.0236 | 0.9144±0.0304 | 1.1707±0.0363 | 1.4977±0.0420 |
| | KFLLLA | 0.6104±0.0080 | 0.7565±0.0108 | 0.9044±0.0144 | 1.1496±0.0183 | 1.4487±0.0230 |
| | SNGP | 0.6439±0.0187 | 0.7874±0.0218 | 0.9304±0.0225 | 1.1773±0.0223 | 1.4671±0.0225 |
| | SGPA | 0.6204±0.0054 | 0.7739±0.0054 | 0.9298±0.0047 | 1.1873±0.0118 | 1.5058±0.0226 |
| | DE | 0.5641 | 0.7028 | 0.8441 | 1.0737 | 1.3595 |
| | SGPAE | 0.5230 | 0.6503 | 0.7814 | 1.0027 | 1.2835 |

Table 21: NLL of CIFAR100-C for ViTs trained on clean data without data augmentation.

| | | | | Skew Intensity | | |
|---|---|---|---|---|---|---|
| | Model | 1 | 2 | 3 | 4 | 5 |
| SDP | MLE | 7.4649±0.0475 | 8.5233±0.0524 | 9.1964±0.0527 | 10.4061±0.0555 | 11.8540±0.0649 |
| Kernel | MLE | 5.5126±0.0578 | 6.7077±0.0416 | 7.5503±0.0237 | 9.0093±0.0353 | 10.5647±0.0699 |
| | MFVI | 3.0138±0.0132 | 3.4762±0.0136 | 3.7921±0.0135 | 4.3295±0.0153 | 4.9042±0.0200 |
| | MCD | 3.8996±0.0266 | 4.8412±0.0178 | 5.5397±0.0269 | 6.7984±0.0591 | 8.2454±0.0959 |
| | KFLLLA | 3.4366±0.0172 | 3.5660±0.0160 | 3.6457±0.0152 | 3.7645±0.0135 | 3.8862±0.0115 |
| | SNGP | 3.6404±0.0297 | 4.3067±0.0360 | 4.7994±0.0446 | 5.6242±0.0502 | 6.5029±0.0504 |
| | SGPA | 3.6565±0.1290 | 4.5188±0.1401 | 5.1522±0.1510 | 6.3096±0.1784 | 7.6306±0.2161 |
| | DE | 3.1930 | 4.0004 | 4.6072 | 5.7357 | 6.9765 |
| | SGPAE | 2.8407 | 3.5393 | 4.0466 | 4.9890 | 6.0803 |

Table 22: NLL of CIFAR100-C for ViTs trained on clean data with data augmentation.

| | | | | Skew Intensity | | |
|---|---|---|---|---|---|---|
| | Model | 1 | 2 | 3 | 4 | 5 |
| SDP | MLE | 2.2595±0.2276 | 2.6313±0.2735 | 2.9786±0.3148 | 3.5393±0.3720 | 4.1865±0.4426 |
| Kernel | MLE | 1.9695±0.0167 | 2.3721±0.0260 | 2.7423±0.0346 | 3.3355±0.0500 | 3.9813±0.0602 |
| | MFVI | 2.3501±0.0073 | 2.5702±0.0084 | 2.7571±0.0112 | 3.0582±0.0123 | 3.3805±0.0147 |
| | MCD | 1.8337±0.0048 | 2.1874±0.0073 | 2.5076±0.0112 | 3.0263±0.0205 | 3.6196±0.0273 |
| | KFLLLA | 1.8645±0.0092 | 2.2008±0.0095 | 2.4935±0.0095 | 2.9465±0.0112 | 3.4418±0.0135 |
| | SNGP | 1.9167±0.0161 | 2.2067±0.0162 | 2.4753±0.0163 | 2.9166±0.0133 | 3.3598±0.0116 |
| | SGPA | 1.9307±0.0075 | 2.3154±0.0104 | 2.6598±0.0130 | 3.2105±0.0151 | 3.8575±0.0180 |
| | DE | 1.7354 | 2.0787 | 2.3857 | 2.8802 | 3.4604 |
| | SGPAE | 1.6583 | 1.9839 | 2.2802 | 2.7592 | 3.2976 |

Table 23: ECE of CIFAR10-C for ViTs trained on clean data without data augmentation.

| | Model | Skew Intensity | | | | |
| | | 1 | 2 | 3 | 4 | 5 |
|---|---|---|---|---|---|---|
| SDP | MLE | 0.0491±0.0002 | 0.0564±0.0002 | 0.0627±0.0003 | 0.0713±0.0003 | 0.0811±0.0003 |
| Kernel | MLE | 0.0425±0.0003 | 0.0530±0.0005 | 0.0617±0.0005 | 0.0734±0.0006 | 0.0846±0.0006 |
| | MFVI | 0.0077±0.0017 | 0.0122±0.0021 | 0.0170±0.0021 | 0.0243±0.0021 | 0.0316±0.0021 |
| | MCD | 0.0242±0.0002 | 0.0328±0.0003 | 0.0405±0.0004 | 0.0515±0.0004 | 0.0621±0.0006 |
| | KFLLLA | 0.0226±0.0027 | 0.0305±0.0028 | 0.0374±0.0028 | 0.0472±0.0031 | 0.0566±0.0034 |
| | SNGP | 0.0233±0.0002 | 0.0317±0.0003 | 0.0389±0.0003 | 0.0487±0.0002 | 0.0588±0.0002 |
| | SGPA | 0.0178±0.0006 | 0.0258±0.0007 | 0.0330±0.0008 | 0.0432±0.0008 | 0.0529±0.0010 |
| | DE | 0.0160 | 0.0227 | 0.0291 | 0.0389 | 0.0484 |
| | SGPAE | 0.0088 | 0.0136 | 0.0191 | 0.0282 | 0.0368 |

Table 24: ECE of CIFAR10-C for ViTs trained on clean data with data augmentation.

| | Model | Skew Intensity | | | | |
| | | 1 | 2 | 3 | 4 | 5 |
|---|---|---|---|---|---|---|
| SDP | MLE | 0.0219±0.0007 | 0.0275±0.0009 | 0.0337±0.0011 | 0.0436±0.0013 | 0.0551±0.0015 |
| Kernel | MLE | 0.0110±0.0007 | 0.0156±0.0008 | 0.0208±0.0011 | 0.0299±0.0014 | 0.0400±0.0017 |
| | MFVI | 0.0086±0.0007 | 0.0106±0.0002 | 0.0144±0.0005 | 0.0226±0.0007 | 0.0291±0.0012 |
| | MCD | 0.0050±0.0003 | 0.0088±0.0005 | 0.0134±0.0008 | 0.0225±0.0011 | 0.0324±0.0013 |
| | KFLLLA | 0.0069±0.0003 | 0.0109±0.0003 | 0.0155±0.0004 | 0.0240±0.0006 | 0.0334±0.0008 |
| | SNGP | 0.0049±0.0002 | 0.0087±0.0004 | 0.0132±0.0005 | 0.0225±0.0006 | 0.0314±0.0006 |
| | SGPA | 0.0053±0.0003 | 0.0090±0.0004 | 0.0136±0.0005 | 0.0225±0.0009 | 0.0320±0.0010 |
| | DE | 0.0074 | 0.0085 | 0.0112 | 0.0181 | 0.0256 |
| | SGPAE | 0.0047 | 0.0061 | 0.0090 | 0.0158 | 0.0239 |

Table 25: ECE of CIFAR100-C for ViTs trained on clean data without data augmentation.

| | Model | Skew Intensity | | | | |
| | | 1 | 2 | 3 | 4 | 5 |
|---|---|---|---|---|---|---|
| SDP | MLE | 0.0104±0.0000 | 0.0113±0.0000 | 0.0118±0.0000 | 0.0127±0.0000 | 0.0135±0.0000 |
| Kernel | MLE | 0.0090±0.0001 | 0.0103±0.0001 | 0.0111±0.0001 | 0.0123±0.0000 | 0.0134±0.0000 |
| | MFVI | 0.0043±0.0000 | 0.0054±0.0000 | 0.0062±0.0000 | 0.0073±0.0001 | 0.0083±0.0001 |
| | MCD | 0.0063±0.0000 | 0.0077±0.0000 | 0.0085±0.0000 | 0.0099±0.0000 | 0.0112±0.0000 |
| | KFLLLA | 0.0022±0.0002 | 0.0016±0.0001 | 0.0013±0.0001 | 0.0010±0.0001 | 0.0008±0.0001 |
| | SNGP | 0.0066±0.0001 | 0.0078±0.0001 | 0.0085±0.0001 | 0.0097±0.0001 | 0.0107±0.0001 |
| | SGPA | 0.0061±0.0001 | 0.0074±0.0001 | 0.0083±0.0001 | 0.0096±0.0001 | 0.0109±0.0001 |
| | DE | 0.0051 | 0.0063 | 0.0072 | 0.0086 | 0.0099 |
| | SGPAE | 0.0042 | 0.0054 | 0.0062 | 0.0076 | 0.0090 |

Table 26: ECE of CIFAR100-C for ViTs trained on clean data with data augmentation.

| | Model | Skew Intensity | | | | |
| | | 1 | 2 | 3 | 4 | 5 |
|---|---|---|---|---|---|---|
| SDP | MLE | 0.0043±0.0006 | 0.0050±0.0007 | 0.0057±0.0007 | 0.0068±0.0007 | 0.0078±0.0007 |
| Kernel | MLE | 0.0031±0.0002 | 0.0040±0.0002 | 0.0049±0.0002 | 0.0061±0.0002 | 0.0072±0.0002 |
| | MFVI | 0.0012±0.0001 | 0.0012±0.0001 | 0.0015±0.0001 | 0.0024±0.0000 | 0.0031±0.0001 |
| | MCD | 0.0017±0.0001 | 0.0025±0.0001 | 0.0033±0.0001 | 0.0045±0.0001 | 0.0057±0.0001 |
| | KFLLLA | 0.0013±0.0001 | 0.0020±0.0001 | 0.0027±0.0001 | 0.0038±0.0001 | 0.0049±0.0001 |
| | SNGP | 0.0011±0.0000 | 0.0017±0.0000 | 0.0025±0.0000 | 0.0036±0.0000 | 0.0046±0.0000 |
| | SGPA | 0.0020±0.0001 | 0.0029±0.0001 | 0.0037±0.0001 | 0.0050±0.0001 | 0.0062±0.0001 |
| | DE | 0.0012 | 0.0017 | 0.0024 | 0.0036 | 0.0047 |
| | SGPAE | 0.0012 | 0.0017 | 0.0024 | 0.0035 | 0.0047 |

Table 27: MCE of CIFAR10-C for ViTs trained on clean data without data augmentation.

| | Model | Skew Intensity | | | | |
| | | 1 | 2 | 3 | 4 | 5 |
|---|---|---|---|---|---|---|
| SDP | MLE | 0.4666±0.0027 | 0.4840±0.0026 | 0.5041±0.0018 | 0.5280±0.0019 | 0.5534±0.0020 |
| Kernel | MLE | 0.4038±0.0028 | 0.4362±0.0020 | 0.4655±0.0020 | 0.5051±0.0033 | 0.5442±0.0036 |
| | MFVI | 0.0598±0.0112 | 0.0852±0.0150 | 0.1125±0.0147 | 0.1586±0.0147 | 0.2011±0.0149 |
| | MCD | 0.1902±0.0021 | 0.2402±0.0025 | 0.2807±0.0026 | 0.3413±0.0036 | 0.3949±0.0041 |
| | KFLLLA | 0.1987±0.0251 | 0.2362±0.0241 | 0.2729±0.0238 | 0.3248±0.0220 | 0.3733±0.0210 |
| | SNGP | 0.1854±0.0018 | 0.2337±0.0021 | 0.2710±0.0011 | 0.3229±0.0021 | 0.3798±0.0008 |
| | SGPA | 0.1273±0.0042 | 0.1801±0.0046 | 0.2212±0.0061 | 0.2807±0.0057 | 0.3340±0.0047 |
| | DE | 0.1051 | 0.1402 | 0.1813 | 0.2476 | 0.3091 |
| | SGPAE | 0.0570 | 0.0816 | 0.1173 | 0.1757 | 0.2208 |

Table 28: MCE of CIFAR10-C for ViTs trained on clean data with data augmentation.

| | Model | Skew Intensity | | | | |
| | | 1 | 2 | 3 | 4 | 5 |
|---|---|---|---|---|---|---|
| SDP | MLE | 0.3500±0.0125 | 0.3649±0.0126 | 0.3870±0.0123 | 0.4186±0.0122 | 0.4561±0.0129 |
| Kernel | MLE | 0.1188±0.0093 | 0.1451±0.0091 | 0.1776±0.0090 | 0.2341±0.0092 | 0.2886±0.0090 |
| | MFVI | 0.0772±0.0048 | 0.0965±0.0025 | 0.1283±0.0052 | 0.1903±0.0067 | 0.2261±0.0090 |
| | MCD | 0.0566±0.0032 | 0.0788±0.0031 | 0.1110±0.0048 | 0.1720±0.0062 | 0.2298±0.0057 |
| | KFLLLA | 0.0808±0.0031 | 0.1035±0.0028 | 0.1328±0.0030 | 0.1845±0.0036 | 0.2396±0.0036 |
| | SNGP | 0.0557±0.0013 | 0.0797±0.0029 | 0.1093±0.0039 | 0.1764±0.0040 | 0.2318±0.0044 |
| | SGPA | 0.0569±0.0022 | 0.0781±0.0026 | 0.1059±0.0030 | 0.1601±0.0054 | 0.2167±0.0050 |
| | DE | 0.0868 | 0.0855 | 0.1003 | 0.1334 | 0.1705 |
| | SGPAE | 0.0627 | 0.0659 | 0.0882 | 0.1262 | 0.1776 |

Table 29: MCE of CIFAR100-C for ViTs trained on clean data without data augmentation.

| | Model | Skew Intensity | | | | |
| | | 1 | 2 | 3 | 4 | 5 |
|---|---|---|---|---|---|---|
| SDP | MLE | 0.6608±0.0014 | 0.6852±0.0014 | 0.7009±0.0022 | 0.7251±0.0020 | 0.7542±0.0011 |
| Kernel | MLE | 0.6037±0.0032 | 0.6428±0.0031 | 0.6649±0.0027 | 0.7037±0.0007 | 0.7385±0.0010 |
| | MFVI | 0.1925±0.0031 | 0.2594±0.0039 | 0.3069±0.0055 | 0.3809±0.0044 | 0.4562±0.0027 |
| | MCD | 0.3424±0.0042 | 0.4103±0.0047 | 0.4554±0.0045 | 0.5255±0.0052 | 0.5902±0.0038 |
| | KFLLLA | 0.7691±0.0165 | 0.7632±0.0181 | 0.7512±0.0194 | 0.7248±0.0146 | 0.6699±0.0228 |
| | SNGP | 0.3631±0.0042 | 0.4201±0.0044 | 0.4624±0.0039 | 0.5177±0.0043 | 0.5749±0.0037 |
| | SGPA | 0.3217±0.0057 | 0.3886±0.0083 | 0.4353±0.0093 | 0.5047±0.0083 | 0.5726±0.0072 |
| | DE | 0.1767 | 0.2257 | 0.2655 | 0.3522 | 0.4202 |
| | SGPAE | 0.1247 | 0.1766 | 0.2209 | 0.3142 | 0.3965 |

Table 30: MCE of CIFAR100-C for ViTs trained on clean data with data augmentation.

| | Model | Skew Intensity | | | | |
| | | 1 | 2 | 3 | 4 | 5 |
|---|---|---|---|---|---|---|
| SDP | MLE | 0.3142±0.0470 | 0.3487±0.0458 | 0.3816±0.0451 | 0.4308±0.0412 | 0.4839±0.0387 |
| Kernel | MLE | 0.2237±0.0136 | 0.2650±0.0122 | 0.3098±0.0131 | 0.3733±0.0117 | 0.4381±0.0140 |
| | MFVI | 0.1330±0.0083 | 0.1173±0.0064 | 0.1275±0.0051 | 0.1751±0.0056 | 0.2208±0.0069 |
| | MCD | 0.0995±0.0066 | 0.1418±0.0067 | 0.1845±0.0067 | 0.2568±0.0081 | 0.3240±0.0082 |
| | KFLLLA | 0.0893±0.0040 | 0.1278±0.0048 | 0.1689±0.0057 | 0.2355±0.0067 | 0.2999±0.0076 |
| | SNGP | 0.0636±0.0012 | 0.0972±0.0013 | 0.1377±0.0015 | 0.2007±0.0017 | 0.2620±0.0012 |
| | SGPA | 0.1120±0.0022 | 0.1582±0.0034 | 0.2062±0.0040 | 0.2776±0.0028 | 0.3455±0.0030 |
| | DE | 0.0920 | 0.0971 | 0.1220 | 0.1906 | 0.2444 |
| | SGPAE | 0.0722 | 0.0888 | 0.1088 | 0.1766 | 0.2291 |

Table 31: AUROC and AUPR metrics for OOD detection using ViTs trained on CIFAR10 without data augmentation.

| | | OOD: CIFAR100 | | OOD: SVHN | | OOD: MINIIMAGENET | |
|---|---|---|---|---|---|---|---|
| | Model | AUROC | AUPR | AUROC | AUPR | AUROC | AUPR |
| SDP | MLE | 0.6893±0.0018 | 0.6433±0.0023 | 0.6983±0.0109 | 0.8206±0.0074 | 0.7115±0.0018 | 0.9219±0.0006 |
| Kernel | MLE | 0.7208±0.0029 | 0.6774±0.0029 | 0.7477±0.0118 | 0.8518±0.0068 | 0.7325±0.0016 | 0.9300±0.0004 |
| | MFVI | 0.7177±0.0055 | 0.6722±0.0050 | 0.7059±0.0195 | 0.7944±0.0134 | 0.7531±0.0036 | 0.9407±0.0008 |
| | MCD | 0.7426±0.0016 | 0.7046±0.0016 | 0.6860±0.0138 | 0.8045±0.0083 | 0.7443±0.0019 | 0.9356±0.0006 |
| | KFLLLA | 0.7228±0.0056 | 0.6889±0.0055 | 0.7599±0.0105 | 0.8345±0.0069 | 0.7855±0.0046 | 0.9487±0.0015 |
| | SNGP | 0.7391±0.0016 | 0.7023±0.0029 | 0.7135±0.0125 | 0.8258±0.0083 | 0.7391±0.0050 | 0.9341±0.0015 |
| | SGPA | 0.7498±0.0010 | 0.7111±0.0023 | 0.7198±0.0085 | 0.8125±0.0067 | 0.7501±0.0013 | 0.9382±0.0004 |
| | DE | 0.7757±0.0000 | 0.7423±0.0000 | 0.7819±0.0000 | 0.8512±0.0000 | 0.8007±0.0000 | 0.9520±0.0000 |
| | SGPAE | 0.7872±0.0000 | 0.7482±0.0000 | 0.7600±0.0000 | 0.8619±0.0000 | 0.8031±0.0000 | 0.9529±0.0000 |

Table 32: AUROC and AUPR metrics for OOD detection using ViTs trained on CIFAR10 with data augmentation.

| | | OOD: CIFAR100 | | OOD: SVHN | | OOD: MINIIMAGENET | |
|---|---|---|---|---|---|---|---|
| | Model | AUROC | AUPR | AUROC | AUPR | AUROC | AUPR |
| SDP | MLE | 0.8245±0.0007 | 0.7780±0.0013 | 0.8738±0.0020 | 0.9308±0.0015 | 0.8375±0.0009 | 0.9585±0.0005 |
| Kernel | MLE | 0.7908±0.0034 | 0.7523±0.0028 | 0.8748±0.0023 | 0.9338±0.0015 | 0.8148±0.0027 | 0.9555±0.0005 |
| | MFVI | 0.7009±0.0043 | 0.6530±0.0045 | 0.7652±0.0179 | 0.8468±0.0109 | 0.7506±0.0073 | 0.9379±0.0027 |
| | MCD | 0.7995±0.0038 | 0.7614±0.0030 | 0.8754±0.0040 | 0.9304±0.0029 | 0.8282±0.0029 | 0.9595±0.0007 |
| | KFLLLA | 0.7989±0.0041 | 0.7617±0.0036 | 0.8884±0.0036 | 0.9415±0.0022 | 0.8273±0.0034 | 0.9591±0.0007 |
| | SNGP | 0.7889±0.0092 | 0.7543±0.0104 | 0.8776±0.0044 | 0.9326±0.0032 | 0.8125±0.0072 | 0.9551±0.0020 |
| | SGPA | 0.8007±0.0014 | 0.7630±0.0013 | 0.8746±0.0061 | 0.9281±0.0043 | 0.8319±0.0014 | 0.9600±0.0004 |
| | DE | 0.8230±0.0000 | 0.7869±0.0000 | 0.9195±0.0000 | 0.9575±0.0000 | 0.8555±0.0000 | 0.9666±0.0000 |
| | SGPAE | 0.8264±0.0000 | 0.7917±0.0000 | 0.9004±0.0000 | 0.9452±0.0000 | 0.8606±0.0000 | 0.9677±0.0000 |

Table 33: AUROC and AUPR metrics for OOD detection using ViTs trained on CIFAR100 without data augmentation.

| | | OOD: CIFAR10 | | OOD: SVHN | | OOD: MINIIMAGENET | |
|---|---|---|---|---|---|---|---|
| | Model | AUROC | AUPR | AUROC | AUPR | AUROC | AUPR |
| SDP | MLE | 0.6091±0.0023 | 0.5756±0.0030 | 0.6242±0.0014 | 0.7856±0.0015 | 0.6512±0.0018 | 0.9042±0.0007 |
| Kernel | MLE | 0.6355±0.0052 | 0.5983±0.0045 | 0.6725±0.0111 | 0.8171±0.0062 | 0.6582±0.0035 | 0.9071±0.0010 |
| | MFVI | 0.6277±0.0036 | 0.5900±0.0037 | 0.6242±0.0108 | 0.7729±0.0079 | 0.6496±0.0054 | 0.9051±0.0024 |
| | MCD | 0.6729±0.0069 | 0.6310±0.0076 | 0.6054±0.0057 | 0.7597±0.0038 | 0.6859±0.0014 | 0.9155±0.0005 |
| | KFLLLA | 0.6297±0.0059 | 0.5920±0.0071 | 0.6733±0.0159 | 0.8101±0.0086 | 0.6915±0.0057 | 0.9194±0.0017 |
| | SNGP | 0.6448±0.0016 | 0.6080±0.0020 | 0.6764±0.0096 | 0.8093±0.0063 | 0.6823±0.0023 | 0.9161±0.0009 |
| | SGPA | 0.6712±0.0052 | 0.6297±0.0057 | 0.6454±0.0147 | 0.7854±0.0099 | 0.6911±0.0028 | 0.9170±0.0009 |
| | DE | 0.6848±0.0000 | 0.6377±0.0000 | 0.7388±0.0000 | 0.8425±0.0000 | 0.7394±0.0000 | 0.9300±0.0000 |
| | SGPAE | 0.6925±0.0000 | 0.6461±0.0000 | 0.6961±0.0000 | 0.8213±0.0000 | 0.7398±0.0000 | 0.9304±0.0000 |

Table 34: AUROC and AUPR metrics for OOD detection using ViTs trained on CIFAR100 with data augmentation.

| | | OOD: CIFAR10 | | OOD: SVHN | | OOD: MINIIMAGENET | |
|---|---|---|---|---|---|---|---|
| | Model | AUROC | AUPR | AUROC | AUPR | AUROC | AUPR |
| SDP | MLE | 0.6778±0.0022 | 0.6312±0.0001 | 0.7652±0.0093 | 0.8769±0.0055 | 0.7430±0.0065 | 0.9346±0.0020 |
| Kernel | MLE | 0.6899±0.0026 | 0.6486±0.0023 | 0.7570±0.0056 | 0.8670±0.0025 | 0.7536±0.0038 | 0.9374±0.0012 |
| | MFVI | 0.6403±0.0042 | 0.5924±0.0035 | 0.7485±0.0106 | 0.8640±0.0068 | 0.7269±0.0099 | 0.9317±0.0030 |
| | MCD | 0.7047±0.0025 | 0.6568±0.0029 | 0.7972±0.0040 | 0.8885±0.0023 | 0.7780±0.0023 | 0.9440±0.0008 |
| | KFLLLA | 0.6924±0.0011 | 0.6510±0.0022 | 0.8164±0.0043 | 0.9104±0.0053 | 0.7726±0.0084 | 0.9435±0.0024 |
| | SNGP | 0.6813±0.0034 | 0.6388±0.0034 | 0.7616±0.0090 | 0.8701±0.0052 | 0.7486±0.0048 | 0.9367±0.0014 |
| | SGPA | 0.7056±0.0012 | 0.6586±0.0011 | 0.8120±0.0035 | 0.8971±0.0027 | 0.7755±0.0012 | 0.9430±0.0004 |
| | DE | 0.7171±0.0000 | 0.6718±0.0000 | 0.8459±0.0000 | 0.9173±0.0000 | 0.8076±0.0000 | 0.9517±0.0000 |
| | SGPAE | 0.7198±0.0000 | 0.6719±0.0000 | 0.8686±0.0000 | 0.9298±0.0000 | 0.8152±0.0000 | 0.9536±0.0000 |

Table 35: AUROC and AUPR metrics for OOD detection using Transformers trained on ZINC.

| | Model | AUROC | AUPR |
|---|---|---|---|
| SDP | MCD | 0.4566±0.0072 | 0.4889±0.0029 |
| Kernel | MFVI | 0.7254±0.0632 | 0.6821±0.0512 |
| | MCD | 0.8797±0.0739 | 0.8629±0.0571 |
| | SGPA | 0.8968±0.0317 | 0.8705±0.0335 |
| | DE | 0.9783±0.0000 | 0.9697±0.0000 |
| | SGPAE | 0.9884±0.0000 | 0.9727±0.0000 |

