# OpenReview forum: "Calibrating Transformers via Sparse Gaussian Processes"
_ICLR.cc/2023/Conference — ICLR 2023 poster_

### Official Review · Reviewer_GcWL · 2022-10-22

**Confidence:** 4
**Correctness:** 3
**Technical Novelty And Significance:** 3
**Empirical Novelty And Significance:** 3
**Recommendation:** 5

**Clarity, Quality, Novelty And Reproducibility:**

The paper is clearly written, but the experiment setup is lacking and the method is not well motivated. The work is somehow novel and the derivation seems alright. As for the reproducibility, it would be nice if the authors could provide an efficient implementation.

**Strength And Weaknesses:**

Strenghts:
- The paper is clearly written, easy to follow
- The inclusion of preliminaries is nicely done and can help people not familiar with the topic
- The method seems to positively affect not only the uncertainty properties but also the accuracies
- The authors perform an adequate selection of the type of experiments to perform
- The idea of developing a computationally efficient GP implementation for attention is interesting

Weaknesses:
- Q1: Some very important baselines seem to be missing, and the used baselines are generally acknowledged to be extremely weak (MCD and MFVI are notorious for being outperformed by much better SOTA methods). I would suggest integrating your experiments (1) for accuracies on in-distribution and covariate-shift datasets, it's important to evaluate against methods that have been observed to produce improved accuracies (and also improved calibration and out-of-distribution performance) like PixMix [1], RegMixup [2], Deep Ensembles [3] (and optionally any of its more computationally efficient variants like Snapshot ensembles [4] or BatchEnsemble), (2) for calibration experiments, as a bare minimum Temperature Scaling [5] should be considered too, and it would be good to compare with more sophisticated calibration techniques too, (3) for out-of-distribution detection performance techniques like G-ODIN [6] and VOS [7]. Also, since your methodology falls in the Bayesian inference with GPs, it would be good to at least compare with KFAC-LLLA [8], but I understand it might be tricky to apply SNGP, DUE and other GP-based techniques to transformers.
- Q2: The motivation of the methodology is not clear. This is a general problem of Bayesian Deep Learning, but it's not clear why having GP on the self-attention should meaningful. While works like SNGP and KFAC-LLLA motivate their usage of (approximations) of GP with a distance-aware uncertainty estimation narrative (though it is questionable that euclidean distances are meaningful at all), I don't feel like the paper provides a clear motivation of why introducing GPs in a few parts of the network should be theoretically meaningful. The paper does not even provide aneddotical evidence that the uncertainty estimates the GP provide in the self-attention can actually detect ambiguous features of the input.
- Important comment on Q1 and Q2: if Q2 cannot be addressed due to the limitations of Bayesian DL, working on Q1 and showing the effectiveness of your method at least empirically would convince me to rise the score.
- Q3: The paper is not really about calibrating for pre-training but for fine-tuning. Modern approaches to fine-tuning are quite sophisticated and might affect uncertainty estimates or the usefulness of the proposed method. Did you consider advanced fine-tuning techniques?

Improvements (Marginal):
- Probably the paper should emphasize more it's about fine-tuning and not really calibrating large-scale pre-training
- The NLL is known to be a questionable metric for calibration, but a lot of literature still reports it



[1] https://arxiv.org/abs/2112.05135
[2] https://arxiv.org/abs/2206.14502
[3] https://arxiv.org/abs/1612.01474
[4] https://arxiv.org/abs/1704.00109
[5] https://arxiv.org/abs/1706.04599
[6] https://arxiv.org/abs/2002.11297
[7] https://arxiv.org/abs/2202.01197
[8] https://arxiv.org/abs/2002.10118

**Summary Of The Paper:**

The authors suggest an efficient implementation of GP for the attention layers of transformers. Through a few experiments they argue it helps in improving the accuracy, the covariate-shift robustness, the calibration and the out-of-distribution detection performance when fine-tuning transformers on smaller scale datasets.

**Summary Of The Review:**

The paper is interesting, but it either needs a better motivation or much stronger experimental evidence. If either (or both) these issues are improved, I am happy to rise the score.

--------------------
POST-REBUTTAL AND DISCUSSION WITH OTHER REVIEWERS
I increase my score to 5.

- Although it is difficult to interpret the meaning of GPs over self-attention, the additional comparisons with strong baselines (both deterministic and non-deterministic) like TS, KFAC-LLLA, SNGP and Deep Ensembles shows the method is at least empirically effective
- The justification provided for the usage of GPs on self-attention is intuitive, and although still not completely supported and questionable given the lack of an understanding of how uncertainties compose through the layers of a network in the presence of additional components (normalization layers, non-linearities, skip-connections etc.), I think it would be unfair considering this to be a limitation of this specific work, as it is a known problem of Bayesian Deep Learning.
- The usefulness of the method might depend on the fact Transformers don't converge to a good minimum without enough pre-training data on such small datasets. The technique can be considered effective probably also because it somehow regularizes the training procedure. Therefore, the improve the impact of the paper, it would be important to show the proposed method can be effective also when fine-tuning pre-trained models.

---

> ### Author Response · Authors · 2022-11-10
> **Response (part 1/4)**
>
> Thank you for your review and suggestions on our work. We would like to address your questions/concerns below and are keen to follow up if you have any further questions.
>
> > 1. On your request for more baselines
>
> Here we include test results for CIFAR10 from some of the baselines the reviewer mentioned. Namely temperature scaling (only for in-distribution calibration), KFAC-LLLA, SNGP and deep ensemble. Note that multiple SGPA models can also be ensembled and it turns out SGPA-ensemble tends to achieve the best performance. In this case, we draw 2 samples from each of the 5 SGPA models to still keep the number of total samples to 10 for fair comparison with other single-model baselines requiring sampling such as single SGPA, MCD and MFVI. Due to the time constraint of rebuttal, we will not able to obtain results from additional baselines for other experiments in the paper, but we will update the paper with these additional baselines for all experiments in the future.
>
> We highlight in bold the top 2 methods from the “single model” category and highlight the best ensemble method.
>
> **In-distribution calibration:**
> |Method|Accuracy|NLL|ECE|MCE|
> |---|---|---|---|---|
> |MLE|0.7811 (0.0019)|1.3395 (0.0135)|0.0331 (0.0004)| 0.3724 (0.0103)|
> |MFVI|0.7202 (0.0024)|0.7995 (0.0073)|**0.0058 (0.0009)**|**0.0664 (0.0064)**|
> |MCD|**0.7910 (0.0015)**|0.7789 (0.0063)|0.0159 (0.0004)|0.1501 (0.0055)|
> |Temerature Scaling|0.7811 (0.0021)|0.8254 (0.0077)|0.0100 (0.0004)|0.2499 (0.0081)|
> KFAC-LLLA |0.7752 (0.0019)|**0.7201 (0.0242)**|0.0151 (0.0020)|0.1497 (0.0216)|
> |SNGP|**0.7837 (0.0025)**|0.7534 (0.0031)|0.0156 (0.0001)|0.1443 (0.0049)|
> |**SGPA**| 0.7787 (0.0024)|**0.6968 (0.0032)**|**0.0100 (0.0004)**|**0.0825 (0.0066)**|
> |Deep Ensemble|**0.8235**|0.6934|0.0110|0.0699|
> |**SGPA Ensemble**|0.8172|**0.5657**|**0.0054**|**0.0579**|
>
> **OOD robustness on CIFAR10-C:**
>
> **Accuracy**
> |Skew intensity|1 |2|3|4|5|
> |---|---|---|---|---|---|
> |MLE|0.7254 (0.0019)|0.6632 (0.0028)|0.6139 (0.0023)|0.5494 (0.0028)|0.4863 (0.0027)|
> |MFVI|0.6606 (0.0093)|0.6109 (0.0083)|0.5703 (0.0077)|0.5266 (0.0073)|0.4778 (0.0065)|
> |MCD|**0.7327 (0.0013)**|**0.6717 (0.0015)**|**0.6229 (0.0016)**|**0.5603 (0.0019)**|**0.4982 (0.0020)**|
> |KFAC-LLLA|0.7186 (0.0031)|0.6580 (0.0078)|0.6092 (0.0085)|0.5462 (0.0094)|0.4825 (0.0085)|
> |SNGP|**0.7281 (0.0053)**|**0.6676 (0.0055)**|**0.6208 (0.0044)**|**0.5622 (0.0045)**|**0.5006 (0.0042)**|
> |**SGPA**|0.7175 (0.0033)|0.6559 (0.0042)|0.6067 (0.0040)|0.5466 (0.0033)|0.4832 (0.0046)|
> |Deep ensemble|**0.7700**|**0.7089**|**0.6585**|**0.5908**|**0.5246**|
> |**SGPA ensemble**|0.7516|0.6911|0.6401|0.5779|0.5173|
>
> **NLL**
> |Skew intensity|1|2|3|4|5|
> |---|---|---|---|---|---|
> |MLE|1.7906 (0.0179)|2.3460 (0.0321)|2.8899 (0.0385)|3.6899 (0.0544)|4.5003 (0.0811)|
> |MFVI|0.9761 (0.0274)|**1.1345 (0.0267)**|**1.2737 (0.0259)**|**1.4705 (0.0237)**|**1.7038 (0.0222)**|
> |MCD|1.1466 (0.0099)|1.5153 (0.0168)|1.8926 (0.0234)|2.4692 (0.0315)|3.1086 (0.0431)|
> |KFAC-LLLA|**0.9573 (0.1113)**|**1.2092 (0.1291)**|**1.4435 (0.1526)**|**1.7889 (0.2095)**|**2.1537 (0.2679)**|
> |SNGP|1.0579 (0.0147)|1.3681 (0.0257)|1.6570 (0.0289)|2.0807 (0.0305)|2.5271 (0.0326)|
> |**SGPA**|**0.9603 (0.0128)**|1.2340 (0.0213)|1.4972 (0.0283)|1.8951 (0.0383)|2.3812 (0.0586)|
> |Deep Ensemble|0.9317|1.2424|1.5716|2.0911|2.6373|
> |**SGPA Ensemble**|**0.7816**|**0.9985**|**1.2125**|**2.5365**|**1.9349**|
>
> **ECE**
> |Skew intensity|1|2|3|4|5|
> |---|---|---|---|---|---|
> |MLE|0.0425 (0.0003)|0.0530 (0.0005)|0.0617 (0.0005)|0.0734 (0.0006)|0.0846 (0.0006)|
> |MFVI|**0.0077 (0.0017)**|**0.0122 (0.0021)**|**0.0170 (0.0021)**|**0.0243 (0.0021)**|**0.0316 (0.0021)**|
> |MCD|0.0242 (0.0002)|0.0328 (0.0003)|0.0405 (0.0004)|0.0515 (0.0004)|0.0621 (0.0006)|
> |KFAC-LLLA|0.0226 (0.0060)|0.0305 (0.0062)|0.0374 (0.0063)|0.0472 (0.0069)|0.0566 (0.0076)|
> |SNGP|0.0233 (0.0004)|0.0317 (0.0006)|0.0389(0.0006)|0.0487 (0.0005)|0.0588 (0.0006)|
> |**SGPA**|**0.0178 (0.0006)**|**0.0258 (0.0007)**|**0.0330 (0.0008)**|**0.0432 (0.0008)**|**0.0529 (0.0010)**|
> |Deep Ensemble|0.0160|0.0227|0.0291|0.0389|0.0484|
> |**SGPA Ensemble**|**0.0088**|**0.0136**|**0.0191**|**0.0282**|**0.0368**|
>
> **MCE**
> |Skew intensity|1|2|3|4|5|
> |---|---|---|---|---|---|
> |MLE|0.4038 (0.0028)|0.4362 (0.0020)|0.4655 (0.0020)|0.5051 (0.0033)|0.5442 (0.0036)|
> |MFVI|**0.0598 (0.0112)**|**0.0852 (0.0150)**|**0.1125 (0.0147)**|**0.1586 (0.0147)**|**0.2011 (0.0149)**|
> |MCD|0.1902 (0.0021)|0.2402 (0.0025)|0.2807 (0.0026)|0.3413 (0.0036)|0.3949 (0.0041)|
> |KFAC-LLLA|0.1987 (0.0561)|0.2362 (0.0539)|0.2729 (0.0533)|0.3248 (0.0493)|0.3733 (0.0470)|
> |SNGP|0.1854 (0.0041)|0.2337 (0.0046)|0.2710 (0.0024)|0.3229 (0.0046)|0.3798 (0.0018)|
> |**SGPA**|**0.1273 (0.0042)**|**0.1801 (0.0046)**|**0.2212 (0.0061)**|**0.2807 (0.0057)**|**0.3340 (0.0047)**|
> |Deep Ensemble|0.1051|0.1402|0.1813|0.2476|0.3091|
> |**SGPA Ensemble**|**0.0570**|**0.0816**|**0.1173**|**0.1757**|**0.2208**|

---

> ### Author Response · Authors · 2022-11-10
> **Response (Part 2/4)**
>
> **OOD detection (OOD set: CIFAR100, SVHN, MINIIMAGENET):**
> |Method|AUROC (C100)|AUPR (C100)|AUROC (SVHN)|AUPR (SVHN)|AUROC (MINI)|AUPR (MINI)|
> |---|---|---|---|---|---|---|
> |MLE| 0.7208 (0.0029)|0.6774 (0.0029)|**0.7477 (0.0118)**|**0.8518 (0.0068)**|0.7325 (0.0016)|0.9300 (0.0004)|
> |MFVI|0.7177 (0.0055)|0.6722 (0.0050)|0.7059 (0.0195)|0.7944 (0.0134)|**0.7531 (0.0036)**|**0.9407 (0.0008)**|
> |MCD|**0.7426 (0.0016)**|**0.7046 (0.0016)**|0.6860 (0.0138)|0.8045 (0.0083)|0.7443 (0.0019)|0.9356 (0.0006)|
> |KFAC-LLLA|0.7228 (0.0056)|0.6889 (0.0055)|**0.7599 (0.0105)**|**0.8345 (0.0069)**|**0.7855 (0.0046)**|**0.9487 (0.0015)**|
> |SNGP|0.7391 (0.0016)|0.7023 (0.0029)|0.7135 (0.0125)|0.8258 (0.0083)|0.7391 (0.0050)|0.9341 (0.0015)|
> |**SGPA**|**0.7498 (0.0010)**|**0.7111 (0.0023)**|0.7198 (0.0085)|0.8125 (0.0067)|0.7501 (0.0013)|0.9382 (0.0004)|
> |Deep Ensemble|0.7757|0.7423|**0.7819**|0.8512|0.8007|0.9520|
> |**SGPA Ensemble**|**0.7872**|**0.7482**|0.7600|**0.8619**|**0.8031**|**0.9529**|
>
> Overall, SGPA-ensemble achieves the best performance in all tasks (best calibration and OOD detection performance, and higher predictive accuracy). Within “single-model” category, SGPA achieves the best in-distribution and OOD robustness performance in the sense that it consistently outperforms MLE, MCD, temperature scaling, KFAC-LLLA and SNGP in terms of calibration, while unlike MFVI, it doesn’t suffer from underfitting. For OOD detection, SGPA achieves the best performance when OOD set is CIFAR100, while KFAC-LLLA achieves the best performance when OOD set is SVHN and MINIIMAGENET.
>
> Indeed, empirical comparison between as many baselines as possible would help fully assess the method. But we would like to point out that the goal of this paper is mostly to propose a new probabilistic method based on our identification of the natural connection between attention and SVGP. The method is novel in the sense that we are not improving over existing methods, instead to our knowledge, this is the first paper establishing the connection between attention and posterior mean of SVGP and applying GP-based approaches to transformers.

---

> ### Author Response · Authors · 2022-11-10
> **Response (Part 3/4)**
>
> > 2. The motivation of the methodology is not clear. This is a general problem of Bayesian Deep Learning, but it's not clear why having GP on the self-attention should meaningful.
>
> Thank you for bringing up this open question. Here we present the motivation of this work, which may also reveal our humble opinions on BDL.
>
> Exact Bayesian inference is a principled framework for uncertainty estimation and is useful for decoupling epistemic/model uncertainty (which can be explained away with an infinite amount of data) and aleatoric/data uncertainty (which is consistent in the data generating process) [1].
>
> The problem of applying Bayesian inference to deep neural networks lies in intractable inference. In BDL, we can only work with approximate Bayesian inference, and the performance of approximate inference highly relies on the specification of the prior distribution and the approximate inference method chosen. Unfortunately, it is a significant challenge to specify meaningful weight space priors for BNNs: it is unclear what inductive bias the implicit function space prior induced by the weight space prior represents [2,3]. Moreover, the non-identifiability of neural networks (distinct modes in weight space may even correspond to the same function) also poses challenges over weight-space inference [4]. Many popular weight space approximate inference methods, such as MFVI and Laplace, can only capture one mode in weight space, which fails to represent the diversity of the functions. This motivates us to bring inference directly over the functions (i.e. inference in output or hidden feature space), and GP is one of the most convenient and well-studied tools for function space inference.
>
> For transformers, since we establish the natural connection between the output of attention and posterior mean of SVGP, it is natural to utilize this insight to carry out function space Bayesian inference for Transformers via GP as a first step. In SGPA, the global keys $\\{k^{l,h}_g\\}_\{l=1,h=1\}^\{L,H\}$, shared across all input sequences, play a similar role as the inducing locations in standard SVGP: they summarise the training set but focus on the uncertainty behavior only. As a result, the posterior variance in Eq (11) will increase for queries that are less similar to the global keys as measured by the kernel. By propagating the uncertainty through each layer, we can still obtain increased uncertainty for input sequences that are very different from the training data, so that users can still be notified “when the model does not know” [1]. The problem is with the deep kernel, it’s less obvious how it measures similarity and some advantages of Bayesian inference still lose as we have no idea what prior/inductive bias the deep kernel represents. Therefore, the next step will be injecting meaningful inductive bias into deep kernels. For datasets where Euclidean distance is meaningful, we can adopt the distance-preserving techniques in SNGP [5] to enforce inductive bias into deep kernels. However, for other types of datasets, it still remains an interesting open question that we are keen to explore.
>
> Still, unlike many other heuristic methods tailored to each application, such as OOD detection, the uncertainty yielded from Bayesian method can be used in any applications requiring decision-making under uncertainty (not restricted to OOD detection), such as active learning [6], continual learning [7], and Bayesian optimization [8].
>
> References
>
> [1] “Uncertainty in Deep Learning”, Yarin Gal, PhD dissertation, University of Cambridge, 2016
>
> [2] “Functional Variational Bayesian Neural Networks”, ICLR 2019
>
> [3] “Functional Variational Inference based on Stochastic Process Generators”, NeurIPS 2021
>
> [4] “Function Space Particle Optimization for {B}ayesian Neural Networks”, ICLR 2019
>
> [5] “Simple and Principled Uncertainty Estimation with Deterministic Deep Learning via Distance Awareness”, NeurIPS 2020
>
> [6] “Deep Bayesian Active Learning with Image Data”, ICML 2017
>
> [7] ”Variational Continual Learning”, ICLR 2018
>
> [8] “Bayesian Optimization with Robust Bayesian Neural Networks”, NeurIPS 2016

---

> ### Author Response · Authors · 2022-11-10
> **Response (Part 4/4)**
>
> > 3. The paper is not really about calibrating for pretraining but for fine-tuning. Modern approaches to fine-tuning are quite sophisticated and might affect uncertainty estimates or the usefulness of the proposed method. Did you consider advanced fine-tuning techniques?
>
> When talking about fine-tuning, one may think (part of) the weights of the model is initialized using a pretrained backbone, which is not the case here. We would like to clarify that we directly train from scratch in all the experiments (i.e. random initialization of all parameters), and we didn’t consider advanced fine-tuning techniques. We would like to point out that SGPA can be directly used in either or both pretraining and fine-tuning stage. Due to the limit of computational resources, we are not able to consider experiments on large-scale pretraining, but using SGPA to calibrate pre-training is definitely an interesting future direction. We refer the reviewer to our answer to Q5 from reviewer o7ME for experimental details, and to our answer to Q3 from reviewer 3Ji8 for more details of training transformers from scratch in small data regime: in this scenario, transformers tend to perform much worse than other CNN-based architectures, that's why we did not achieve SOTA results one would see in the pretraining scheme. However, our results are competitive or improved against results reported in other papers considering training transformers from scratch on small datasets [1,2,3].
>
> > 4. The NLL is known to be a questionable metric for calibration, but a lot of literature still reports it.
>
> NLL is a proper scoring rule so in theory it is a valid measure for calibration. But to circumvent the potential preferential bias of NLL towards certain calibration behavior, we also reported other measures such as ECE, MCE, where all of them are designed to be (in line with) a proper scoring rule. We believe that we can trust the experimental results as multiple scoring rules agree on the ranking of models in terms of calibration.
>
>
> References
>
> [1] “How do Vison Transformers Work?”, ICLR 2022
>
> [2] “Locality Guidance for Improving Vision Transformers on Tiny Datasets”, ECCV 2022
>
> [3] “Nested Hierarchical Transformer: Towards Accurate, Data-Efficient and Interpretable Visual Understanding”, AAAI 2022

---

> > ### Author Response · Authors · 2022-11-16
> > **Reminder for Reviewer GcWL**
> >
> > Dear Reviewer GcWL,
> >
> > Thank you once again for your review of our work. As the discussion period is approaching its end, we would be grateful if you could confirm whether our rebuttal addressed your concerns, and let us know if any issues remain.

---

> > > ### Comment · Reviewer_GcWL · 2022-11-16
> > > **Thanks for your responses**
> > >
> > > Thank you for addressing most of my concerns. Most of the experimental results look fine, the ones in Part 2/4 look a bit concerning, but ensembling seems to fix the issue (although ensembling has the problem of being expensive).
> > >
> > > A few comments on the response about Bayesian Deep Learning:
> > > - Exact Bayesian Inference is principled as long as its priors and likelihood (and the models on which they are cast) encode some beliefs that have some meaning. BDL is known to have the problem it is not clear what the priors and likelihoods are supposed to represent given the underlying model (Neural Networks) are difficult to interpret. Therefore exact inference is not the only problem of BDL, and you also cite references about this when talking about weight-space priors and inference. I would not start discussing about aleatoric and epistemic uncertainty, as there is no real discussion of it in the paper you submitted and it is a delicate topic.
> > > - I think the discussion you make about SVGP and SGPA is reasonable, and enough to justify the approach. I think it should be included in the revision of the paper.
> > >
> > > I will update the score accordingly.

---

> > > > ### Author Response · Authors · 2022-11-16
> > > > **Thank you for the response and suggestions**
> > > >
> > > > Thank you very much for the response and suggestions.
> > > >
> > > > - Indeed, for weight-space BDL, it is hard to specify meaningful prior. For GP, traditionally our prior belief on the behavior of the functions is represented by the kernel (for example, the functions are believed to be smooth when RBF kernel is used). However, with deep kernel, it is still hard to understand what prior it represents. That's why we seek to inject meaningful inductive bias tailored to the problem into the deep kernel in our next step.
> > > >
> > > > - We have updated the revision of the paper, and the discussion about SVGP and SGPA is now highlighted in blue in Appendix C. We will put this discussion in the main text, if the paper is accepted. However, at this stage, we prefer to leave the main text mostly unchanged to avoid confusion.

---

> > > > > ### Author Response · Authors · 2022-11-18
> > > > > **End of Rebuttal Period**
> > > > >
> > > > > Dear reviewer GcWL,
> > > > >
> > > > > We would like to thank you for spending your time carefully evaluating our submission, providing valuable feedback, and engaging in discussion with us.
> > > > >
> > > > > As the rebuttal period is expected to conclude today, we look forward to hearing your feedback if you have any further requests for changes to the revision. We hope that you can update your rating in light of our responses and changes to the paper.
> > > > >
> > > > > Best regards, Authors

---

> > > > ### Author Response · Authors · 2022-11-25
> > > > **Post Rebuttal Reminder**
> > > >
> > > > Dear Reviewer GcWL,
> > > >
> > > > Thank you once again for your response. According to your response, it seems we have addressed most of your concerns and you are happy to update the score. We also added our discussion about the motivation of our method in the revision as requested. We hope that you can reconsider your rating in light of our responses and changes to the paper.
> > > >
> > > > Best regards,
> > > > Authors

---

### Official Review · Reviewer_o7ME · 2022-10-24

**Confidence:** 4
**Correctness:** 3
**Technical Novelty And Significance:** 3
**Empirical Novelty And Significance:** 3
**Recommendation:** 6

**Clarity, Quality, Novelty And Reproducibility:**

**Clarity & Quality:** The paper is clear in general, providing a great review/perspective of the Transformers architectures. On the quality side, all decision are properly justified as well as the experimental results seems rigorous and sufficient to me.

**Novelty & Reproducibility:** The paper seems novel in the way that it aims to improve performance of Transformers with uncertainty estimation via GPs. Perhaps, the use of SVGPs is not particularly novel, but its use in this context for the attention products is justified and significant. (This may open a door to other ideas or probabilistic approaches in the future).

The reproducibility is perhaps another weak point in this submission, or at least, I perceive that are quite a lot of missing details on the initialization, setup of key/inducing-points, need or no-need of pretraining, optimization, limitations on dimensionality, convergence and time metrics. I say this because SVGPs usually struggle on these points and extra efforts/tricks are required. (I.e. one can easily find details, explanations and discussion on these aspects in GP submissions). Here, no particular details are added, and I think they are somehow necessary.

**Strength And Weaknesses:**

**Strengths:** The main strength of the paper is on the identifiable equivalences. Find the simplification of transformers models as certain parallelisation of multi-head self-attention modules, where some scaled dot-product operations are similar to the core linear-operators of GPs when considering kernels is an important point. Additionally, exploiting this equivalence and building a new uncertainty estimation framework on top of the transformer's architecture is also nice. The paper provides a lot of details to make sure that is clear where the linking points between SVGPs and Transformers join. Experimental results are somehow clear and provide a fair comparison which the chosen baselines.

**Weaknesses:** In my opinion, there are perhaps several equivalences and connections (i.e. with multi-output GPs and Deep GPs) that are introduced but not really developed or consistently analysed. For example in the paragraph before Section 3.2, the following is added:

> (...) multi-output SVGP (...) and the output of a kernel attention block

I see and appreciate the similarity between K-attention and the posterior mean of SVGP, but the MO-SVGP connection with multi-head attention is not entirely clear to me if I think in papers like (Alvarez, 2008). What I want to remark is that MOGP with SVGPs seems a bit more complicated to me, and does not seem that easy to say that both are equivalent... At least I would like to see this justified.

Similarly, for Deep GPs, the connection is done, but no experiments or further derivation is added on this line. That for me is a flaw of the paper, which I think is not that difficult to solve...

**Questions:**

*[Q1].* How expensive is to train wrt Eq. (14). The use of Monte-Carlo in expectations for SVGPs is in general not a good idea (long training, not very scalable wrt dimensionality, noisy evaluation on the likelihood model), so I am trying to figure out what would be the effect here? Could the authors clarify a bit on this point?

*[Q2].* Thinking in terms of how difficult is to bring posterior computations to standard NNs, and how the "Gaussian nature" is broken in Deep GPs when one layer after the other is connected passing GPs through, I am somehow surprised that SVGPs fit that well in multiple layers of the Transformer without any of these problems. Are non-linearities or multiple layers affecting the approximation considered?

**References:**

(Alvarez, 2008) https://proceedings.neurips.cc/paper/2008/file/149e9677a5989fd342ae44213df68868-Paper.pdf


**Summary Of The Paper:**

The paper presents a new methodology to introduce uncertainty quantification in Transformer models. This probabilistic approach is based on Gaussian processes. The driving idea is to exploit the similarity between the scaled dot-product (Vaswani, 2017) and the Kernel-attention (Tsai, 2019) with the posterior mean of sparse Gaussian processes (SVGP), to obtain posterior approximations in the attention heads of transformers. Despite the computational cost, which is reduced using global and amortized "keys/inducing-points", the methodology shows a significant performance which is compared with other baselines based on probability in several datasets for different tasks.

**Summary Of The Review:**

The paper is valuable and relevant for the community, as it builds a new door to introduce probabilistic methods into Transformers via the equivalences in kernel-attention heads. The methods brought are perhaps revisited and there are not of extreme novelty, but the ideas presented for the solution, the equivalences and their connection are still important. The experiments are thorough and I believe the results presented. However, there are some weaknesses and missing details that make me decrease my score for strong acceptance to weak acceptance. If these ones are clarified, I would be glad to raise my score.

---

> ### Author Response · Authors · 2022-11-10
> **Response (part 1/2)**
>
> Thank you for your encouraging review. We will address your questions below.
>
> > 1. The MO-SVGP connection with multi-head attention is not entirely clear to me if I think in papers like (Alvarez, 2008).
>
> We agree that the use of the term “MO-SVGP” may be misleading here, since one would naturally think we also consider correlation between each dimension of the attention output, which is not the case here. For each dimension ($d$) of the attention output, we independently fit an SVGP according to Eq (11): in each layer, both prior and posterior of $F^l$ (which is multi-dimensional) factorize between dimensions as in [1]. We will update the paper to clarify that in the future.
>
> > 2. Similarly, for Deep GPs, the connection is done, but no experiments or further derivation is added on this line.
>
> We would like to clarify that most of our experiments consider fitting deep GPs and more details and derivations can be found in **Appendix A.3**. An $L$-layer Transformer based on SGPA fits a deep GP (with depth $L$), and we train it using the doubly stochastic variational inference framework [1]: in each layer $l$, we generate samples using reparameterization trick (as illustrated in Figure 1 (c)) and these samples are treated as the inputs for the GP in next layer.
>
> Except for sentiment analysis, where we only use 1 layer, all the other experiments consider multiple layers and therefore require fitting deep GPs (details of architecture hyperparameters for the experiments, such as the number of layers, can be found in **Appendix D** Experimental Details).
>
> > 3. How expensive is to train wrt Eq. (14). The use of Monte-Carlo in expectations for SVGPs is in general not a good idea (long training, not very scalable wrt dimensionality, noisy evaluation on the likelihood model), so I am trying to figure out what would be the effect here? Could the authors clarify a bit on this point?
>
> Unlike traditional deep GP, where in each layer the inducing points are independent of previous layer’s output, in SGPA, the amortized inducing points at layer $l$ ($k_a^{l,h}$) depend on the output from layer $l-1$ ($F^{l-1}$). Therefore, we are not able to work out the marginal distribution $q(F^l)$ at each layer analytically. In particular,
> $q(F^l|F^0, \\{ k_g^{j,h} \\}_{j=1, h=1}^{l,H})$ (where $F^0 = X$) in Eq (14) is not analytically tractable so we have to resort to iterative sampling through layers to draw samples from it as in [1].
>
> Within all experiments we consider, for each input sequence in a batch, we only draw one sample to estimate the ELBO (Eq 14). We understand the training for SVGP with deep kernels might not be stable [2]. However, we observe no optimization issue: the ELBO is consistently minimized during the training without significant spikes.
>
> There are some subtle differences between SGPA and traditional SVGP: for traditional SVGP, the inducing points are defined in the space related to the inputs, while for SGPA, each input is a sequence and the inducing points are defined in the space related to the tokens in the input sequence. Furthermore, in standard SVGP the inducing locations and variational parameters are free parameters, while for SGPA, the inducing locations and variational parameters of amortized inducing points are input-dependent. Our conjecture is that these differences might cause different behavior for SGPA from standard deep kernel SVGP, which may include optimization stability. Still, it is an open question and we consider it as an interesting future work.

---

> ### Author Response · Authors · 2022-11-10
> **Response (part 2/2)**
>
> > 4.  Difficulty in bringing posterior computations to standard NNs. Broken "Gaussian nature" in Deep GPs. Are non-linearities or multiple layers affecting the approximation considered?
>
> This is a very interesting question but first we would like to clarify the setups:
>
> Posterior computations in standard NNs: typical Bayesian neural networks methods mainly consider posterior computation for network weights, which has its own challenges but is less relevant to the GP-based methods (which compute posteriors for functions).
>
> Indeed Deep GPs have “broken Gaussian nature”, but the ability to modelling non-Gaussian behaviour is exactly the point of this type of model. The doubly stochastic variational inference (DSVI) approach [1] is a well-known variational sparse approximation method to fit scalable Deep GPs, and our approach uses DSVI for approximate posterior inference.
>
> In terms of why our approach works, we would like to highlight that our approach can be viewed as building a (sparse) Deep GP with deep kernels, and this brings in the best of both worlds.
>
> Deep GP allows modelling non-Gaussian behaviour in a stochastic process sense, however traditionally the kernels in use are rather limited (e.g., RBF kernel). Instead, in our case each SGPA layer contains an MLP before attention, which means the kernel is “deep” and thus much more flexible.
>
> On the other hand, deep kernel learning [3], although benefiting from flexible kernel design, still assumes a GP model and therefore cannot handle non-Gaussian behaviour efficiently.
>
> > 5. Reproducibility
>
> Here we clarify these implementation details and we will include them in the appendix in the future update.
>
> **Initialization:** Apart from the global inducing points parameters, the rest of the parameters are the same as in standard transformers, and are initialized via the default method of the deep learning platform (here we use Pytorch). Each dimension of global inducing locations and the global variational mean are randomly initialized with standard Gaussian. For the Cholesky factor used to parameterize the global variational covariance, we randomly initialize each element in the lower triangular part and each element in the log-diagonal with a standard Gaussian.
>
> **Pretraining:** We train all our models from scratch (i.e. all parameters are randomly initialized without any pretraining). We refer the reviewer to our answer to Q3 from reviewer 3Jj8 for more details of training transformers from scratch in small data regime.
>
> **Dimensionality:** Typically, in high dimension, one needs more inducing points to cover the input space for SVGP to work well. However, since in SGPA, the deep GP is defined over the space of tokens instead of the input sequences, empirically we found setting the number of inducing points for each head to be $O(\frac{T_{avg}}{H})$ (where $T_{avg}$ is the average number of tokens in the input sequences, and $H$ is the number of heads) suffice for good performance of SGPA. It could be because the variability in the token space is less than the variability in the input sequence space. Also for ViT, each token is a patch taken from the image and the dimensionality of each patch is much smaller than the dimensionality of the image.
>
> **Optimization and Convergence:** We use Adam optimizer and as mentioned in Q3, we observe no difficulty in optimization. The ELBO (Eq 4) is consistently minimized until convergence without significant spikes during the training.
>
> **Time metrics:** Here we present the training time (in s) of one epoch for SGPA and MLE on CoLA (batch size = 32) and CIFAR10 (batch size = 100)  (results obtained using a single Nvidia GTX 2080 Ti GPU card):
>
> CoLA:
> |Model|MLE|SGPA|
> |---|---|---|
> |Time (in s)|17.834|20.089|
>
> CIFAR10:
> |Model|MLE|SGPA|
> |---|---|---|
> |Time (in s)|30.593|138.609|
>
> The computational cost depends on the number of global inducing points used. For CoLA, we only used 5 global inducing points for each head while for CIFAR10, we use 32 global inducing points for each head. However, we haven’t done extensive tuning for the number of global inducing points. It is possible that SGPA can still work well with an even smaller number of global inducing points. For example, if we reduce the number of global inducing points from 32 to 16 for CIFAR10, we can reduce the training time for one epoch from 138.609s to 109.298s, while maintaining comparable test set performance (accuracy: 0.7790, NLL: 0.7259, ECE: 0.0119, MCE: 0.0819).
> We refer the reviewer to our answer to Q5 from reviewer PkUZ for inference time. These running time metrics are added to **Appendix E** in the revision.
> Some additional details of the experiments can be found in **Appendix D**. Please let us know if you need clarification.
>
> References
>
> [1] “Doubly Stochastic Variational Inference for Deep Gaussian Processes”, NeurIPS 2017
>
> [2] “The promises and pitfalls of deep kernel learning”, UAI 2021
>
> [3] "Deep Kernel Learning", AISTATS 2016

---

> > ### Author Response · Authors · 2022-11-16
> > **Reminder for Reviewer o7ME**
> >
> > Dear Reviewer o7ME,
> >
> > Thank you once again for your review of our work. As the discussion period is approaching its end, we would be grateful if you could confirm whether our rebuttal addressed your concerns, and let us know if any issues remain.

---

> > > ### Author Response · Authors · 2022-11-18
> > > **Reminder for Reviewer o7ME (2)**
> > >
> > > Dear Reviewer o7ME,
> > >
> > > As the discussion period is expected to conclude today. We are therefore wondering if there is still any chance for you to let us know whether we have addressed your concerns with our rebuttal. We would very much appreciate it. Thank you!

---

> ### Author Response · Authors · 2022-11-25
> **Post Rebuttal Reminder**
>
> Dear Reviewer o7ME,
>
> Thank you once again for your review. We believe we have clarified most of the questions you raised and added more implementation details as requested to the revision. Since you mentioned you are happy to update your score according to the rebuttal, we would be grateful if you could confirm whether our rebuttal addressed your concerns. Thank you!
>
> Best regards,
> Authors

---

> ### Comment · Reviewer_o7ME · 2022-12-11
> **After rebuttal update**
>
> Thanks to the authors for their effort on the rebuttal, the long and clarifying answers to my questions and the update of the manuscript/appendix. To me, the main answers to my concerns around the probabilistic details of the method and GPs were satisfactory.
>
> However, there are additional details raised by other reviewers around the limitations of the work or at least, the lack of evaluation in certain scenarios related to transformers (i.e. use of pre-training). For that reason, I keep my score as it is now, as I think that the paper is worth to be accepted. This opinion has been also considered in the discussion between the AC and reviewers.

---

### Official Review · Reviewer_3Jj8 · 2022-10-24

**Confidence:** 4
**Correctness:** 4
**Technical Novelty And Significance:** 3
**Empirical Novelty And Significance:** 3
**Recommendation:** 3

**Clarity, Quality, Novelty And Reproducibility:**

The paper is generally clear and well written. The experiments are sounds, but rather limited in terms of models to which the approach is compared and the model is not evaluated on a task where state of the are performance of vision transformers is know. In terms of novilty, the approach is to some degree novel, but related to the work on kernel attention and set transformers.

**Strength And Weaknesses:**

**Strengths:**

 * The paper is mostly clearly written and the motivation of the method is easy to follow and explained well.
 * The approach is to some degree new and the experiments look promising.

**Weaknesses:**

 * In the introduction of the method, it would be beneficial to differentiate the application of Gaussian processes for interpolation compared to classification. Both approaches are common in machine learning, yet in one case the goal is to condition on the observations of the instance in order to predict values at unknown query locations whereas a classification GP predicts the output of an instance on the test data, while conditioning on the training data. The former is closer to what the authors are proposing in their work.
 * The work does not compare against approaches relying on deterministic uncertainty estimates while citing papers from that field (for instance Liu et al., NeurIPS 2020). This is particularly relevant in the OOD experiment.
 * The performance of all models is relatively low on the datasets presented. For instance, 77% accuracy is far from SOTA on CIFAR10. This questions the usefulness of the experiments on these data. While the trends compared to baseline approaches are generally ok, it is difficult to rely on these results when models are not applied on dataset/model combinations where there is existing data. I would suggest to at least run one comparison on imagenet, where sufficient performance measurements of vision transformers are available in other papers.
 * The approach is very similar to Set Transformers, which use a similar idea to speed up the computations. A reference would be appropriate.
 * Utilization of `[]` in the equations is slightly confusing. I would recommend to discard these or indicate that they solely are used to keep the subscripts separated in the text.

**Summary Of The Paper:**

The paper proposes to formulate the mult-head attention operation used in transformer models as a sparse gaussian process. The work shows how the computation of such a GP attention mechanism can be sped up by relying on inducing points and standard approaches for increasing Gaussian process scalability. Multiple such attention operations can be stacked by sampling between layers. In the experiments of the paper, the proposed model shows comparable performance in terms of accuracy, but scores better in terms of NNL and calibration metrics on small vision and graph datasets. The baselines used in the paper where Maximum likelihood, mean field VI in weight space and Monte Carlo Dropout.

**Summary Of The Review:**

The approach is interesting and to some degree novel. While the results look promising, further experiments are necessary to completely convince me. The model is never trained on a data set where there are external reports on the current maximal performance achievable. The rather low performance on CIFAR10 and 100 is in this context slightly alarming. If the authors, can show me references where a transformer model reached similarly low performance or if they can demonstrate the utility of their method on a data set where the performance of transformers is already well quantified I will be happy to increase my score accordingly.

---

> ### Author Response · Authors · 2022-11-09
> **Reponse (part 1/3)**
>
> Thank you for your review and suggestions on our work. We would like to address your questions/concerns below and are keen to follow up if you have any further questions.
>
> > 1. Differentiate the application of Gaussian processes for interpolation compared to classification.
>
> It is an interesting question. For sequence data, each instance consists of multiple tokens which are treated as inputs for the GP in our case (i.e. for each input sequence, we fit a deep GP over its tokens). However, the task we consider here is not about classifying tokens but about classifying sequences (our framework also allows tokens classification though). A similar task is set classification, such as point cloud classification, where each input is a set with multiple elements. One can obtain hidden representations for every element in the set and then combine them to classify the set.
>
> > 2. Compare against Liu et al., NeurIPS 2020.
>
> Here we show the results of SNGP (Liu et al. NeurIPS 2020) for CIFAR10 below. We also include results of other additional baselines (temperature scaling, KFAC-LLLA and deep ensemble) in our reply to Q2 from reviewer GcWL.
> We report mean (standard error) for each metric obtained from 5 independent runs.
>
> **In-distribution calibration:**
> |Method|Accuracy|NLL|ECE|MCE|
> |---|---|---|---|---|
> |MLE|0.7811 (0.0019)|1.3395 (0.0135)|0.0331 (0.0004)| 0.3724 (0.0103)|
> |MFVI|0.7202 (0.0024)|0.7995 (0.0073)|**0.0058 (0.0009)**|**0.0664 (0.0064)**|
> |MCD|**0.7910 (0.0015)**|0.7789 (0.0063)|0.0159 (0.0004)|0.1501 (0.0055)|
> |SNGP|**0.7837 (0.0025)**|**0.7534 (0.0031)**|0.0156 (0.0001)|0.1443 (0.0049)|
> |**SGPA**| 0.7787 (0.0024)|**0.6968 (0.0032)**|**0.0100 (0.0004)**|**0.0825 (0.0066)**|
>
> **OOD robustness on CIFAR10-C:**
>
> **Accuracy**
> |Skew intensity|1 |2|3|4|5|
> |---|---|---|---|---|---|
> |MLE|0.7254 (0.0019)|0.6632 (0.0028)|0.6139 (0.0023)|0.5494 (0.0028)|0.4863 (0.0027)|
> |MFVI|0.6606 (0.0093)|0.6109 (0.0083)|0.5703 (0.0077)|0.5266 (0.0073)|0.4778 (0.0065)|
> |MCD|**0.7327 (0.0013)**|**0.6717 (0.0015)**|**0.6229 (0.0016)**|**0.5603 (0.0019)**|**0.4982 (0.0020)**|
> |SNGP|**0.7281 (0.0053)**|**0.6676 (0.0055)**|**0.6208 (0.0044)**|**0.5622 (0.0045)**|**0.5006 (0.0042)**|
> |**SGPA**|0.7175 (0.0033)|0.6559 (0.0042)|0.6067 (0.0040)|0.5466 (0.0033)|0.4832 (0.0046)|
>
> **NLL**
> |Skew intensity|1|2|3|4|5|
> |---|---|---|---|---|---|
> |MLE|1.7906 (0.0179)|2.3460 (0.0321)|2.8899 (0.0385)|3.6899 (0.0544)|4.5003 (0.0811)|
> |MFVI|**0.9761 (0.0274)**|**1.1345 (0.0267)**|**1.2737 (0.0259)**|**1.4705 (0.0237)**|**1.7038 (0.0222)**|
> |MCD|1.1466 (0.0099)|1.5153 (0.0168)|1.8926 (0.0234)|2.4692 (0.0315)|3.1086 (0.0431)|
> |SNGP|1.0579 (0.0147)|1.3681 (0.0257)|1.6570 (0.0289)|2.0807 (0.0305)|2.5271 (0.0326)|
> |**SGPA**|**0.9603 (0.0128)**|**1.2340 (0.0213)**|**1.4972 (0.0283)**|**1.8951 (0.0383)**|**2.3812 (0.0586)**|
>
> **ECE**
> |Skew intensity|1|2|3|4|5|
> |---|---|---|---|---|---|
> |MLE|0.0425 (0.0003)|0.0530 (0.0005)|0.0617 (0.0005)|0.0734 (0.0006)|0.0846 (0.0006)|
> |MFVI|**0.0077 (0.0017)**|**0.0122 (0.0021)**|**0.0170 (0.0021)**|**0.0243 (0.0021)**|**0.0316 (0.0021)**|
> |MCD|0.0242 (0.0002)|0.0328 (0.0003)|0.0405 (0.0004)|0.0515 (0.0004)|0.0621 (0.0006)|
> |SNGP|0.0233 (0.0004)|0.0317 (0.0006)|0.0389(0.0006)|0.0487 (0.0005)|0.0588 (0.0006)|
> |**SGPA**|**0.0178 (0.0006)**|**0.0258 (0.0007)**|**0.0330 (0.0008)**|**0.0432 (0.0008)**|**0.0529 (0.0010)**|
>
> **MCE**
> |Skew intensity|1|2|3|4|5|
> |---|---|---|---|---|---|
> |MLE|0.4038 (0.0028)|0.4362 (0.0020)|0.4655 (0.0020)|0.5051 (0.0033)|0.5442 (0.0036)|
> |MFVI|**0.0598 (0.0112)**|**0.0852 (0.0150)**|**0.1125 (0.0147)**|**0.1586 (0.0147)**|**0.2011 (0.0149)**|
> |MCD|0.1902 (0.0021)|0.2402 (0.0025)|0.2807 (0.0026)|0.3413 (0.0036)|0.3949 (0.0041)|
> |SNGP|0.1854 (0.0041)|0.2337 (0.0046)|0.2710 (0.0024)|0.3229 (0.0046)|0.3798 (0.0018)|
> |**SGPA**|**0.1273 (0.0042)**|**0.1801 (0.0046)**|**0.2212 (0.0061)**|**0.2807 (0.0057)**|**0.3340 (0.0047)**|
>
> **OOD detection (OOD set: CIFAR100, SVHN, MINIIMAGENET):**
> |Method|AUROC (C100)|AUPR (C100)|AUROC (SVHN)|AUPR (SVHN)|AUROC (MINI)|AUPR (MINI)|
> |---|---|---|---|---|---|---|
> |MLE| 0.7208 (0.0029)|0.6774 (0.0029)|**0.7477 (0.0118)**|**0.8518 (0.0068)**|0.7325 (0.0016)|0.9300 (0.0004)|
> |MFVI|0.7177 (0.0055)|0.6722 (0.0050)|0.7059 (0.0195)|0.7944 (0.0134)|**0.7531 (0.0036)**|**0.9407 (0.0008)**|
> |MCD|**0.7426 (0.0016)**|**0.7046 (0.0016)**|0.6860 (0.0138)|0.8045 (0.0083)|0.7443 (0.0019)|0.9356 (0.0006)|
> |SNGP|0.7391 (0.0016)|0.7023 (0.0029)|0.7135 (0.0125)|**0.8258 (0.0083)**|0.7391 (0.0050)|0.9341 (0.0015)|
> |**SGPA**|**0.7498 (0.0010)**|**0.7111 (0.0023)**|**0.7198 (0.0085)**|0.8125 (0.0067)|**0.7501 (0.0013)**|**0.9382 (0.0004)**|
>
> Overall, SGPA achieves best performance in the sense that it consistently outperforms MLE, MCD and SNGP in terms of calibration, while unlike MFVI, it doesn’t suffer from underfitting.
>
> Due to the time constraint of the rebuttal, we are not able to obtain results of SNGP for all experiments in the paper, but we will include them in the future update.

---

> ### Author Response · Authors · 2022-11-09
> **Reponse (part 2/3)**
>
> > 3. Relatively low performance of ViTs
>
> To the best of our knowledge, **SOTA results obtained by vision transformers (ViTs) all rely on large-scale pretraining**. However, in all of our experiments, **we directly train from scratch on small datasets such as CIFAR (i.e. the initial weights are randomly initialized)**. In this scenario, the performance of ViT will be much worse and is not comparable with CNN-based architectures.
>
> ---
> **We find our results are on par with papers considering training ViT from scratch on small datasets.** Here are some references:
> > [1] “How do Vison Transformers Work?”, ICLR 2022 (https://arxiv.org/abs/2202.06709)
>
> With data augmentation, they achieve ~90% accuracy for CIFAR10 (Figure C.1 (a)) and ~66% accuracy for CIFAR100 (Figure 5 (a)). In our case, with a smaller architecture and a simpler data augmentation strategy, **we achieve ~89% accuracy for CIFAR10 (Table 8 in Appendix F) and ~60% accuracy for CIFAR100 (Table 10 in Appendix F)** (complete results of ViTs trained with data augmentation can be found in Fig. 7 - Fig. 10 in Appendix B.2).
> >[2] “Locality Guidance for Improving Vision Transformers on Tiny Datasets”, ECCV 2022 (https://arxiv.org/abs/2207.10026)
>
> With data augmentation, they achieve 58% accuracy on CIFAR100 (Fig. 1).
> >[3] “Nested Hierarchical Transformer: Towards Accurate, Data-Efficient and Interpretable Visual Understanding”, AAAI 2022 (https://arxiv.org/abs/2105.12723)
>
> In Table 1, with data augmentation they achieve 88% and 68% accuracy for CIFAR10 and CIFAR100 with DeiT-Tiny [8]. Note that DeiT uses a CNN as a teacher model to distill transformer. Without any distillation, we achieve 89% and 60% accuracy for CIFAR10 and CIFAR100 respectively, with a much smaller architecture and simpler data augmentation. All the other variants of ViTs in the table requires significant modification of the original ViT architecture to adapt to small data.
> > [4] “Sinkformers: Transformers with Doubly Stochastic Attention” AISTATS 2022 (https://proceedings.mlr.press/v151/sander22a.html)
>
> In Figure 5, they achieve <80% test accuracy on Dogs vs. Cats dataset (https://www.kaggle.com/c/dogs-vs-cats/data) with a ViT, while over 80% test accuracy can be achieved in 2008 via SVM (Table 4 in [9]).
>
> ---
> **Here we include references including general discussions about ViT’s worse performance than CNN in small data regime when trained from scratch.**
> > [5] “An Image is Worth 16X16 Words: Transformers For Image Recognition At Scale”, ICLR 2021 (https://arxiv.org/abs/2010.11929)
>
> The fourth paragraph in section 1 (at the end of page 1):
> “When trained on mid-sized datasets such as ImageNet without strong regularization, these models yield modest accuracies of a few percentage points below ResNets of comparable size …… and therefore do not generalize well when trained on insufficient amounts of data.”
> > [6] “Escaping the Big Data Paradigm with Compact Transformers”, CVPR LLID Workshop 2021 (https://arxiv.org/abs/2104.05704)
>
> In the second paragraph of page 2:
> “As a result, CNNs are still the go-to models for smaller datasets because they are more efficient, both computationally and in terms of memory, when compared to transformers. Additionally, local inductive bias shows to be more important in smaller images. They require less time and data to train while also requiring a lower number of parameters to accurately fit data”
> >[7] “Transformers in Vision: A Survey”, ACM Computing Surveys 2022 (https://dl.acm.org/doi/pdf/10.1145/3505244)
>
> Section 4.2 Large Data Requirements
>
> ---
> Unfortunately, due to the time constraint of rebuttal and the limit of computing resources, we are not able to run experiments on larger datasets like Imagenet during the rebuttal. We would like to point out that instead of achieving SOTA results in the pretraining scheme, our goal in this paper is to utilize the natural connection between multi-head attention and SVGP and based on this construct a novel probabilistic method that is able to calibrate the uncertainty in transformers. Our method can be directly applied in large-scale pretraining and it would be interesting to investigate the effectiveness of SGPA in calibrating for pretraining in the future
>
>
> References
>
> [8] “Training data-efficient image transformers & distillation through attention”, ICML 2021 (https://arxiv.org/abs/2012.12877)
>
> [9] “Machine Learning Attacks Against the Asirra CAPTCHA”, ACM 2008 (https://crypto.stanford.edu/~pgolle/papers/dogcat.pdf)

---

> ### Author Response · Authors · 2022-11-09
> **Response (part 3/3)**
>
> > 4. The approach is very similar to Set Transformers, which use a similar idea to speed up the computations. A reference would be appropriate.
>
> We agree with the reviewer that both SGPA and set transformer utilize a small number of inducing points to reduce computational cost. However, the ways of using inducing points in these two methods are different: the inducing points in set transformer are used to reduce the computational cost of full attention between all elements in a set, and it does not solve the problem of uncertainty estimation at all, there’s no Bayesian modeling concept in this approach. However, in our method, we still compute the full attention (i.e. kernel matrix) between all tokens in each sequence (since amortized inducing points are obtained by passing hidden representations of tokens through an NN). The global inducing points are introduced to avoid the computational cost of inverting the kernel matrix between amortized inducing points, which only shows up since GP is used: now the inversion is done on the kernel matrix corresponding to the global inducing points and the cost of it can be controlled by setting the number of global inducing points to use.
>
> > 5. Utilization of [] in the equations is slightly confusing. I would recommend to discard these or indicate that they solely are used to keep the subscripts separated in the text.
>
> Thanks for your suggestion on notation improvement. We will indicate this in the future update.

---

> > ### Author Response · Authors · 2022-11-16
> > **Reminder for Reviewer 3Jj8**
> >
> > Dear Reviewer 3Jj8,
> >
> > Thank you once again for your review of our work. As the discussion period is approaching its end, we would be grateful if you could confirm whether our rebuttal addressed your concerns, and let us know if any issues remain.

---

> > > ### Author Response · Authors · 2022-11-18
> > > **Reminder for Reviewer 3Jj8 (2)**
> > >
> > > Dear Reviewer 3Jj8,
> > >
> > > As the discussion period is expected to conclude today. We are therefore wondering if there is still any chance for you to let us know whether we have addressed your concerns with our rebuttal. We would very much appreciate it. Thank you!

---

> ### Author Response · Authors · 2022-11-25
> **Post Rebuttal Reminder**
>
> Dear Reviewer 3Jj8,
>
> Thank you once again for your review. We believe we have clarified most of the questions you raised. In particular, we have listed references showing what performance transformers typically achieve with our experimental setting (i.e., trained from scratch on small datasets) as requested, and we believe we achieve comparable performance. Since you mentioned you are happy to update your score according to the rebuttal, we would be grateful if you could confirm whether our rebuttal addressed your concerns. Thank you!
>
> Best regards,
> Authors

---

### Official Review · Reviewer_PkUZ · 2022-10-27

**Confidence:** 3
**Correctness:** 4
**Technical Novelty And Significance:** 3
**Empirical Novelty And Significance:** 3
**Recommendation:** 6

**Clarity, Quality, Novelty And Reproducibility:**

The quality and novelty of this work are good (see Strengths). However, the clarity of this paper may be improved. (See Weaknesses (2)-(4))


**Strength And Weaknesses:**

Strength:
* This paper is the first to consider a Gaussian process in attention block and extends SGPA in Transformer for uncertainty estimation.
* The authors carefully provide limitations of the proposed method and propose decoupled SGPA which overcomes the time and memory consumption issues.
* Derivations of the evidence lower-bound objective (ELBO) for optimizing the variational parameters in SGPA and of the decoupled SGPA for the Transformer's attention are provided.
* This method shows significant improvement in in-distribution calibration quality and OOD detection performance while achieving competitive accuracy against the standard Transformers.
* The results are well-presented and clearly discussed.

Weaknesses:
* Not mentioned other approaches that consider stochastic evaluation in the attention block for UE with Transformers. For example, [1] uses stochastic self-attention with the Gumbel-Softmax trick.
* It would be useful to see the results of OOD detection in NLP tasks, for example, for the following tasks: [2], [3].
* It is not clear how exactly the variational parameters are calculated in the decoupled SGPA in equation (11).
* It is not clear what method refers to MLE in the Experiments section. Is it a method that leverages maximum probability? If no, please, add a reference or exact formula.
* It would be interesting to see the actual computational time of the inference stage of the proposed method with 10 Monte Carlo samples in comparison with the classical MC dropout.
* The improvement of the proposed method over the MLE baseline in the OOD detection task is not stable taking into account the cost of computation.

Typos:
* On page 5 after equation (12): “viewd” -> “viewed”
* On page 8 in section 4.4: “kernel based” -> “kernel-based”

References
[1] Jiahuan Pei, Cheng Wang, and György Szarvas. 2022. Transformer uncertainty estimation with hierarchical stochastic attention. In AAAI 2022.
[2] Larson, S. et al. An Evaluation Dataset for Intent Classification and Out-of-Scope Prediction. In EMNLP 2019.
[3] Gangal, V. et al. Likelihood Ratios and Generative Classifiers for Unsupervised Out-of-Domain Detection in Task Oriented Dialog. In AAAI 2020.

**Summary Of The Paper:**

The paper proposes a Sparse Gaussian Process Attention (SGPA) for the Transformer architecture. The SGPA is used instead of the classical scaled dot-product attention and directly performs an approximation of the Bayesian inference in the attention blocks in Transformers. In addition, the authors provide decoupled SGPA to overcome the time and memory consumption limitations of the standard SGPA. This approach is evaluated on image classification, graph property regression, and NLP classification tasks. The authors show that this method is able to produce better calibrated predictions. The SGPA also helps to produce better uncertainty estimates and shows improvements in the OOD detection task in CV.

**Summary Of The Review:**

This paper provides a theoretically derived method that helps to calibrate prediction and capture better uncertainty estimates for Transformers models. The authors make a significant contribution with minor weaknesses that can be solved.

[After the review] The response by authors resolved part of my concerns and I decided to increase the score.

---

> ### Author Response · Authors · 2022-11-09
> **Response (part 1/2)**
>
> Thank you for your encouraging review and your suggestions on clarity improvements. We will address your questions below.
>
> > 1. Not mentioned other approaches that consider stochastic evaluation in the attention block for UE with Transformers. For example, [1] uses stochastic self-attention with the Gumbel-Softmax trick.
>
> Indeed [1] also introduces uncertainty in the attention block by sampling attention weights using Gumbel-Softmax. However, it is hard to reason what type of uncertainty this method captures: even with an infinite amount of data, since they still sample the attention weights each time, this uncertainty can not be explained away, therefore it doesn’t correspond to the epistemic/model uncertainty that Bayesian methods capture. Moreover, in their paper, they only consider predictive accuracy based metrics to assess the performance. The calibration of uncertainty itself is not assessed by calibration metrics such as ECE.
>
> > 2. It would be useful to see the results of OOD detection in NLP tasks, for example, for the following tasks: [2], [3].
>
> Thanks for pointing out the interesting tasks. We will consider them in the future update of the paper. Unfortunately, we will not be able to consider experiments on new tasks during the rebuttal due to the time constraint.
>
> > 3. It is not clear how exactly the variational parameters are calculated in the decoupled SGPA in equation (11).
>
> We refer the reviewer to Eq (12) for the calculation of variational parameters ($v_a^{l,h}$) for amortized inducing points (for amortized inducing points, we only need the variational mean because we consider decoupled-SGP). In each layer $l$, for head $h$, $v_a^{l,h}$ is obtained by first passing the output from previous layer ($F^l$) through a neural network ($G_{\phi^l}(\cdot)$) and then projecting it with a projection matrix $W_v^{l,h}$: $v_a^{l,h}=G_{\phi^l}(F^{l-1})W_v^{l,h}$.
> The variational parameters ($v_g^{l,h}$, $S_g^{l,h}$) for global inducing points in each layer are treated as free parameters since the inducing locations are not input-dependent.
> The computation mentioned above is also illustrated in Figure 1 (c).
>
> > 4. It is not clear what method refers to MLE in the Experiments section. Is it a method that leverages maximum probability? If no, please, add a reference or exact formula.
>
> By MLE, we mean maximizing the log-likelihood with respect to the parameters. For classification, it is equivalent to minimizing the cross-entropy loss.
>
> > 5. It would be interesting to see the actual computational time of the inference stage of the proposed method with 10 Monte Carlo samples in comparison with the classical MC dropout.
>
> Here we present the computational time for a single batch at the inference stage with 10 Monte Carlo samples for CoLA (batch size = 227) and CIFAR10 (batch size = 200) (results obtained using a single Nvidia GTX 2080 Ti GPU card). For SGPA, we first pay a one-off cost of inverting the kernel matrices related to global inducing points ($K_{k_g k_g}^{-1}$). Once this is done, we treat them as constant matrices and plug them in Eq (11). When generating samples that are passed to the next layer, we diagonalize the covariance ($\Sigma_d$) in Eq (11) to avoid the costly computation of the Cholesky factor of $\Sigma_d$.
>
> CoLA:
> | Model         | MCD   | MFVI   | SGPA   |
> |---|---|---|---|
> | Time (in s)  | 1.839  | 1.882   | 2.358   |
>
> CIFAR10:
> | Model         | MCD   | MFVI   | SGPA  |
> |---|---|---|---|
> | Time (in s)  | 0.234| 0.395 | 0.986 |
>
> The computational cost depends on the number of global inducing points used. For CoLA, we only used 5 global inducing points for each head, and the relative difference between inference times for MCD and SGPA is less than that for CIFAR10, where we use 32 global inducing points for each head.
> It is noteworthy that we haven’t done extensive hyperparameter tuning for the number of global inducing points. It is likely that SGPA can still work well with a smaller number of global inducing points. For example, for CIFAR10, we recently trained an SGPA model with 16 global inducing points for each head, and we didn’t see a considerable performance drop:
> |Accuracy|NLL |ECE | MCE|
> |---|---|---|---|
> |0.7790|0.7259|0.0119|0.0819|
>
> In this case, the inference time can be further reduced from 0.986 to 0.807.

---

> > ### Comment · Reviewer_PkUZ · 2022-11-17
> > **Thanks for the comments and additional experiments**
> >
> > Dear authors,
> >
> > I appreciate your detailed response and clarifications made. I recommend to incorporate the response into the manuscript and raise my score.

---

> > > ### Author Response · Authors · 2022-11-17
> > > **Thank you for the response and suggestions**
> > >
> > > Thank you very much for the response and suggestions. We have included the wall clock running time for SGPA in Appendix E in the revision (highlighted in blue)

---

> ### Author Response · Authors · 2022-11-09
> **Response (part 2/2)**
>
> > 6. The improvement of the proposed method over the MLE baseline in the OOD detection task is not stable taking into account the cost of computation.
>
> We agree with the reviewer that SGPA doesn’t outperform other baselines in all cases for OOD detection. Overall, SGPA still achieves better performances in a more robust manner: in all cases, it ranks either first or second (without being far away from the first). While the other three methods can outperform SGPA occasionally, there are always cases where they achieve considerably worse results than SGPA. An interesting future extension that we will investigate is to incorporate inductive bias that may help in OOD detection into the deep kernel. For example, we can include the hidden mapping distance preserving trick (Liu et al., NeurIPS 2020) in SGPA, which has been shown to be useful for uncertainty estimation in deep kernel learning.
>
> > Spelling typos
>
> Thank you for pointing out the typos and we will fix that in the future update.
>
> References
>
> Jeremiah Zhe Liu et al. Simple and principled uncertainty estimation with deterministic deep learning via
> distance awareness. In NeurIPS, 2020

---

> > ### Author Response · Authors · 2022-11-16
> > **Reminder for Reviewer PkUZ**
> >
> > Dear Reviewer PkUZ,
> >
> > Thank you once again for your review of our work. As the discussion period is approaching its end, we would be grateful if you could confirm whether our rebuttal addressed your concerns, and let us know if any issues remain.

---

### Author Response · Authors · 2022-11-10
**Summary**

We thank all reviewers for their constructive feedback. We summarize below the strength of the paper and our response to the main criticisms from the reviewers.

**Strengths & reviewers’ support:**

1. **High novelty:** In this paper we identify **the natural connection between attention and posterior mean of SVGP.** Based on this key insight, we propose **a novel method to calibrate uncertainty for transformers via deep SVGP.** All the reviewers acknowledge the novelty and relevance of our method:

> - **PkUZ:** “This paper is the first to consider a Gaussian process in attention block and extends SGPA in Transformer for uncertainty estimation.”
> - **3Jj8:** “The approach is interesting and to some degree novel. ”
> - **o7ME:** “The paper is valuable and relevant for the community, as it builds a new door to introduce probabilistic methods into Transformers via the equivalences in kernel-attention heads.”
> - **GcWL:** “The work is somehow novel and the derivation seems alright.”

2. **Improved performance** against standard transformer in our experimental setting (i.e. training from scratch on small datasets)

    By replacing scaled dot-product with a valid kernel and fitting deep GP based on deep kernel, SGPA achieves **much better calibration** than standard transformer and **competitive or even improved predictive accuracy**, when trained from scratch on small datasets. **Three reviewers (Reviewers PkUZ, 3Jj8 and GcWL) agree with us in this regard:**

> - **PkUZ:** "This method shows significant improvement in in-distribution calibration quality and OOD detection performance while achieving competitive accuracy against the standard Transformers."
> - **3Jj8:** "In the experiments of the paper, the proposed model shows comparable performance in terms of accuracy, but scores better in terms of NNL and calibration metrics on small vision and graph datasets."
> - **GcWL:** "The method seems to positively affect not only the uncertainty properties but also the accuracies".

**Main issues raised by the reviewers:**

 1. Reviewer **3Jj8** brings up concerns (Q3) about **relatively low performance of ViTs in our experiments.** It is worth noting that we train ViTs from scratch directly on CIFAR (i.e. all parameters are randomly initialized) and to the best of our knowledge, **ViTs directly trained on small datasets without large-scale pretraining are usually noticeably worse than CNN-based architectures.** In the original ViT paper [1], they also identified this issue (4th paragraph of section 1): “When trained on mid-sized datasets such as ImageNet without strong regularization, these models yield modest accuracies of a few percentage points below ResNets of comparable size …… and therefore do not generalize well when trained on insufficient amounts of data.” There are other published papers identifying the pathology of ViTs in small data regime, for example:[2] and [3].

    Moreover, **our results on CIFAR are comparable with results in many published papers in which ViTs are trained from scratch.** With data augmentation, we achieve 89% (Table 8 in Appendix F) for CIFAR10 and 60% (Table 10 in Appendix F) for CIFAR100 with vanilla ViT using scaled-dot product attention (complete results of ViTs trained with data augmentation can be found in Fig. 7 - Fig. 10 in Appendix B.2), which is on par with the following works (note that they all use deeper architecture and [4] and [6] use more advanced data augmentation strategies than ours. Moreover, [6] uses a CNN teacher to distill ViT, while we don’t consider distillation):

    ||CIFAR10|CAIFAR100|
    |---|---|---|
    |[4]|90%|66%|
    |[5]|-|58%|
    |[6]|88%|68%|

    For more detailed discussions of training ViTs from scratch in small data regime, please see our answer to **Q3 from Reviewer 3Jj8.** Due to the constraint of computational resources, we are not able to do experiments on large-scale pretraining, but we consider calibrating transformers in the pretraining stage as an interesting future work.

2. Reviewer **GcWL** asked for results from more baselines. Although the original goal of this paper is to propose novel probabilistic method rather than extensive empirical comparison, we agree such comparison can be beneficial for fully assessing the strength of our method. We include results for CIFAR10 using deep ensemble and other Bayesian methods GcWL mentioned. For in-distribution calibration, we also further include temperature scaling as a bare minimum. Among all these additional baselines, SGPA still achieves the best trade-off between predictive accuracy and calibration (for details, please see our answer to Q2 from Reviewer GcWL). Due to the time constraint of rebuttal, we are not able to include all these baselines for all experiments in the paper, but we will update the paper with these additional baselines in the future.

---

> ### Author Response · Authors · 2022-11-10
> **References**
>
> References
>
> [1] “Transformers For Image Recognition At Scale”, ICLR 2021
>
> [2] “Escaping the Big Data Paradigm with Compact Transformers”, CVPR LLID Workshop 2021
>
> [3] “Transformers in Vision: A Survey”, ACM Computing Surveys 2022
>
> [4] “How do Vison Transformers Work?”, ICLR 2022
>
> [5] “Locality Guidance for Improving Vision Transformers on Tiny Datasets”, ECCV 2022
>
> [6] “Nested Hierarchical Transformer: Towards Accurate, Data-Efficient and Interpretable Visual Understanding”, AAAI 2022

---

> ### Author Response · Authors · 2022-11-14
> **Changes to revised manuscript**
>
> We removed the misleading term “multi-output SVGP” at the end of section 3.1 (raised by Reviewer **o7ME**).
>
> ---
>
> We included more implementation details (highlighted in blue) in Appendix D (raised by Reviewer **o7ME**).
>
> ---
>
> We included wall clock running time for SGPA (highlighted in blue) in Appendix E (raised by Reviewer **PkUZ** and Reviewer **o7ME**)
>
> ---
>
> We edited discussions about how global inducing points/keys may help detect "when the model does not know" and why injecting meaningful inductive bias into deep kernel may be an interesting future direction (highlighted in blue) in Appendix C (raised by Reviewer **GcWL**).
>
> ---
>
> We fixed a few typos:
> - On page 5 after equation (12): “viewd” -> “viewed” (raised by Reviewer **PkUZ**).
> - On page 8 in section 4.4: “kernel based” -> “kernel-based” (raised by Reviewer **PkUZ**).
> - In Eq (11), (21) and (22): $K_{k_g k_g}^{-1}$ (in the parentheses of the formula for posterior covariance) -> $K_{k_g k_g}$.

---

### Decision · Program_Chairs · 2023-01-20

**Decision:**

Accept: poster

**Justification For Why Not Higher Score:**

* Concerns about practical significance (scalability, robustness, demonstration on the most popular settings)

**Justification For Why Not Lower Score:**

* Original and interesting connection between attention and posterior
* Manuscript of good quality (writing, structure)

**Metareview: Summary, Strengths And Weaknesses:**

The reviewers and meta reviewer all carefully checked and discussed the rebuttal. They thank the authors for their response and their efforts during the rebuttal phase.

The reviewers and meta reviewer all acknowledge the original and sound contribution of considering a (sparse) GP modeling in the attention layers of transformers. Similarly, they all call out the overall good quality of the manuscript (writing, structure). The nice unveiled connection between the structure of the attention and the form of the posterior is likely to inform future research (as acknowledged by the reviewers). This was regarded as the most important contribution of the submission.

The rebuttal has strengthened some aspects of the work (e.g., measure of computational overhead, more explanation about the from-scratch training, additional comparisons). However, there are still some remaining important concerns. _As a result, the reviewers and the meta reviewer are weakly inclined to accept the paper._

In particular, the authors are urged to carefully update their final manuscript with the following points:

(i)  Practical significance: The results would be more convincing in the most standard and popular settings where ViT shines, i.e., with a pre-training. Why not load an existing pre-trained checkpoint (to minimize the cost of the experiments, acknowledging the concerns of the authors) and introduce the new GP-related parameters only at fine-tuning time? (e.g., similar checkpoint surgery successfully done in [A, B]). Moreover, having experiments based on a pre-trained model would help further understand whether the gains observed are also due to an implicit regularizing effect (see Reviewer GcWL), especially important in the training-from-scratch setup.

(ii) Scalability and comparison with other approaches to capture uncertainty for ViT: In [A], it is shown that simple and scalable components (efficient ensemble and/or last-layer input-dependent noise modeling) greatly help for tasks similar to those tackled in the submission. Adding a comparison with such an approach, _at ImageNet scale_, would be important to strengthen the practical significance of the paper. This is all the more important as the proposed approach is arguably more complex (and substantially more expensive, as already measured at cifar-scale) than the components of [A]. Furthermore, hyperparameters such as the number of inducing points are difficult to set in a reliable way at a larger scale.

If the paper was submitted to a journal, it would be accepted conditioned on those key changes, the meta-reviewer thus expects all those changes to be carefully implemented.

**References**

[A] Tran et al., 2022, Plex: Towards Reliability using Pretrained Large Model Extensions.

[B] Allingham et al., 2022, Sparse MoEs meet Efficient Ensembles.


**Note From Pc:**

if the above contains the word "oral" or "spotlight" please see: "oral" presentation means -> notable-top-5% and "spotlight" means -> notable-top-25%. As stated in our emails, we are disassociating presentation type from AC recommendations

**Summary Of Ac-Reviewer Meeting:**

The meta reviewer and three (out of the four) reviewers met on 12/8.

In the meeting, we quickly converged to a “weak accept“ decision by collectively articulating the two main points listed in the meta review, namely:

(i) The paper would be more convincing if it could show results in the most popular setting of ViT, with pretraining.

(ii) Scalability and robustness of the method with respect to scale (in other words, if one faces a given problem, say at ImageNet scale, and we want to use a robust ViT with reliable uncertainty quantification, would the proposed method the go-to method?)

Moreover, everyone agrees with the fact that the connection unveiled in the paper, between attention and posterior, is interesting and original.

This is the tensions between those two threads that led to the weak accept.